# OSWORLD: Benchmarking Multimodal Agents for Open-Ended Tasks in Real Computer Environments

Tianbao Xie [h] Danyang Zhang [h] Jixuan Chen [h] Xiaochuan Li [h] Siheng Zhao [h] Ruisheng Cao [h]

Toh Jing Hua [h] Zhoujun Cheng [h] Dongchan Shin [h] Fangyu Lei [h] Yitao Liu [h] Yiheng Xu [h]

Shuyan Zhou [c] Silvio Savarese [s] Caiming Xiong [s] Victor Zhong [w] Tao Yu [h]

[h] The University of Hong Kong    [c] Carnegie Mellon University
[s] Salesforce Research    [w] University of Waterloo

## Abstract

Autonomous agents that accomplish complex computer tasks with minimal human interventions can significantly enhance accessibility and productivity of human-computer interactions. Existing benchmarks either lack interactive environments or are limited to specific applications/domains, failing to reflect the diversity and complexity of real-world computer use and limiting agent scalability. We introduce OSWORLD, the *first-of-its-kind scalable real computer environment* for multimodal agents, supporting task setup, interactive learning, and execution-based evaluation of open-ended computer tasks across arbitrary applications in Ubuntu, Windows, and macOS. Using OSWORLD, we create a benchmark of 369 tasks involving real web and desktop apps in open domains, OS file I/O, and multi-app workflows. Each example derives from real-world use cases and includes detailed setup and execution-based evaluation for reproducibility. Extensive evaluation of state-of-the-art LLM/VLM agents on OSWORLD reveals deficiencies in their ability to serve as computer assistants. While humans accomplish 72.4% of the tasks, the best agents achieve <12.2%, struggling with GUI grounding and operational knowledge. Comprehensive analysis using OSWORLD provides valuable insights for developing multimodal generalist agents that were not possible with previous benchmarks. Implementation and experiments are at `https://os-world.github.io`.

## 1 Introduction

Humans interact with computers to perform essential tasks in the digital realm, including web browsing, video editing, file management, data analysis, and software development. These task workflows often involve multiple applications through graphical user interfaces (GUI) and command line interfaces (CLI). Autonomous agents powered by large vision-language models (VLMs) can revolutionize how we interact with computer environments [32, 48, 1]. By following natural language instructions, these agents can make computers more accessible and vastly increase human productivity.

A major challenge in developing multimodal agents is the absence of a benchmark that covers interactive, diverse, and complex real-world computer use across operating systems, interfaces, and applications. Prior benchmarks that provide demonstration datasets without executable environments [12, 44, 25] assume a single solution for each task and limit potential research in interactive learning and real-world exploration. Prior work with executable environments simplify the observation and action spaces of agents and limit task scopes to specific applications/domains such as navigation of specific websites [48, 34, 63, 71], coding [62] and the combination [36, 59, 38]. These restricted environments do not fully reflect real-world computer use, as they do not evaluate scenarios that require navigating between applications and interfaces in open domains (Fig. 1).

38th Conference on Neural Information Processing Systems (NeurIPS 2024) Track on Datasets and Benchmarks.

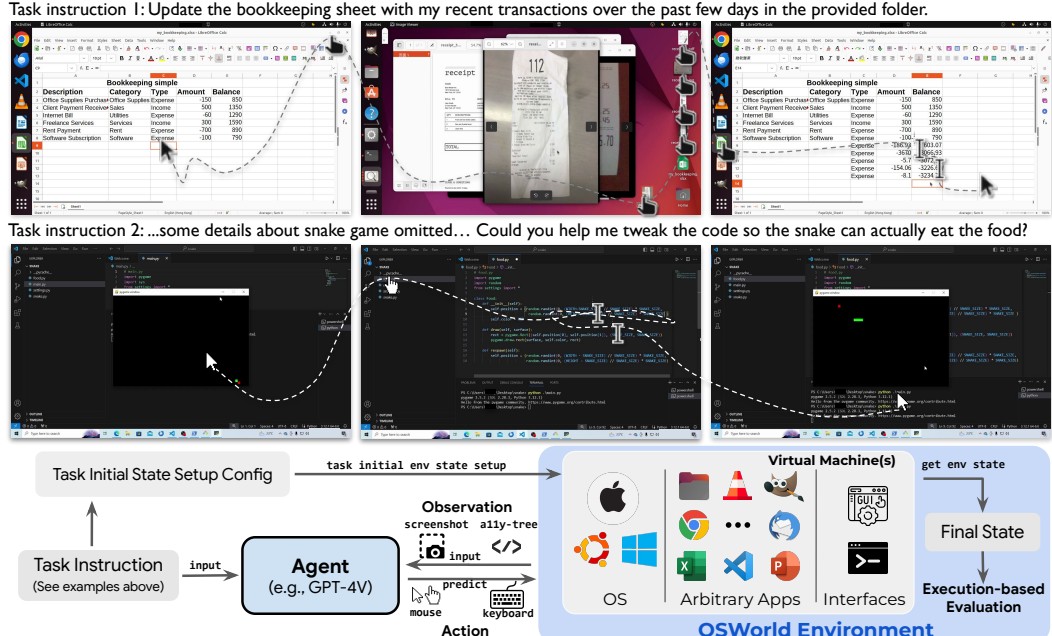

Figure 1: OSWORLD is a *first-of-its-kind scalable real computer env* for multimodal agents, supporting task setup, execution-based evaluation, and interactive learning across systems. It is a unified environment for evaluating *open-ended* computer tasks that involve arbitrary apps. Using OSWORLD, we create a benchmark of 369 real-world tasks with reproducible setup and evaluation scripts.

To address this gap, we introduce OSWORLD, the *first-of-its-kind scalable, real computer environment* for developing multimodal agents capable of executing a wide range of real computer tasks beyond isolated interfaces and applications. This executable environment allows free-form raw keyboard and mouse control of real computer applications and supports initial task state configuration, execution-based evaluation, and interactive learning across mainstream operating systems (Ubuntu, Windows, macOS). OSWORLD enables evaluation of *open-ended* computer tasks that involve arbitrary applications, ranging from image viewing to software functionality integration and programming (Fig. 1). OSWORLD serves as a unified, real computer environment that allows users to define their agent tasks without the need to build application/domain-specific simulated environments.

Building upon OSWORLD, we create a benchmark with 369 real-world tasks that involve widely-used web and desktop apps in open domains, OS file I/O, and multi-app workflows through both GUI and CLI. Each example is based on real-world use cases and often requires interactions with multiple applications and interfaces. To ensure reliable, reproducible evaluation, 9 authors with computer science backgrounds carefully annotate each example with an initial state setup configuration to simulate in-progress human work and a custom execution-based evaluation script to verify task completion. Our benchmark has 134 unique evaluation functions — significantly larger than prior work [71] — showcasing the complexity, diversity, and evaluation challenges of tasks in our benchmark. The human performance study indicates that task examples from OSWORLD are more time-consuming and challenging compared to those in prior work.

We evaluate state-of-the-art LLM and VLM agent baselines, including GPT-4V [43], Gemini [54, 45], Claude-3 Opus [3] and Qwen-Max [5], as well as Mixtral [23], Llama-3 [39] and CogAgent [21] from the open-source community. The performance of these experiments ranges from 0.99% to 12.24%, with subsets of applications reaching 0%, for workflow tasks that involve cooperation from multiple apps, the highest performance of the baseline agent is 6.57%. This indicates that current LLMs and VLMs are far from capable of serving as computer assistants (§4.2). Results also show that while additional knowledge such as the accessibility tree and Set-of-Mark (§4.1) can be helpful, it can also lead to potential misguidance and varies across models. Finally, we find that VLM-based agents struggle to ground screenshots to predict precise coordinates for actions, tend to predict repetitive actions, are unable to handle noise from unexpected application windows and exhibit limited knowledge of basic GUI interactions and domain-specific features of apps (§5, §D.4).

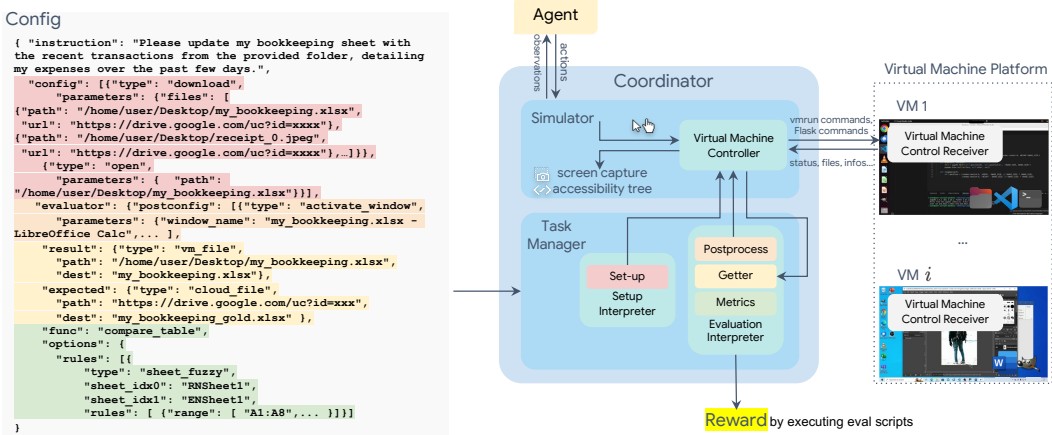

Figure 2: Overview of OSWORLD infrastructure. The environment uses a config file for initializing tasks (red), agent interaction, post-processing upon agent completion (orange), retrieving files and information (yellow), and executing the evaluation function (highlighted in green). Environments can run in parallel on a single host machine for learning or evaluation. Headless operation is supported.

## 2 OSWORLD Environment

In this section, we will introduce the task definition of autonomous agents, the components and implementation of the OSWORLD environment, and the supported observation and action spaces.

### 2.1 Task Definition

An autonomous digital agent task can be formalized as a Goal-Augmented Partially Observable Markov Decision Process (GA-POMDP) $(\mathcal{S}, \mathcal{O}, \mathcal{A}, \mathcal{T}, \Omega, r, \gamma, \rho_0, \mathcal{G}, p_g, \phi)$ where $\mathcal{S}$ is the full state space (including hidden system states), $\mathcal{O}$ is the observation space (§2.3, what's visible or accessible to the agent), $\mathcal{A}$ is the action space (§2.4), $\mathcal{T} : \mathcal{S} \times \mathcal{A} \rightarrow \mathcal{S}$ is the transition function, $\Omega$ is the observation function, $r : \mathcal{S} \times \mathcal{A} \times \mathcal{G} \rightarrow \mathbb{R}$ is the reward function, $\gamma$ is the discount factor, $\rho_0$ is the initial state distribution, $\mathcal{G}$ is the space of goals (instructions in our case), $p_g$ is the distribution of desired goals (instructions), and $\phi : \mathcal{O} \rightarrow \mathcal{G}$ is a mapping function from observations to goals. Given current observation $o_t \in \mathcal{O}$ (a natural language instruction observation and a screenshot, accessibility a11y tree, or their combination according to facilities available), an agent generates executable action $a_t \in \mathcal{A}$ (*e.g.*, clicking on the certain pixel of the screen — `.click(300, 540, button='right')`, press key combination — `.hotkey('ctrl', 'alt', 't')`), which results in a new state $s_{t+1} \in \mathcal{S}$ (*e.g.*, current computer state) and a new partial observation $o_{t+1} \in \mathcal{O}$. The interaction loop repeats until an action that marks termination (`DONE` or `FAIL`, see Sec. 2.4) is generated or the agent reaches the max number of steps (*e.g.*, 15 in our experiments). OSWORLD implements an execution-based reward function $r : \mathcal{S} \times \mathcal{A} \times \mathcal{G} \rightarrow [0, 1]$. The reward function awards a value of 1 or a positive decimal under 1 at the final step if the state transitions meet the expectations of the task objective (i.e., the goal is successfully achieved or partially achieved), or if the agent accurately predicts failure for an infeasible task. In all other scenarios, it returns 0.

### 2.2 Real Computer Environment Infrastructure

OSWORLD is an executable and controllable environment that supports task initialization, execution-based evaluation, and interactive learning in real operating systems (*e.g.*, Ubuntu, Windows, macOS) using virtual machines (VMs), shown in Fig. 2. VMs offer safe, isolated, and resettable/reversible (via snapshotting) environments that prevent irreversible damage to the real host machine.

**Initialization**   Prior to agent interactions, we initializing the VM environment via config file. This includes downloading files, opening software, and adjusting interface layout. Many real-world assistance scenarios occur not at the beginning of digital activities, such as right after launching an application or starting the computer, but rather at intermediate stages, such as when certain software

Table 1: Examples of annotated evaluation scripts, which involve retrieving data from configuration files, environment, and cloud, and executing functions to obtain results and assess correctness.

| Initial State | Task Instruction | Evaluation Script (Simplified) |
|---|---|---|
|  | *Can you help me clean up my computer by getting rid of all the cookies that Amazon might have saved?* | `cookie_data = get_cookie_data(env)`
`rule = {"type":"domains",`
`"domains":[".amazon.com"]}`
`is_cookie_deleted(cookie_data, rule)` |
|  | *Rename "Sheet 1" to "LARS Resources". Then make a copy of it. Place the copy before "Sheet 2" and rename it by appending a suffix "(Backup)", ...* | `result = get_file(env)`
`expected = get_file(cloud)`
`rules = [{"type":"sheet_name"},`
`         {"type":"sheet_data",`
`          "sheet_idx0":0,`
`          "sheet_idx1":1}...]`
`compare_table(result, expected, rules)` |
|  | *I've drafted an e-mail reminder for those who haven't paid tuition. Please help me to check out their e-mails from the payment record and add to the receiver field.* | `tree = get_a11y_tree(env)`
`rules = [{"selectors":`
`    ["tool-bar[attr|id=MsgHeadersToolbar]`
`    label[name=To]`
`    [attr|class=\"address-pill\"]>`
`    label[attr|class=\"pill-label\"]`
`    [name*=\"fox@someuniversity.edu...]`
`check_a11y_tree(tree, rules)` |

is already open or the computer has experienced a crash. Therefore, we reproduce these intermediate states during the initialization to replicate real-world scenarios. See B.5 for more details.

**Evaluation**    After agent interactions, we post-process the environment during the evaluation phase. This includes activating certain windows, saving some files for easy retrieval of information, and acquiring files and information for evaluation such as the final spreadsheet file for spreadsheet tasks, cookies for Chrome tasks. Finally, we apply the appropriate evaluation functions and parameters. We construct a vast collection of functions that make final wrangling and retrieve files and data information of varying types, categories, and granularities from the cloud and software from virtual machines as well as evaluation functions covering different aspects and their combinations, inputting this information as parameters to assess the outcomes. Tab. 1 illustrates evaluation processes including extracting cookie data, fetching files from both virtual machines and cloud services, retrieving the current interface's accessibility tree, and validating success by checking cookie deletions, table accuracy, and interface access. See more in B.6.

## 2.3    Observation Space

The observation space in OSWORLD contains the same **desktop screenshot** that human users perceive. This includes the mouse's position and shape, application windows, files, and folders that are opened in different sizes and orders. Also, similar to previous agent-building web and mobile research [34, 31, 12, 71] that provide and support the use of the webpage's DOM and app's view hierarchy, OSWORLD also provides **accessibility (a11y) tree** which can support additional information for modeling. These raw observations allow rich interactions between multiple applications but induce challenges in long-horizon decision-making from high-resolution images (*e.g.*, 4k screenshots) and structured long text (*e.g.*, accessibility trees). A.2 describes observation space in more detail.

## 2.4    Action Space

Action space $\mathcal{A}$ in OSWORLD encompasses all mouse and keyboard actions, including movement, clicks (left-key, right-key, multiple clicks), dragging, keystrokes, hotkeys, and others, covering all human-computer action space. Some action examples are shown on the left and the complete action list can be found in A.3. Timing is also crucial, as highlighted in previous studies on mobile devices [55], as well as the ability to determine whether a task is infeasible or completed. Therefore, we add three special actions named WAIT, FAIL, and DONE to enhance the aforementioned action spaces.

Previous efforts towards creating domain-specific agents, such as MiniWoB++ [48, 34], CC-Net [22], and WebArena [71, 26], have defined action spaces that include clicks and typing, as well as some actions specially designed for web browsing. However, they do not model all possible actions on a computer, leading to limitations when attempting actions like right-clicking and clicking with the `ctrl` key held to select items. This imposes an upper bound on agent learning capabilities.

Table 2: Example mouse and keyboard actions $\mathcal{A}$ in OSWORLD. See App. A.3 for the complete list.

| Function | Description |
|---|---|
| `moveTo(x, y)` | Moves the mouse to the specified coordinates. |
| `click(x, y)` | Clicks at the specified coordinates. |
| `write('text')` | Types the specified text at the current cursor location. |
| `press('enter')` | Presses the Enter key. |
| `hotkey('ctrl', 'c')` | Performs the Ctrl+C hotkey combination (copy). |
| `scroll(200)` | Scrolls up by 200 units. |
| `dragTo(x, y)` | Drags the mouse to the specified coordinates. |
| `keyDown('shift')` | Holds down the Shift key. |
| `keyUp('shift')` | Releases the Shift key. |
| `WAIT` | Agent decides it should wait. |
| `FAIL` | Agent decides the task is infeasible. |
| `DONE` | Agent decides the task is finished. |

## 3  OSWORLD Benchmark

The OSWORLD benchmark encompasses 369 real computing tasks defined and executed on Ubuntu, as well as 43 tasks on Windows. The environment preparation, annotation process, data statistics, and human performance are described in this section.

### 3.1  Operating System and Software Environments

OSWORLD supports the development of automated computer agents across real operating systems like Windows, macOS, and Ubuntu, focusing on Ubuntu for its open-source advantages and accessible APIs for comprehensive example creation and task evaluation. For Windows, we provide annotated examples targeting similar application functionalities. This framework is designed for open-domain tasks involving multiple applications and interfaces, such as GUIs and CLIs. It prioritizes a balanced benchmark across eight key applications: Chrome, VLC, Thunderbird, VS Code, LibreOffice suite (Calc, Writer, Impress), GIMP, and essential OS utilities (terminal, file manager, image viewer, PDF viewer), highlighting the need for varied operational skills, including commonsense reasoning, software navigation, and precise input control. Further details are available in B.1 and B.2.

### 3.2  Tasks

We create a suite of 369 real-world computer tasks on Ubuntu from diverse sources such as forums, tutorials, and guidelines to demonstrate open-ended task creation within OSWORLD. Each example is carefully annotated with a natural language instruction, a setup configuration with corresponding files and setup actions for environment initialization, and a manually crafted evaluation script to check if the task is successfully executed. We also adapt 43 tasks from the Ubuntu set for analytic usage on Windows. Overall, it took 9 computer science students (all student authors) over 3 months, consuming approximately 1800 man-hours (650 hours on single-app tasks, 750 hours on workflow tasks and 400 hours for double-checking).

**Task instructions and scenarios**  We collect realistic examples from diverse sources including official guidelines & tutorials, video pieces giving tips and tutorials on the Internet (TikTok,

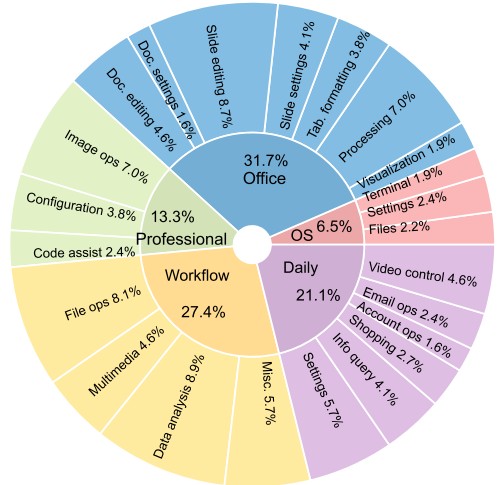

Figure 3: Distribution of task instructions in OS-WORLD by app domains and operation types.

YouTube), how-to websites (WikiHow), Q&A forums ( Reddit, Quora, Superuser, StackOverflow), formal video courses (Coursera, Udemy), and publicly-available personal blogs & guidelines. B.3 details resources used in our benchmark. We select examples by their popularity, helpfulness, and diversity, revealed by view counts and votes. While descriptions of single-application tasks are easily found, those of tasks that involve the coordination of multiple applications are scarce. Therefore, we authors combine existing examples and designed examples inspired by daily-life scenarios to compile the tasks. The instructions and task-related files are then crafted from these real-world guidelines and questions by the authors. After selection, each example is cross-checked by two other authors on the feasibility, ambiguity, and alignment with the source. We not only collect feasible tasks, but also tasks inherently infeasible due to feature deprecation or hallucinated features raised by real users, which results in 30 infeasible examples in our benchmark. Additionally, to demonstrate the unification ability of OSWORLD for the creation of open-ended computer tasks, we also integrate 84 examples from other benchmarks focusing on single-application or domain-specific environments such as NL2Bash [33], Mind2Web [12], SheetCopilot [29], PPTC [18], and GAIA [40]. Refer to B.4 for more details and B.7 for sampled examples for the showcase. A total of about 400 man-hours were spent to collect these examples. §2.2 outlines the procedure for creating config and evaluation for tasks. Initial state design took 1 man-hour per example and is detailed in B.5. Evaluation design took two man-hour per example and is detailed in B.6.

**Quality control**   After annotation, each example is attempted by two authors who did not participate in its annotation, acting as agents to complete the task. This process evaluates the current example's quality and provides feedback to the annotators (such as unclear instructions or inability to complete the task, crashes in corner cases, serious instances of false positives and negatives, *etc*.), and involves joint revisions and supplements. During experiments for human performance and baselines, we further fixed examples found to have issues, dedicating over 400 man-hours for four rounds of checks.

### 3.3   Data Statistics

**Statistics**   To facilitate analysis, we cluster the examples into software categories. Specifically, these categories include OS, Office (LibreOffice Calc, Impress, Writer), Daily (Chrome, VLC Player, Thunderbird), Professional (VS Code and GIMP), and Workflow (tasks involving multiple apps). The main statistics of OSWORLD are presented in Tab. 3 and Fig. 3, showcasing the outline and a broad spectrum of tasks. Specifically, OSWORLD contains a total of 369 tasks (and an additional 43 tasks on Windows for analysis), with the majority (268 tasks or 72.6%) aiming at single application functionalities and a section of workflow-related tasks (101 tasks or 27.4%). We also consider infeasible examples, totaling 30 tasks or 8.1% of the dataset. Additionally, a total of 84 tasks (22.8%) are integrated from related datasets. The

Table 3: OSWORLD statistic. Supp. refers to Windows tasks that are usable only after activation due to copyright.

| Statistic | Number |
|---|---|
| Total tasks (Ubuntu) | 369 (100%) |
| - Multi-App Workflow | 101 (27%) |
| - Single-App | 268 (73%) |
| - Integrated | 84 (23%) |
| - Infeasible | 30 (8%) |
| Supp. tasks (Windows) | 43 |
| Initial States | 302 |
| Eval. Scripts | 134 |

final dataset incorporates 302 distinct initial states and 134 different evaluation scripts, underscoring the comprehensive approach towards evaluating the tasks' complexity and requirements. More statistic details are available in B.4.

**Comparison with existing benchmarks**   Tab. 4 compares OSWORLD to prior benchmarks. First, instead of focusing on specific computer applications such as a browser [71, 12], OSWORLD utilizes raw **multimodal** observations and keyboard/mouse actions used by humans, which are universal across different applications and allows the development of generalizable agents. Second, instead of providing static demonstrations, OSWORLD **executable environment** supports agent exploration during learning and evaluation — behavior critical in generalizing to new applications. Third, instead of focusing on interactions within a single application, OSWORLD considers **cross-app** interactions found in real-world computer usage. Fourth, instead of limiting to a single task type with a success definition, OSWORLD provides example-wise, **execution-based evaluation** for tasks. Specifically, the total of 134 unique execution-based evaluation functions in our benchmark is significantly more than previous work, demonstrating the complexity, diversity, and evaluation challenges of OSWORLD tasks. Finally, instead of focusing on clean initialization, OSWORLD tasks require operation from **intermediate initialization**, as is typical in realistic computer usage.

Table 4: Comparison of different environments for benchmarking digital agents.

| | # Instances (# Templates) | Control. Exec. Env.? | Environment Scalability? | Multimodal Support? | Cross-App? | Intermediate Init. State? | # Exec.-based Eval. Func. |
|---|---|---|---|---|---|---|---|
| GAIA [40] | 466 | ✗ | - | ✗ | ✗ | ✗ | 0 |
| MIND2WEB [12] | 2350 | ✗ | - | ✓ | ✗ | ✓ | 0 |
| WEBLINX [37] | 2337 | ✗ | - | ✓ | ✗ | ✓ | 0 |
| PIXELHELP [31] | 187 | ✗ | - | ✓ | ✗ | ✗ | 0 |
| METAGUI [52] | 1125 | ✗ | - | ✓ | ✗ | ✗ | 0 |
| AITW [44] | 30k | ✗ | - | ✓ | ✗ | ✓ | 0 |
| SCREENAGENT [42] | 70 | ✗ | - | ✓ | ✗ | ✓ | 0 |
| OMNIACT [25] | 9802 | ✗ | - | ✓ | ✗ | ✓ | 0 |
| AGENTBENCH [36] | 1091 | Multi-isolated | ✗ | ✗ | ✗ | ✗ | 7 |
| INTERCODE [62] | 1350 (3) | Code | ✗ | ✗ | ✗ | ✗ | 3 |
| MINIWOB++ [34] | 125 | Web | ✗ | ✓ | ✗ | ✗ | 125 |
| WEBSHOP [63] | 12k (1) | Web | ✗ | ✓ | ✗ | ✗ | 1 |
| WEBARENA [71] | 812 (241) | Web | ✗ | ✓ | ✗ | ✗ | 5 |
| VWEBARENA [26] | 910 (314) | Web | ✗ | ✓ | ✗ | ✗ | 6 |
| WORKARENA [13] | 23k (29) | Web | ✗ | ✓ | ✗ | ✓ | 7 |
| WIKIHOW [66] | 150 (16) | Mobile | ✗ | ✓ | ✗ | ✗ | 16 |
| ASSISTGUI [17] | 100 | ✗ | ✗ | ✓ | ✗ | ✓ | 2 |
| OSWORLD | 369 | Computer | ✓ | ✓ | ✓ | ✓ | **134** |

## 3.4 Human Performance

We conduct human evaluations on each example in our dataset, with annotators being computer science major college students who possess basic software usage skills but have not been exposed to the samples or software before. We recorded the time required to complete each example and whether their completion of the example was correct. For comparison, we also sampled 100 examples from WebArena [71] under the same evaluation setup.

As illustrated, tasks from our dataset generally required more time to complete, with a median completion time of 111.94 seconds (compared to 35.38 seconds in WebArena), and a significant number of examples distributed at 900 seconds or even more. In terms of accuracy, the human

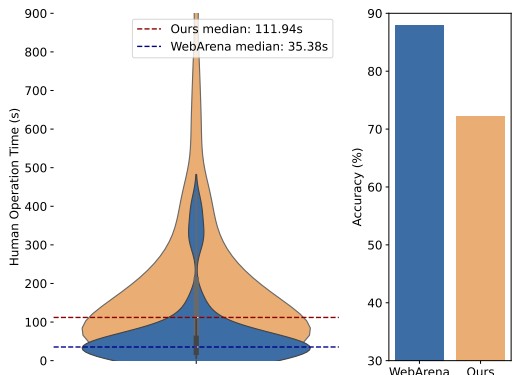

Figure 4: Human operation time and accuracy on OSWORLD and WebArena.

performance on our tasks was approximately 72.36%, significantly lower than the 88% observed on the pure web task dataset. These findings highlight the complexity and challenge of tasks in our dataset, which demand more time and effort. The lower accuracy rate further indicates that our tasks require a higher level of understanding and proficiency, underscoring the need for advanced models and techniques to tackle them effectively.

## 4 Benchmarking LLM and VLM Agent Baselines

In this section, we present the implementation details and experimental settings for several state-of-the-art LLM and VLM agent baselines on OSWORLD benchmark, as well as their performance.

### 4.1 LLM and VLM Agent Baselines

We evaluate state-of-the-art open-source LLMs and VLMs such as Mixtral [23] and Llama-3 [39], and closed-source ones such as GPT, Gemini, Claude on OSWORLD. We also explore methods such as the Set-of-Marks aided approach [61, 14], which has been demonstrated to improve spatial capabilities for visual reasoning. For each method, we provide the 3 most recent observation-action pairs and generate actions with the temperature of 1.0 and top-p of 0.9. The prompts used in the experiments are provided in C.1. We request the agents to complete the tasks within a max step limit of 15, which is enough for most tasks. We present a summary of the results in Tab. 5 and analysis in

Table 5: Success rates of baseline LLM and VLM agents on OSWORLD, grouped by task categories: OS, Office (LibreOffice Calc, Impress, Writer), Daily (Chrome, VLC Player, Thunderbird), Professional (VS Code and GIMP) and Workflow (tasks involving multiple apps), for gaining insights from interfaces and operation logic. See C.1 and C.6 for more details.

| Inputs | Model | Success Rate (↑) | | | | | |
|---|---|---|---|---|---|---|---|
| | | OS | Office | Daily | Profess. | Workflow | Overall |
| A11y tree | Mixtral-8x7B | 12.50% | 1.01% | 4.79% | 6.12% | 0.09% | 2.98% |
| | Llama-3-70B | 4.17% | 1.87% | 2.71% | 0.00% | 0.93% | 1.61% |
| | GPT-3.5 | 4.17% | 4.43% | 2.71% | 0.00% | 1.62% | 2.69% |
| | GPT-4 | 20.83% | 3.58% | 25.64% | 26.53% | 2.97% | **12.24%** |
| | Gemini-Pro | 4.17% | 1.71% | 3.99% | 4.08% | 0.63% | 2.37% |
| | Gemini-Pro-1.5 | 12.50% | 2.56% | 7.83% | 4.08% | 3.60% | 4.81% |
| | Qwen-Max | 29.17% | 3.58% | 8.36% | 10.20% | 2.61% | 6.87% |
| | GPT-4o | 20.83% | 6.99% | 16.81% | 16.33% | **7.56%** | 11.36% |
| Screenshot | CogAgent | 4.17% | 0.85% | 2.71% | 0.00% | 0.00% | 1.11% |
| | GPT-4V | 12.50% | 1.86% | 7.58% | 4.08% | **6.04%** | 5.26% |
| | Gemini-ProV | 8.33% | 3.58% | 6.55% | 16.33% | 2.08% | **5.80%** |
| | Gemini-Pro-1.5 | 12.50% | 6.99% | 2.71% | 6.12% | 3.60% | 5.40% |
| | Claude-3-Opus | 4.17% | 1.87% | 2.71% | 2.04% | 2.61% | 2.42% |
| | GPT-4o | 8.33% | 3.58% | 6.07% | 4.08% | 5.58% | 5.03% |
| Screenshot + A11y tree | CogAgent | 4.17% | 0.85% | 2.71% | 0.62% | 0.09% | 1.32% |
| | GPT-4V | 16.66% | 6.99% | 24.50% | 18.37% | 4.64% | **12.17%** |
| | Gemini-ProV | 4.17% | 4.43% | 6.55% | 0.00% | 1.52% | 3.48% |
| | Gemini-Pro-1.5 | 12.50% | 3.58% | 7.83% | 8.16% | 1.52% | 5.10% |
| | Claude-3-Opus | 12.50% | 3.57% | 5.27% | 8.16% | 1.00% | 4.41% |
| | GPT-4o | 41.67% | 6.16% | 12.33% | 14.29% | **7.46%** | 11.21% |
| Set-of-Mark | CogAgent | 4.17% | 0.00% | 2.71% | 0.00% | 0.53% | 0.99% |
| | GPT-4V | 8.33% | 8.55% | 22.84% | 14.28% | **6.57%** | **11.77%** |
| | Gemini-ProV | 4.17% | 1.01% | 1.42% | 0.00% | 0.63% | 1.06% |
| | Gemini-Pro-1.5 | 16.67% | 5.13% | 12.96% | 10.20% | 3.60% | 7.79% |
| | Claude-3-Opus | 12.50% | 2.72% | 14.24% | 6.12% | 4.49% | 6.72% |
| | GPT-4o | 20.83% | 3.58% | 3.99% | 2.04% | 3.60% | 4.59% |
| Human Performance | | 75.00% | 71.79% | 70.51% | 73.47% | 73.27% | 72.36% |

Sec. 4.2. We implement the following four types of input settings on LLM and VLM: **Accessibility tree**, **Screenshot**, **Screenshot + accessibility tree**, and **Set-of-Marks**. Details see App. C.3.

## 4.2 Results

**LLMs and VLMs are still far from being digital agents on real computers.** Table 5 shows that screenshots-only agents that generate keyboard/mouse actions via `pyautogui` achieve 5.26% to 5.80% success rate (VLMs GPT-4V, Gemini-Pro-vision) while the text-only agents using using a11y tree as input achieve 2.37% to 12.24% (LLMs GPT-4, GPT-4o). These results from state-of-the-art VLMs and LLMs significantly trail the performance of humans not familiar with the software (72.36%), which indicates further research is required to develop capable digital assistants. While Claude-3 Opus is competitive with GPT-4V on common benchmarks [2], it underperforms GPT-4V significantly as a digital agent in OSWORLD. D.4 present qualitative analysis and infer reasons.

**Agents have much higher variance than humans in different types of computer tasks.** Tab. 5 shows that agent performance varies significantly across different software types, performing better on CLI-oriented interfaces (such as OS-type tasks) compared to GUI-oriented interfaces (such as Office tasks involving clicks on spreadsheet interfaces and document processing). Moreover, the CLI vs. GUI gap between models and settings is inconsistent, with some >20%. Similarly, performance on workflow-type tasks involving multiple software (<5%) significantly trails single software performance. Unlike agent performance, human performance is consistent across these

tasks, fluctuating around 70% with <5% deviation. This suggests that the way humans understand and complete tasks may differ significantly from the current logic and methods based on LLMs and VLMs.

**A11y tree and SoM's effectiveness varies by models.** The a11y tree contains some attribute information of visible elements, including window position and size, as well as some semantic labels of the window. The performance gap illustrated in Table 5 between GPT-4V and Claude-3 with additional a11y tree information and under a pure screenshot setup suggests that it still has significant room for improvement in accurately perceiving and reasoning GUI elements. Conclusions are reversed for Gemini-Pro. While applying SoM setting, there is a decline for GPT-4V in performance compared to directly providing the model with screenshots and a11y tree inputs, which contradicts the widely shown effectiveness of SoM in classic image understanding tasks [61], as well as in application areas like web agents [70, 20]. We speculate that this is due to the tasks performed within operating systems having higher resolution and much more elements, (*e.g.*, the cells in a spread table), leading to a significant amount of noise that counteracts the auxiliary role of bounding boxes. Some tasks also require coordinate-level operations, which cannot be modeled by SoM bounding boxes.

**VLM agents with screenshot-only setting show lower performance, but it should be the ultimate configuration in the long run.** The setting that relies solely on screenshots exhibits the lowest performance, at only 5.26%, among all. Despite the performance, it is worth mentioning that this is the only configuration that does not require additional information, such as an a11y tree, making it concise and in alignment with intuitive human perception since the a11y tree may not be well-supported across all software or cannot be obtained under noisy conditions (*e.g.*, when the agent is restricted to viewing the computer through peripheral screens), and the massive amount of tokens contained in the a11y tree (even just the leaf nodes can have tens of thousands of tokens) can also impose an additional inference burden on the model. Future work on purely vision-based agents could lead to stronger generalization capabilities, efficiency, and, ultimately, the potential for integration with the physical world on a larger scale.

# 5 Analysis

**Higher resolution typically improves performance** Despite the availability of high-res displays, most VLMs are trained on far lower resolutions. We evaluate performance using screenshot-only and SoM by down-sampling the original resolution by 0.2-0.8 (Figure for 10% of examples on the right). The output coordinates of the model for

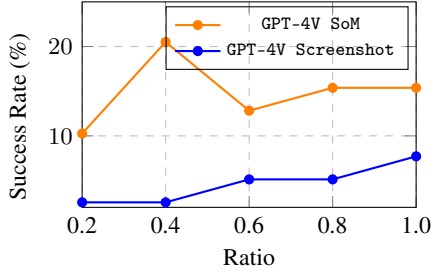

the screenshot setting are still expected to align with the original resolution (*i.e.*, 1080P). Resolution impact on performance is shown on the right (for a subset of 10% of examples). Screenshot-only performance improves with higher resolution, which may arise from the discrepancy between the downsampled input resolution and the coordinates of the output (which is for the original resolution). In contrast for SoM, a reduction to 768×432 (down-sampling 0.4) improves performance and further reduction in resolution to 0.2 noticeably degrades performance.

**Longer text-based trajectory history context improves performance, unlike screenshot-only history, but poses efficiency challenges** We include current and past `N` rounds of observations and actions in the constructed prompts (see App. C.1 for more details) to explore the impact of context length on agent performance. We set `N` to 1, 2, 3, and all where we put as much context as we can. The results (on 10% of examples) on the right show the performance increase with more history context for

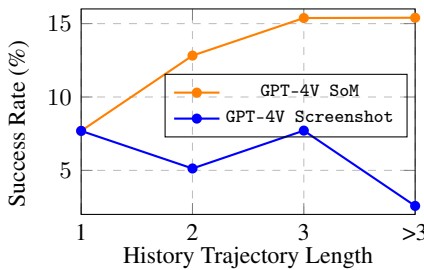

SoM. Future work on constructing models with enhanced capabilities for longer context support and understanding reasoning, improving model efficiency, and designing new agent architectures for efficient memory storage will have a significant impact on digital agents. However, we also note that the inclusion of additional trajectory history does not enhance performance under the pure screenshot

setting. This suggests that contemporary advanced VLMs might not be as adept at extracting robust contextual information from images as they are from textual data.

**VLM agents struggle with perturbation of application windows size/position and irrelevant information** We consider the best SoM setting and a subset of 28 OS-WORLD tasks that agents perform well on (with a success rate of 50.79%). At the beginning of each task, we introduce disturbances to the windows by 1) changing the *position* of the window; 2) changing the *size* of the window to the minimal; 3) opening some irrelevant software and maximizing them to *clutter* the screen. We find that current agents are not robust to these superficial changes,

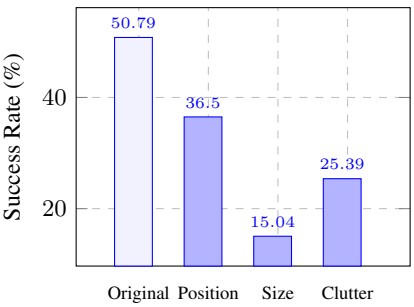

which lead to performance drops of 60% and even 80%. Surprisingly, agents are able to switch windows to a certain degree but fail to maximize the window as an intermediate step. This suggests that while agents possess some capability to navigate between windows, they lack a comprehensive strategy for managing window states effectively.

## 6 Related Work

**Benchmarks for multimodal agents** Recent research has made significant progress in evaluating multimodal agents, including physical world [49, 11, 8] and digital world [48, 15]. In the digital realm, interactive evaluation of agents primarily spans coding, web scenarios, and mobile applications. Prior work on coding provides frameworks and datasets for evaluating agents across programming languages and software engineering activities [62, 24, 28, 50]. Prior work on web agents develop platforms for interacting with websites through keyboard and mouse actions, as well as datasets focusing on open-ended web tasks and realistic web navigation [48, 34, 63, 12, 71, 26, 13]. Mobile device interaction research develop simulators for mobile UI interactions and platforms dedicated to InfoUI tasks [31, 52, 56, 55, 44, 66, 58, 65, 57]. Further, environments connecting to real computers and datasets for GUI grounding, albeit without interactive capability, have emerged [17, 10, 42, 25, 53]. Comprehensive task evaluation across different aspects also sees innovations [36, 40]. In contrast to prior work that address specific domains/applications/datasets, OSWORLD facilitates the development of general-purpose digital agents that openly interact with OS. See Tab. 4 for comparison.

**Vision-language models for multi-modal agents** Many existing works on GUI interaction utilize structured data such as HTML, accessibility trees, and view hierarchies as a grounding source [12, 19, 31, 41, 69, 51, 67, 71]. However, source code often tends to be verbose, non-intuitive, or inaccessible, which necessitates multi-modal/visual understanding. Prior work on multimodal models consider screenshots for interaction with websites [4, 16, 22, 27, 47] and mobile devices [21, 68]. Additionally, general-purpose foundation models [5, 30, 35, 72, 9] also demonstrate potential for multi-modal digital agents. Prompt-based reasoning methods [17, 20, 60, 70] have further improved digital agents for web pages, mobile apps, and desktops. This work evaluates state-of-the-art text, vision, and multi-modal methods, demonstrating that existing multi-modal models are far from capable computer agents.

## 7 Conclusion

OSWORLD addresses critical gaps in existing interactive learning environments to advance the development of autonomous digital agents. By providing a rich, realistic setting that spans multiple operating systems, interfaces, and applications, OSWORLD broadens the scope of tasks digital agents can perform, and enhances their potential for real-world applications. Despite the promise shown by advancements in vision-language models, evaluations within OSWORLD reveal notable challenges in agents' abilities, particularly in GUI understanding and operational knowledge, pointing to essential areas for future research and development.

## Acknowledgements

We thank Sida Wang, Peter Shaw, Alane Suhr, Luke Zettlemoyer, Chen Henry Wu, Pengcheng Yin, Shunyu Yao, Xing Han Lu, Siva Reddy, Ruoxi Sun, Zhiyuan Zeng, Chengyou Jia, Haoyuan Wu, Jiaqi Deng, Yuhao Yang and Lei Li for their helpful feedback on this work.

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

# A  Details of OSWORLD Environment

## A.1  Environment Infrastructure

As compared to core commonly used techniques like Docker [1], virtual machines can operate their own kernel and system, enabling compatibility with a wide variety of operating systems (such as Windows, macOS, Linux, etc.) across different CPU hardware types (x64, ARM, etc.), and supports training and evaluation in a multiprocess manner on both headless servers and personal computers.

## A.2  Observation Space

We implement three kinds of observation: **complete screenshot**, **accessibility tree** and **terminal output**. We also implement a video recorder of the environment but don't put it into our modeling due to the agent's ability limitations. OSWORLD supports observation refactoring and extending if needed, such as getting data from certain opening applications that we want to focus on.

### A.2.1  Screenshot

To align with the perception of a human user, we capture a screenshot of the entire computer screen. Including the mouse cursor also proves helpful in certain cases where mouse information is crucial. For screen resolution, we default to 1920×1080, as it is the most commonly used screen resolution according to Internet Users Screen Resolution Realtime Statistics for 2023[2]. This resolution also offers a 16:9 aspect ratio. OSWORLD also supports modifying the resolution of virtual machines to avoid potential memorization of absolute pixel values and to assist studies on topics like generalization across different resolutions.

### A.2.2  Accessibility Tree

An accessibility tree (or a11y tree, same logic to kubernetes and k8s), refers to an intricate structure generated by the browser or OS accessibility APIs that renders a representative model of the web content, providing a means of interaction for assistive technologies. Each node within the accessibility tree hosts important information about a UI element. This could range from the nature of the object (a button, checkbox, or paragraph of text), its current state (checked or unchecked, for checkboxes), and even its spatial orientation on the screen.

Different operating systems employ varied accessibility APIs and tools to construct and manipulate the accessibility tree. These include Microsoft Active Accessibility (MSAA) and User Interface Automation (UIA) for Windows, NSAccessibility Protocol and macOS Accessibility Inspector for macOS, and Assistive Technology Service Provider Interface (ATSPI) [3] for GNOME dekstop used on Ubuntu. We adopt `pyatspi` to get the accessibility tree on Ubuntu and `pywinauto` on Windows. We further convert it into XML format for message passing. Partial pieces of the XML formatted accessibility tree are shown in Figure 5.

Tools such as `Accerciser` can help visualize the corresponding relationship of tree nodes and GUI components in the accessibility tree as shown in Figure 6.

## A.3  Action Space

We implement two kinds of action space: `pyautogui` and `computer_13`. We mainly use `pyautogui` action space, since it saves tokens for describing action space definition in prompting, compared with self-designed actions.

### A.3.1  PYAUTOGUI

`pyautogui` is an open-source, cross-platform Python module utilized for programmatically controlling the mouse and keyboard. It can control simple movements, clicks, as well as keyboard inputs, and can provide screen capture features or locate where the screen-specific images reside that can be

---

[1] https://www.docker.com/
[2] https://www.screenresolution.org/year-2023/
[3] https://docs.gtk.org/atspi2/

This XML file does not appear to have any style information associated with it. The document tree is shown below.

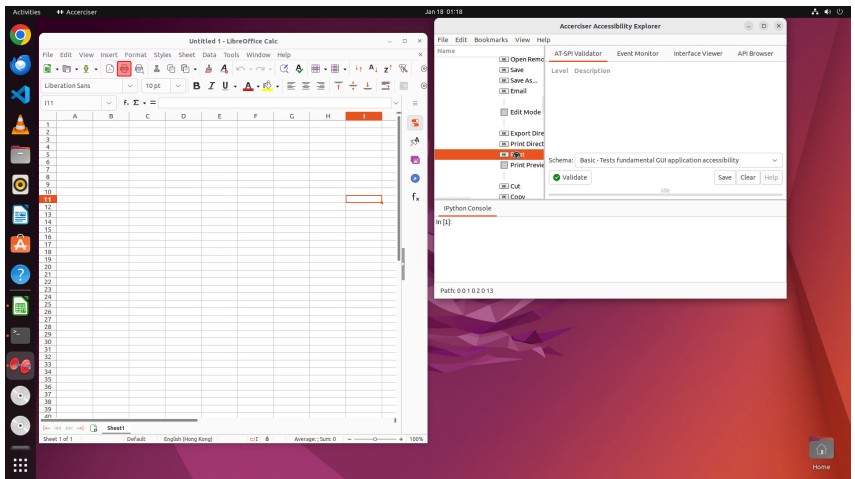

Figure 5: Raw XML formatted accessibility tree visualization.

Figure 6: Use `Accerciser` to visualize components in accessibility tree. This example shows the corresponding relationship between the "Print" button (which is a printer icon on the bar of the LibreOffice app as shown on the left) and the "Print" tree node (which is the "Print" of the Accerciser Accessibility Explorer as shown on the right).

useful for GUI automation. Compared with other Python packages such as xdotool [4], mouse [5] and keyboard [6], `pyautogui` has better compatibility across different OSes as found in our early attempts, so that we adapt that as our core component of computer controlling as well as an official valid action space.

Herein, we will demonstrate some use cases of `pyautogui` and illustrate how it can be wielded as an action space.

**Mouse Controlling Functions**

```
import pyautogui

# move mouse to XY coordinates over num_second seconds
pyautogui.moveTo(x, y, duration=num_seconds)

# drag mouse to XY
pyautogui.dragTo(x, y, duration=num_seconds)
```

---

[4]https://github.com/jordansissel/xdotool
[5]https://github.com/boppreh/mouse
[6]https://github.com/boppreh/keyboard

```
pyautogui.click(x=moveToX, y=moveToY, clicks=num_of_clicks, button='left')
pyautogui.rightClick(x=moveToX, y=moveToY)
pyautogui.middleClick(x=moveToX, y=moveToY)
pyautogui.doubleClick(x=moveToX, y=moveToY)
pyautogui.tripleClick(x=moveToX, y=moveToY)
pyautogui.scroll(amount_to_scroll, x=moveToX, y=moveToY)
pyautogui.mouseDown(x=moveToX, y=moveToY, button='left')
pyautogui.mouseUp(x=moveToX, y=moveToY, button='left')
```

**Keyboard Controlling Functions**

```
import pyautogui

# useful for entering text, newline is Enter
pyautogui.typewrite('Hello world!\n', interval=secs_between_keys)

pyautogui.typewrite(['a', 'b', 'c', 'left', 'backspace', 'enter', 'f1'], interval=secs_between_keys)
pyautogui.hotkey('ctrl', 'c')  # ctrl-c to copy
pyautogui.hotkey('ctrl', 'v')  # ctrl-v to paste
pyautogui.keyDown(key_name)
pyautogui.keyUp(key_name)
```

pyautogui **as an Action Space**    Given the various controls it provides, pyautogui can readily be used as an action space in building automation software or testing interfaces with minor adjustments. More formally, an action is within the action space when it meets the syntax of pyautogui or is one of three special actions WAIT, FAIL, and DONE. This might include actions like clicking at a certain location, entering text or key sequences, or even resting for a span (Pause). Each action could be mapped to an identifying label or number, forming a discrete action space. For example:

```
import pyautogui

def perform_action(action):
  if action == 0:
    pyautogui.moveTo(100, 100)
  elif action == 1:
    pyautogui.write('Hello world!', interval=0.25)
  else:
    pyautogui.pause(1)
```

In this scheme, the "perform_action" function constitutes the action space, where each unique action is associated with a unique integer (its action ID). The function interprets these action IDs and performs the corresponding action, forming a rudimentary discrete action space.

One interesting finding is that language models generate screenshot locate functions like:

```
pyautogui.locateOnScreen('Apple.png')
```

When there is insufficient grounding evidence (such as when no screenshot is inputted, the accessibility tree lacks a specific element, or the multimodal model cannot comprehend the user interface), employing this function to retrieve the correct icon image could present an interesting method.

### A.3.2   COMPUTER_13

To facilitate potential reinforcement learning applications, we have created a variant of pyautogui, which we named computer_13. In this variant, we wrap pyautogui into a finite action class with parameterized enumeration, such that it features 13 action types, excluding three special ones for task process control. Utilizing this structured approach allows more effective reinforcement learning by providing a distinct and finite set of actions to be learned and optimized. As summarized in Table 6, each action type has certain parameters, detailed in the collection, confirming the type, range, and whether each parameter is optional for that action.

Table 6: Action types and parameters defined in action space `computer_13`, a variance we created for the potential reinforcement learning research based on our environment.

| Action Type | Parameters | Note |
|---|---|---|
| MOVE_TO | *x, y* | *Move the cursor to the specified position* |
| CLICK | *button,* *x, y,* *num_clicks* | *Click the left button if the button not specified, otherwise click the specified button; click at the current position if x and y are not specified, otherwise click at the specified position* |
| MOUSE_DOWN | *button* | *Press the left button if the button not specified, otherwise press the specified button* |
| MOUSE_UP | *button* | *Release the left button if the button not specified, otherwise release the specified button* |
| RIGHT_CLICK | *x, y* | *Right click at the current position if x and y are not specified, otherwise right click at the specified position* |
| DOUBLE_CLICK | *x, y* | *Double click at the current position if x and y are not specified, otherwise double click at the specified position* |
| DRAG_TO | *x, y* | *Drag the cursor to the specified position with the left button pressed* |
| SCROLL | *dx, dy* | *Scroll the mouse wheel up or down* |
| TYPING | *text* | *Type the specified text* |
| PRESS | *key* | *Press the specified key and release it* |
| KEY_DOWN | *key* | *Press the specified key* |
| KEY_UP | *key* | *Release the specified key* |
| HOTKEY | *keys* | *Press the specified key combination* |
| WAIT | - | *Wait until the next action* |
| FAIL | - | *Decide the task cannot be performed* |
| DONE | - | *Decide the task is done* |

# B  Details of OSWORLD Benchmark

## B.1  Operating System Selection

As of 2023, the most popular desktop operating systems are Windows (69.5%), macOS (20.4%), ChromeOS (3.2%), and Linux (3.1%) [7]. While Windows and macOS dominate the market share and boast the richest software ecosystems, their closed-source nature raises potential copyright concerns for direct usage. ChromeOS, being a web-based operating system, heavily depends on a Google account for its functionalities, rendering it less suitable for a public benchmark.

In contrast, Linux desktops offer a wealth of open-source software for most daily tasks, supported by an active community for both basic and advanced use. Essential applications such as Office Suite, browsers, email clients, multimedia apps, and thousands of games and applications are either pre-installed or readily available through the software center of the distribution. Consequently, we select Ubuntu, the most representative Linux desktop OS, as the foundation for the main part of our benchmark intended for public use. Additionally, we have developed components to facilitate agent interaction on a Windows virtual machine and have created a relatively small set of examples focusing on the Microsoft Office suite, including Excel, PowerPoint, and Word. This serves as a counterpart to the LibreOffice suite available on Ubuntu. These components can be utilized in-house or officially with the purchase of a license. Regarding macOS, theoretically, it is illegal to install macOS on non-Apple devices, which leads us to refrain from developing our benchmark on this platform to avoid copyright issues.

## B.2  Software Selection

Due to the high cost of obtaining operation and evaluation script annotation data, we have chosen a representative set of software for the examples of Ubuntu part. We adopt standards that consider: 1) Availability - the software must be available on Ubuntu 22.04; 2) Open-source - the software should be open-sourced with an appropriate license to prevent copyright issues; 3) Popularity - the software should take a high download number and frequency of recommendations in blogs and tutorials; 4) Strong user community and good support resources - it is preferable to have an active and robust user community as well as official documents, which can serve as ample resources for task collection and agent learning; 5) Diversity of categories - the software should be diverse to adequately represent and cover a wide range of real-world cases.

---

[7]`https://www.statista.com/statistics/218089/global-market-share-of-windows-7/`

As a result, we have shortlisted the software into two categories: general usage and professional usage. For general usage, we have VLC for media playback, Chrome for web browsing, and Thunderbird for email management. For professional usage, we have VS Code as a coding IDE, and LibreOffice (Calc, Writer, and Impress) for handling spreadsheets, documents, and presentations respectively, along with GIMP for image editing. This brings our total to eight different types of software.

## B.3  Task Example Sources

We detail the task example sources in Table 7.

Table 7: Task Example Resources

| App | Resources | Link |
|---|---|---|
| OS | Ubuntu Documentations | https://help.ubuntu.com/ |
| | Ask Ubuntu | https://askubuntu.com/ |
| | Super User | https://superuser.com/ |
| | Stack Overflow | https://stackoverflow.com |
| | YouTube | https://www.youtube.com/ |
| Calc | LibreOffice Help | https://help.libreoffice.org/ |
| | Microsoft Tech Community | https://techcommunity.microsoft.com/ |
| | libreofficehelp.com | https://www.libreofficehelp.com/ |
| | Reddit r/LibreOfficeCal | https://www.reddit.com/r/LibreOfficeCalc/ |
| | Reddit r/Excel | https://www.reddit.com/r/Excel/ |
| | Super User | https://superuser.com/ |
| | Medium | https://medium.com/ |
| | Quora | https://www.quora.com/ |
| | YouTube | https://www.youtube.com/ |
| | Ryan and Debi & Toren Personal Site | https://www.ryananddebi.com/ |
| Writer | LibreOffice Help | https://help.libreoffice.org/ |
| | LibreOffice Forum | https://ask.libreoffice.org/ |
| | libreofficehelp.com | https://www.libreofficehelp.com/ |
| | Super User | https://superuser.com/ |
| | Stack Overflow | https://stackoverflow.com |
| | Ask Ubuntu | https://askubuntu.com/ |
| | Quora | https://www.quora.com/ |
| | YouTube | https://www.youtube.com/ |
| | SearchStar Personal Site | https://seekstar.github.io/ |
| Impress | LibreOffice Help | https://help.libreoffice.org/ |
| | libreofficehelp.com | https://www.libreofficehelp.com/ |
| | Reddit r/LibreOffice | https://www.reddit.com/r/LibreOffice/ |
| | Super User | https://superuser.com/ |
| | Stack Overflow | https://stackoverflow.com |
| | Technical Tips | https://technical-tips.com/ |
| | Just Click Here | https://justclickhere.co.uk/ |
| | TikTok | https://www.tiktok.com/ |
| VLC | VLC Documentation | https://docs.videolan.me |
| | VLCHelp.com | https://www.vlchelp.com/ |
| | VideoLAN's Wiki | https://wiki.videolan.org/ |
| | Ubuntu Documentations | https://help.ubuntu.com/ |
| | Reddit r/Fedora | https://www.reddit.com/r/Fedora/ |
| | Super User | https://superuser.com/ |
| | Medium | https://medium.com/ |
| | YouTube | https://www.youtube.com/ |
| | Dedoimedo | https://www.dedoimedo.com/index.html |
| Thunderbird | Thunderbird Support | https://support.mozilla.org/en-US/products/thunderbird |
| | Reddit r/Thunderbird | https://www.reddit.com/r/Thunderbird/ |
| | Reddit r/Automation | https://www.reddit.com/r/automation/ |
| | Super User | https://superuser.com/ |
| | WikiHow | https://www.wikihow.com/ |
| | Quora | https://www.quora.com/ |
| | BitRecover | https://www.bitrecover.com/ |
| | AdSigner | https://www.adsigner.com/ |
| Chrome | Google Chrome Help | https://support.google.com/chrome |

*Continued on next page*

| App | Resources | |
|---|---|---|
| | Reddit r/Chrome | `https://www.reddit.com/r/Chrome/` |
| | Super User | `https://superuser.com/` |
| | WikiHow | `https://www.wikihow.com/` |
| | in5steps.com | `https://in5stepstutorials.com/` |
| | How-To Geek | `https://www.howtogeek.com/` |
| | Medium | `https://medium.com/` |
| | Quora | `https://www.quora.com/` |
| | YouTube | `https://www.youtube.com/` |
| | Laptop Mag | `https://www.laptopmag.com` |
| VS Code | Super User | `https://superuser.com/` |
| | Stack Overflow | `https://stackoverflow.com` |
| | Quora | `https://www.quora.com/` |
| | YouTube | `https://www.youtube.com/` |
| | Campbell Muscle Lab GitHub | `https://campbell-muscle-lab.github.io/` |
| GIMP | Reddit r/GIMP | `https://www.reddit.com/r/GIMP/` |
| | Super User | `https://superuser.com/` |
| | Stack Overflow | `https://stackoverflow.com` |
| | Quora | `https://www.quora.com/` |
| | Make-Use-Of | `https://www.makeuseof.com/` |
| | YouTube | `https://www.youtube.com/` |
| Workflow | UniPath Marketplace | `https://marketplace.uipath.com/` |
| | sync.blue | `https://www.sync.blue/` |
| | Device Tests | `https://devicetests.com/` |
| | Make Tech Easier | `https://www.maketecheasier.com/` |
| | Unix & Linux Stack Exchange | `https://unix.stackexchange.com/` |
| | Geeks for Geeks | `https://www.geeksforgeeks.org/` |
| | I Love Free Software | `https://www.ilovefreesoftware.com/` |
| | The Geek Diary | `https://www.thegeekdiary.com/` |
| | Zyxware | `https://www.zyxware.com/` |
| | GNOME Discourse | `https://discourse.gnome.org/` |
| | It's FOSS | `https://itsfoss.com/` |
| | Super User | `https://superuser.com/` |
| | Stack Overflow | `https://stackoverflow.com` |
| | LibreOffice Forum | `https://ask.libreoffice.org/` |
| | ImpressExtractNotes | `https://github.com/danielrcollins1/ImpressExtractNotes` |
| | Medium | `https://medium.com/` |
| | YouTube | `https://www.youtube.com/` |
| | Kelvin Smith Library | `https://case.edu/library/` |

## B.4 Task Examples Collection

Here we show the detailed statistics of OSWORLD benchmark, including the main set on Ubuntu (369 in total) and the analytic set on Windows (43 in total).

Table 8: Detailed statistics of OSWORLD benchmark suite about examples number, average instruction tokens, infeasible instructions and integrated instructions.

| | OS | Calc | Impress | Writer | VLC | Thunderbird | Chrome | VSCode | GIMP | **Workflow** | **Overall** |
|---|---|---|---|---|---|---|---|---|---|---|---|
| Examples | 24 | 47 | 47 | 23 | 17 | 15 | 46 | 23 | 26 | 101 | 369 |
| Avg. Inst. Tokens | 22.38 | 33.30 | 25.19 | 35.30 | 35.82 | 34.07 | 22.07 | 20.78 | 16.23 | 51.24 | 33.36 |
| #Infeasible | 5 | 1 | 0 | 1 | 3 | 1 | 3 | 5 | 10 | 1 | 30 |
| #Integrated | 7 | 19 | 30 | 0 | 0 | 0 | 26 | 0 | 0 | 2 | 84 |

Table 9: Detailed statistics of Windows analytic set benchmark suite. This set contains no infeasible tasks and integrated tasks.

| | Excel | Word | PPT | **Workflow** | **Overall** |
|---|---|---|---|---|---|
| Examples | 11 | 9 | 7 | 16 | 43 |
| Avg. Inst. Tokens | 19.45 | 21.44 | 21.86 | 47.57 | 32.48 |

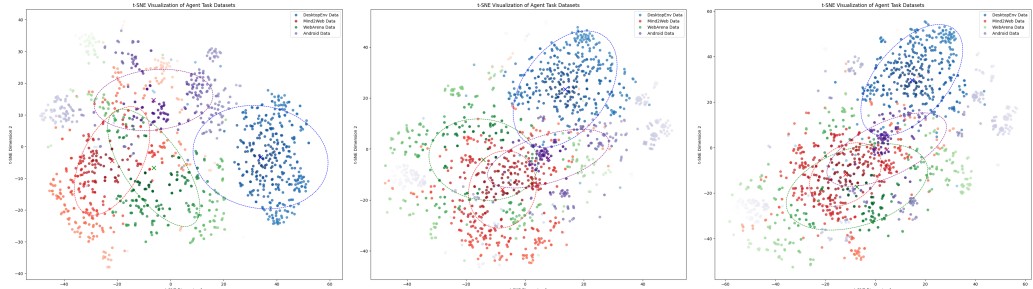

Figure 7: Comparison of instructions distribution. All datasets are sampled to 300 to make a fair comparison. The hyper-parameters of t-SNE are randomly sampled for each plot.

We also visualize the intent distribution (We obtain sentence embeddings for instructions using OpenAI's embedding model, and then apply t-SNE to reduce the dimensionality to two dimensions for visualization.) and compare it with other benchmarks which also focus on the digital agent. We randomly sample 300 examples from each dataset and randomly choose three different hyperparameters for t-SNE. Visualization results are shown in Figure 7. From the figure, we can observe that the semantic distribution of the instructions alone has reached the most comprehensive level. Additionally, our environment remains controllable and executable, offering a more reliable evaluation. It is also noticeable that the clustering centers of the other three are closely positioned, whereas the points in our distribution approaches are inconsistent with theirs, indicating that we can serve as a unique choice for a more comprehensive assessment of the capabilities of future intelligent agents.

## B.5  Initial State Setup Details

The setup of the initial state contains three stages: 1) Start emulator. The specified virtual machine is activated and automatically reverted to the corresponding snapshot, which records the initial system settings of the machine. 2) Prepare files (Optional). The file or software that specifies the initial state of the task to be executed is downloaded to the virtual machine and opened. The system is configured to first download the files to the host through a direct link and then upload them to the VM via a LAN connection. Specifically, some initial files are set up for OS-related tasks by manipulating the file system directly from the command line. 3) Execute reprocessing commands (Optional). For tasks that require additional preprocessing, task-specific operations are executed after the completion of the first two phases. For example, taking the currently open LibreOffice Impress file to page five, clicking in the center of the screen to return to the main interface, *etc*. We provide convenient APIs to configure initial conditions and world settings, standardizing our tasks to make this process user-friendly and easily extendable.

## B.6  Evaluation Configuration Details

In this section, we will show details of preparations for the evaluation of the selected apps (LibreOffice – Calc, Writer and Impress, Thunderbird, VLC Media Player, Chrome, VS Code, GIMP) and OS (Ubuntu and Windows).

### B.6.1  Ubuntu

**LibreOffice: Calc, Writer, and Impress**   LibreOffice is a popular open-source fully-featured office suite for Linux desktops. Our benchmark is built upon version 7.3.7.2, the version pre-installed in Ubuntu 22.04. Calc, Writer, and Impress are selected to build tasks on them. As the majority of tasks are to conduct a little revision to a file, we evaluate these tasks mainly by checking the final result file (in xlsx, docx, or pptx format). The check can be done by comparing the result file with a golden reference, or inspecting some particular attributes or settings of the file, *e.g.,*, page style, freezing, and locale. Usually, the xlsx, docx, and pptx files are mainly accessed through `openpyxl`[8], `python-docx`[9], and `python-pptx`[10]. For some properties not supported by the current libraries, we also look them up directly via parsing the Office Open XML format[11].

**Thunderbird**   Thunderbird is a popular open-source fully-featured email client for Linux desktops. Version 115.6.0 of Thunderbird is pre-installed in Ubuntu 22.04. We crafted an account profile to set up a feasible initial state. Evaluation for Thunderbird configurations is mainly performed by reading various configurations or data

---

[8]`https://openpyxl.readthedocs.io/en/stable/`
[9]`https://github.com/python-openxml/python-docx`
[10]`https://github.com/scanny/python-pptx`
[11]`https://learn.microsoft.com/en-us/office/open-xml/about-the-open-xml-sdk`

files in the profile folder. An open-source reverse engineering tool Firefox Decrypt[12] is adopted to decrypt the stored account information for evaluation. The involved account information is just for examples and contains no information about the real person. Besides, there are tasks instructing to help to compose a new email. In these cases, the accessibility tree is leveraged to inspect the contents in the composing window before really sending it.

**VLC Media Player**    VLC Media Player is a popular open-source cross-platform multimedia player and framework that plays most multimedia files. The evaluation for VLC Media Player is multifold, ranging from utilizing VLC HTTP interface[13], reading the VLC configuration file, comparing final result files, and leveraging accessibility tree to inspect the desired content.

**Chrome**    Google Chrome is one of the most popular and powerful cross-platform web browsers developed on Google's free and open-source software project Chromium. The evaluation of Chrome is mainly based on the utilization of Playwright[14], a browser automation library to control Chromium, Firefox, and WebKit with a single API. To connect Playwright running on host machine with Chrome running on virtual machine, port transferring tool socat[15] is leveraged. Additional information such as the HTML source codes of websites is also leveraged in the evaluation of some tasks.

**VS Code**    VS Code is a popular open-source multi-functional cross-platform editor for source-code editing. The evaluation of VS Code tasks is primarily divided into two different categories. One subset of tasks is predominantly oriented towards file manipulation. In the context of these tasks, a comparative analysis is conducted between the resultant file and an anticipated reference gold file. Another subset of tasks is centered around how to utilize the intrinsic functionalities of the VS Code software itself, such as modifying color themes, initiating workspace sessions, and modifying settings. In these instances, it becomes important to extract relevant internal information and configurations from the VS Code environment.

In the course of this research, we principally leverage the capabilities offered by the VS Code Extension API[16] and information in the settings JSON file[17] to obtain the requisite internal signal for the evaluation process. Our methodology involves the development of a custom VS Code extension, its installation within the VS Code software deployed on our virtual machine, and the subsequent invocation of the extension's command each time an evaluation is required, as well as checking whether the settings JSON has the correct value for a specific key.

**GIMP**    GIMP is an open-source raster graphics editor used for image manipulation, editing, free-form drawing, format transcoding, and more specialized tasks. The evaluation for GIMP tasks is also mainly divided into two different categories, just like the VS Code evaluation. One type of task is mainly oriented to file operations. In these tasks, the resulting files are compared and analyzed with the expected reference golden files, mainly relying on some powerful image processing libraries such as `pillow`[18]. Another category of tasks revolves around taking advantage of the inherent capabilities of the GIMP software itself. In these instances, we primarily read GIMP's configuration files to obtain internal information to evaluate the tasks.

### B.6.2   Windows

**Microsoft Office: Excel, Word, and PowerPoint**    Microsoft Office is the most popular office suite on Windows desktops. These three components share the same functions with the corresponding LibreOffice components by and large. They are used to edit xlsx, docx, and pptx files, respectively. Thus, the evaluation for LibreOffice tasks can be reused for Microsoft Office tasks.

**Thunderbird**    Thunderbird is a cross-platform email client. Only the structure of profile folder on Windows is sightly different from that on Linux. We thus revised the account profile and reuse it to set up the same initial state on Windows.

**Chrome**    Chrome is a cross-platform web browser. To evaluate tasks on Chrome, only the port transferring tool needs to be replaced with Ncat[19]. Other configurations and the evaluations can be shared with Linux-version tasks.

---

[12]`https://github.com/unode/firefox_decrypt`

[13]`https://wiki.videolan.org/Control_VLC_via_a_browser/`

[14]`https://playwright.dev/`

[15]`http://www.dest-unreach.org/socat/,https://linux.die.net/man/1/socat`

[16]`https://code.visualstudio.com/api`

[17]`https://code.visualstudio.com/docs/getstarted/settings#_settingsjson`

[18]`https://pypi.org/project/pillow/`

[19]`http://www.dest-unreach.org/socat/`

## B.7 More Task Examples

In this section, we curate a collection of examples from various app sets, each characterized by distinct operational logic and requiring different capabilities. These examples are carefully chosen to illustrate the diverse challenges and requirements encountered when interacting with different types of applications.

Table 10: More Example Showcase from Each Subset of Domains.

| Related App(s) | Instruction(s) | Screenshot | Abilities Needed |
|---|---|---|---|
| OS | *I want to install Spotify on my current system. Could you please help me?* |  | `knowledge of OS; omit distractions` |
| Calc | *Check the names in column "Names with duplicates" and put the unique ones in column "Unique Names". Keep the original order.* |  | `massive elements; knowledge tricks or reasoning over long actions` |
| Calc | *I have a lookup table for the officers of each branch. Please, here is another table in which I need to fill with the officer names according the headoffice (i.e., the branch name). Help me to complete this.* |  | `massive elements; knowledge of formulas and functions` |
| Calc | *Given a partial calendar, please highlight all the weekends (Saturday & Sunday) by setting the cell background as red (#ff0000).* |  | `massive elements; commonsense reasoning; software tricks` |
| Impress | *I closed the slide pannel on the left and idk how to get it back please help* |  | `software knowledge; imagine about UI layouts; overcome typos in instruction` |

*Continued on next page*

| Related App(s) | Task Instruction | Screenshot of Initial State | Abilities Needed |
|---|---|---|---|
| Impress | *On it Whenever I launch a LibreOffice Impress, it uses both screens, one for current slide and next slide and another for actual presentation. What I want is to use only one monitor which shows presentation. I dont want the screen with Current slide and Next slide so that it can be used for other purposes. How should I achieve this?* | | `reason from unprofessional phenomenon expression` |
| Writer | *Copy the screenshot 1.png from the desktop to where my cursor is located* | | `locate the position of cursor; switch from desktop and app` |
| Chrome | *Can you help me clean up my computer by getting rid of all the tracking things that Amazon might have saved? I want to make sure my browsing is private and those sites don't remember me.* | | `understanding the unprofessional expression` |
| VLC | *I am reading lecture note in PDF while a music video is running in VLC media player. But I find I need to switch to the player every time I need to pause/start.Could you help me change the setting to allow pausing the video using keyboard shortcut without minimizing the PDF reader? I want to focus on the lecture note and don't be disturbed by the app switching.* | | `understanding the reference from unprofessional expression; software knowledge` |
| VLC | *Hey, could you turn this video the right way up for me? And once it's flipped around, could you save it for me with the name '1984_Apple.mp4' on the main screen where all my files are?* | | `software knowledge; spatial judgment ability` |

*Continued on next page*

| Related App(s) | Task Instruction | Screenshot of Initial State | Abilities Needed |
|---|---|---|---|
| Thunderbird | *Create a local folder called "Promotions" and create a filter to auto move the inbox emails whose subject contains "discount" to the new folder* |  | `software knowledge` |
| Thunderbird | *Attach the my AWS bill to the email. The bill is stored at /aws-bill.pdf. Don't close it or send it. I haven't finish all the contents.* |  | `file management; extra requirement` |
| Thunderbird | *I've got a bunch of email accounts in Thunderbird, and it's a hassle to check them one by one. Can you show me how to set up a unified inbox so I can see all my emails in one place?* |  | `deep-hided feature, need to be explored even by human users; pop-up window` |
| VS Code | *Please modify VS Code's settings to disable error reporting for Python missing imports.* |  | `software knowledge to deal with settings; reasoning to understand the cause and solution of the error` |
| VS Code | *Please help me install the autoDocstring extension in VS Code.* |  | `software knowledge to deal with Extensions; reasoning to search and install the extension successfully` |
| GIMP | *Could you make the background of this image transparent for me?* |  | `precise and intricate operations` |

| Related App(s) | Task Instruction | Screenshot of Initial State | Abilities Needed |
|---|---|---|---|
| GIMP | *Help me choose the yellow triangle and position it at the center of my picture.* |  | spatial perception and reasoning, as well as precise control of actions |
| Multiple (VLC+GIMP) | *Could you help me create an Animated GIF from a video file using VLC and GIMP from the source of video "src.mp4", 5-second clip beginning at 00:03?* |  | software knowledge to undergo sophisticated processes and ability to process multi-step procedure successfully |
| Multiple (ThunderBird+ Writer+Chrome) | *Help me export charts, graph or other images from docx files received in email "Lecture Document" in Notes folder and upload these png files to the figures/ folder in Google Drive for later use (use numbers to name them).* |  | ability to selectively export charts, graphs and images from docx file; software knowledge for google drive file upload |
| Multiple (Chrome+Calc) | *Could you help me extract data in the table from a new invoice uploaded to my Google Drive, then export it to a Libreoffice calc .xlsx file in the desktop?* |  | ability to do table data extraction; export data to .xlsx file |

# C   Details of Baseline Methods

## C.1   Hyper-Parameter of the Baseline Agents

We utilize the versions of `gpt-3.5-turbo-16k`, `gpt-4-0125-preview`, and `gpt-4-vision-preview`, respectively for GPT results, need to be noted that result could be changed from time since it is close-sourced. We also employ the `gemini-pro` and `gemini-pro-vision` versions for the Gemini models For all language models, we set the temperature parameter to 1.0, and top_p to 0.9, and the maximum number of tokens for generation is set to 1500. We set the maximum steps of interaction to 15 and the maximum time limits to 30 minutes for all tasks since the agent could lead to a stuck environment under some unexpected cases.

## C.2 Prompt Details

### C.2.1 Prompt for A11y Tree, Screenshot and their Combination Setting

```
You are an agent which follow my instruction and perform desktop computer
↪   tasks as instructed.
You have good knowledge of computer and good internet connection and assume
↪   your code will run on a computer for controlling the mouse and keyboard.
For each step, you will get an observation of an image, which is the
↪   screenshot of the computer screen and you will predict the action of the
↪   computer based on the image.

You are required to use `pyautogui` to perform the action grounded to the
↪   observation, but DONOT use the `pyautogui.locateCenterOnScreen` function
↪   to locate the element you want to operate with since we have no image of
↪   the element you want to operate with. DONOT USE `pyautogui.screenshot()`
↪   to make screenshot.
Return one line or multiple lines of python code to perform the action each
↪   time, be time efficient. When predicting multiple lines of code, make
↪   some small sleep like `time.sleep(0.5);` interval so that the machine
↪   could take; Each time you need to predict a complete code, no variables
↪   or function can be shared from history
You need to to specify the coordinates of by yourself based on your
↪   observation of current observation, but you should be careful to ensure
↪   that the coordinates are correct.
You ONLY need to return the code inside a code block, like this:
```python
# your code here
```

Specially, it is also allowed to return the following special code:
When you think you have to wait for some time, return ```WAIT```;
When you think the task can not be done, return ```FAIL```, don't easily say
↪   ```FAIL```, try your best to do the task;
When you think the task is done, return ```DONE```.

My computer's password is 'password', feel free to use it when you need sudo
↪   rights.
First give the current screenshot and previous things we did a short
↪   reflection, then RETURN ME THE CODE OR SPECIAL CODE I ASKED FOR. NEVER
↪   EVER RETURN ME ANYTHING ELSE.
```

For a11y tree setting and a11y tree + screenshot setting, the prompts are basically the same, just replace the screenshot words with a11y tree words.

### C.2.2 Prompt for SoM Setting

```
You are an agent which follow my instruction and perform desktop computer
↪   tasks as instructed.
You have good knowledge of computer and good internet connection and assume
↪   your code will run on a computer for controlling the mouse and keyboard.
For each step, you will get an observation of the desktop by 1) a screenshot
↪   with interact-able elements marked with numerical tags; and 2)
↪   accessibility tree, which is based on AT-SPI library. And you will
↪   predict the action of the computer based on the image and text
↪   information.

You are required to use `pyautogui` to perform the action grounded to the
↪   observation, but DONOT use the `pyautogui.locateCenterOnScreen` function
↪   to locate the element you want to operate with since we have no image of
↪   the element you want to operate with. DONOT USE `pyautogui.screenshot()`
↪   to make screenshot.
```

```
You can replace x, y in the code with the tag of the element you want to
↪   operate with. such as:
```python
pyautogui.moveTo(tag_3)
pyautogui.click(tag_2)
pyautogui.dragTo(tag_1, button='left')
```

When you think you can directly output precise x and y coordinates or there
↪   is no tag on which you want to interact, you can also use them directly.
But you should be careful to ensure that the coordinates are correct.
Return one line or multiple lines of python code to perform the action each
↪   time, be time efficient. When predicting multiple lines of code, make
↪   some small sleep like `time.sleep(0.5);` interval so that the machine
↪   could take; Each time you need to predict a complete code, no variables
↪   or function can be shared from history
You need to to specify the coordinates of by yourself based on your
↪   observation of current observation, but you should be careful to ensure
↪   that the coordinates are correct.
You ONLY need to return the code inside a code block, like this:
```python
# your code here
```

Specially, it is also allowed to return the following special code:
When you think you have to wait for some time, return ```WAIT```;
When you think the task can not be done, return ```FAIL```, don't easily say
↪   ```FAIL```, try your best to do the task;
When you think the task is done, return ```DONE```.

My computer's password is 'password', feel free to use it when you need sudo
↪   rights.
First give the current screenshot and previous things we did a short
↪   reflection, then RETURN ME THE CODE OR SPECIAL CODE I ASKED FOR. NEVER
↪   EVER RETURN ME ANYTHING ELSE.
```

## C.3   Input Settings

**Accessibility tree**   We evaluate reasoning capabilities of LLM agents on textual descriptions of the observation in the form of accessibility trees. To make accessibility trees (usually millions of tokens) tractable for LLM contexts, we filter out non-essential elements by their tag, visibility, availability, *etc* as described in  C.4. Only the *tag*, *name*, *text*, *position*, and *size* of the remaining elements are kept and concatenated by tab character in the input. As the raw coordinates are provided within the accessibility tree, the LLM is required to ground its action predictions to accurate coordinates.

**Screenshot**   To evaluate the visual reasoning ability of VLMs, we give the raw screenshot of the VM directly to the VLM, which is understand the screenshot and predict correct actions with precise coordinates. The raw resolution of the screen is set to $1920 \times 1080$. To investigate robustness to resolution changes, we perform ablation studies by manually downsampling the screenshot.

**Screenshot + accessibility tree**   We test whether the combination of screenshots with the accessibility tree can improve the capacity of VLM for spatial grounding.

**Set-of-Marks**   Set-of-Marks (SoM) [61] enhances the grounding capabilities of VLMs marking image segments with annotations like alphanumerics, masks, or boxes. We leverage the information from the filtered accessibility tree and mark the elements on the screenshot with a numbered bounding box. Following VisualWebArena [26] and UFO [64], we further combine the annotated screenshot with the text metadata from accessibility tree, including the *index*, *tag*, *name*, and *text* of the elements. Instead of predicting precise coordinates, the VLM is supposed to specify the action object by its number index, which will be mapped into our action space by post-processing. Ablation studies are also conducted with different resolutions for SoM setting.

## C.4 Accessibility Tree Filtering

Since the original tree is large (usually over 1 million tokens in XML format), we filter the accessibility tree nodes by their tags, visibilities, availabilities, *etc*. The concrete rules are illustrated in the following Table 11.

Table 11: Criteria for keeping a11y tree nodes on Ubuntu and Windows platforms

| Condition | Ubuntu | Windows |
|---|---|---|
| Node Tags | document∗, ∗item, ∗button, ∗heading, ∗label, ∗scrollbar, ∗searchbox, ∗textbox, ∗link, ∗tabelement, ∗textfield, ∗textarea, ∗menu, alert, canvas, checkbox, combo-box, entry, icon, image, paragraph, scroll-bar, section, slider, static, table-cell, terminal, text, netuiribbontab, start, trayclockwclass, traydummysearchcontrol, uiimage, uiproperty, uiribboncommandbar | Same as Ubuntu |
| Showing | True | Not Applicable |
| Visible | True | True |
| Enabled or Editable or Expandable or Checkable | True | True |
| Has Name or Text or Image | True | True |
| Position | >= 0 | >= 0 |
| Size | > 0 | > 0 |

## C.5 Set-of-Mark Implementation Details

Our methodology involves an initial transformation of the original observational image acquired from our environment into the standardized *SoM* format and be putted into VLMs together with a table of the marks with metadata information such as tags and names. This format consists of bounding boxes that bound the sub-images of interest, each associated with a corresponding integer mark. Notably, our approach incorporates the utilization of the accessibility tree to identify the bounding boxes associated with all clickable buttons within the current image observation, instead of using segmentation models like the original *SoM* paper. Some examples of screenshots after applying SoM are shown in Figure 8. We can observe the emergence of some low-quality, unmodelable tasks, and even misleading bounding boxes, depending on the level of support from the software ecosystem. This could be another reason for the poor performance of SoM. Future improvements could be made in this regard.

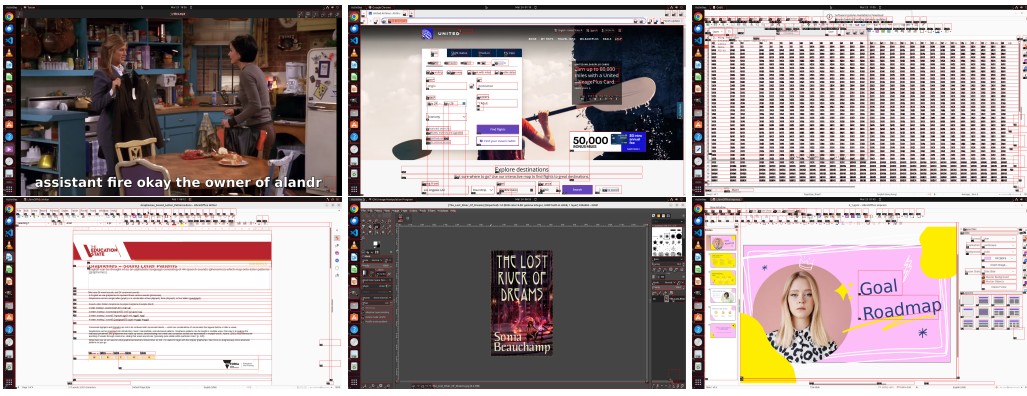

Figure 8: Showcase of example screenshots marked by SoM across different app GUI.

Table 12: Success rates of more baseline LLM and VLM agents on OSWORLD, grouped by task categories: OS, Office (LibreOffice Calc, Impress, Writer), Daily (Chrome, VLC Player, Thunderbird), Professional (VS Code and GIMP) and Workflow (tasks involving multiple apps), for gaining insights from interfaces and operation logic. See C.1 and C.6 for more details.

| Inputs | Model | Success Rate (↑) | | | | | |
|---|---|---|---|---|---|---|---|
| | | OS | Office | Daily | Profess. | **Workflow** | **Overall** |
| Screenshot | GPT-4o-mini | 12.50% | 3.58% | 3.99% | 4.08% | 1.62% | 3.77% |
| | InternVL2 | 12.50% | 1.87% | 2.71% | 8.16% | 0.99% | 3.33% |
| | MiniCPM-V-2.6 | 8.33% | 2.72% | 1.42% | 0.00% | 0.63% | 1.88% |
| | Llava-OneVision | 8.33% | 2.72% | 2.71% | 0.00% | 1.62% | 2.42% |
| Human Performance | | 75.00% | 71.79% | 70.51% | 73.47% | 73.27% | 72.36% |

## C.6 Full Results of Baseline Methods

Here we show the break-down results of baseline methods from different LLMs and VLMs for follow-up reference.

We have also compiled the distribution of steps taken by the GPT-4V model under our four settings: Accessibility Tree (A11y Tree), Screenshot, Screenshot combined with Accessibility Tree (Screenshot+A11y Tree), and Set-of-Mark. This data (as shown in Fig. 9) provides potential clues for future work. Overall, there are observable differences in how many steps the agent chooses to execute and when it decides to terminate under different settings. More detailed control and analysis of these behaviors can be explored in subsequent research efforts.

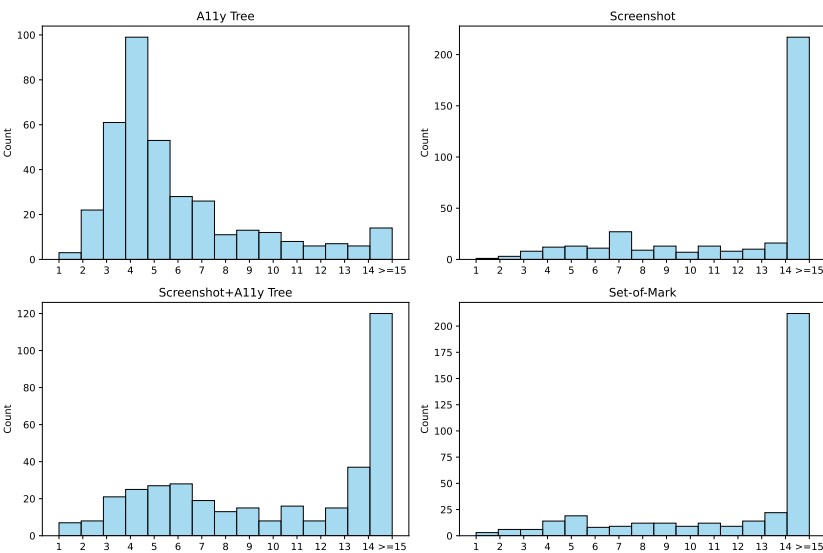

Figure 9: Distribution of steps taken by the GPT-4V based agents across four different settings.

## C.7 Running Time and Cost

We calculate and monitor the running time and cost for a portion of our experiments, which use the APIs of GPT-4V, Gemini-ProV, and Claude-3 Opus under different settings in April 2024. These results are shown in Table 14.

# D Further Analysis

## D.1 Performance by Task Difficulty, Feasibility and App Involved

We analyze the success rate across several additional subsets of tasks, as summarized in Tab. 15 and will be discussed in the following sections.

Table 13: Detailed success rates of baseline LLM and VLM agents on OSWORLD, divided by apps (domains): OS, LibreOffice Calc, LibreOffice Impress, LibreOffice Writer, Chrome, VLC Player, Thunderbird, VS Code, GIMP and Workflow which is comprehensive with multiple apps, for gaining insights from interfaces and operation logics.

| Inputs | Model | Success Rate (↑) | | | | | | | | | |
|--------|-------|------|------|---------|--------|------|------|--------|------|------|----------|
| | | OS | Calc | Impress | Writer | VLC | TB | Chrome | VSC | GIMP | Workflow |
| A11y | Mixtral-8x7B | 12.50 | 0.00 | 0.39 | 4.34 | 10.22 | 6.67 | 2.17 | 8.69 | 3.85 | 0.10 |
| | GPT-3.5 | 4.17 | 2.13 | 6.77 | 4.35 | 6.53 | 0.00 | 2.17 | 0.00 | 0.00 | 1.62 |
| | Gemini-Pro | 4.17 | 0.00 | 2.13 | 4.35 | 12.41 | 0.00 | 2.17 | 0.00 | 7.69 | 0.63 |
| | GPT-4 | 20.83 | 0.00 | 6.77 | 4.35 | 23.53 | 26.67 | 26.09 | 30.43 | 23.08 | 2.97 |
| | Gemini-Pro-1.5 | 12.50 | 2.13 | 2.13 | 4.35 | 6.53 | 0.00 | 10.87 | 8.70 | 0.00 | 3.60 |
| | Llama-3-70B | 4.17 | 0.00 | 0.39 | 8.70 | 6.53 | 0.00 | 2.17 | 0.00 | 0.00 | 0.63 |
| | GPT-4o | 20.83 | 6.38 | 6.77 | 8.69 | 12.41 | 20.00 | 17.39 | 21.74 | 11.54 | 7.56 |
| | Qwen-Max | 29.17 | 0.00 | 2.52 | 13.04 | 8.95 | 0.00 | 10.87 | 8.70 | 11.54 | 2.61 |
| Screen | CogAgent | 4.17 | 0.00 | 0.00 | 4.34 | 6.53 | 0.00 | 2.17 | 0.00 | 0.00 | 0.00 |
| | Gemini-ProV | 8.33 | 0.00 | 6.77 | 4.35 | 12.41 | 0.00 | 6.52 | 8.70 | 23.08 | 2.08 |
| | GPT-4V | 12.50 | 0.00 | 2.52 | 4.35 | 18.34 | 0.00 | 6.52 | 0.00 | 7.69 | 6.04 |
| | Claude-3-Opus | 4.17 | 0.00 | 2.52 | 4.34 | 6.53 | 0.00 | 2.17 | 0.00 | 3.84 | 2.61 |
| | Gemini-Pro-1.5 | 12.50 | 0.00 | 13.16 | 8.70 | 6.53 | 0.00 | 2.17 | 0.00 | 11.54 | 3.60 |
| | GPT-4o | 8.33 | 0.00 | 6.77 | 4.35 | 16.10 | 0.00 | 4.35 | 4.35 | 3.85 | 5.58 |
| Screen + A11y | CogAgent | 4.17 | 2.17 | 0.00 | 4.35 | 6.53 | 0.00 | 2.17 | 0.00 | 0.00 | 0.10 |
| | Gemini-ProV | 4.17 | 2.13 | 6.77 | 4.35 | 18.30 | 0.00 | 4.35 | 0.00 | 0.00 | 1.52 |
| | GPT-4V | 16.67 | 0.00 | 6.77 | 21.73 | 24.18 | 33.33 | 21.74 | 21.74 | 15.38 | 4.59 |
| | Claude-3-Opus | 12.50 | 2.13 | 4.65 | 4.34 | 18.30 | 0.00 | 2.17 | 8.69 | 7.69 | 0.99 |
| | Gemini-Pro-1.5 | 12.50 | 0.00 | 4.65 | 8.70 | 12.41 | 0.00 | 8.70 | 4.35 | 11.54 | 1.56 |
| | GPT-4o | 41.67 | 4.26 | 6.81 | 8.70 | 9.50 | 6.67 | 15.22 | 30.43 | 0.00 | 7.46 |
| SoM | CogAgent | 4.17 | 2.17 | 0.00 | 4.34 | 6.53 | 0.00 | 2.17 | 0.00 | 0.00 | 0.00 |
| | Gemini-ProV | 4.17 | 0.00 | 0.39 | 4.34 | 6.53 | 0.00 | 0.00 | 0.00 | 0.00 | 0.63 |
| | GPT-4V | 8.33 | 8.51 | 6.38 | 13.04 | 10.66 | 33.33 | 23.91 | 17.39 | 11.54 | 6.57 |
| | Claude-3-Opus | 12.50 | 2.13 | 0.39 | 8.70 | 6.53 | 13.33 | 17.39 | 0.00 | 11.54 | 4.49 |
| | Gemini-Pro-1.5 | 16.67 | 0.00 | 10.64 | 4.35 | 24.18 | 13.33 | 8.70 | 4.35 | 15.38 | 3.60 |
| | GPT-4o | 20.83 | 0.00 | 6.77 | 4.35 | 6.53 | 0.00 | 4.35 | 4.35 | 0.00 | 3.60 |
| Human Performance | | 75.00 | 61.70 | 80.85 | 73.91 | 70.59 | 46.67 | 78.26 | 73.91 | 73.08 | 73.27 |

Table 14: Summary of expected time and budget cost for different settings. Calculated in April 2024; these values may change with updates from the API providers.

| Setting | Expected Time | Budget Cost (Full Test Set/Small Test Set) |
|---------|---------------|--------------------------------------------|
| GPT-4V (screenshot) | 10h | $100 ($10) |
| Gemini-ProV (screenshot) | 15h | $0 ($0) |
| Claude-3 Opus (screenshot) | 15h | $150 ($15) |
| GPT-4V (a11y tree, SoM, etc.) | 30h | $500 ($50) |

**Task difficulty** We categorize the tasks based on the time required for human completion into three groups: 0∼60s (Easy), 60s∼180s (Medium), and greater than 180 seconds (Hard), as an indicator of difficulty. Across these groups, the model's success rate drops as the required time increases, with tasks taking longer than 180 seconds becoming almost impossible to complete (considering we have infeasible examples for agent's luckiness), whereas human performance across these three groups is 84.91%, 81.08% and 49.57%, showing a slight decline of the same trend but not to the extent of being unachievable.

Table 15: Success rate (SR) of GPT-4V (SoM) across different types of tasks.

| Task Subset | % of Total | SR (↑) |
|-------------|------------|--------|
| Easy | 28.72% | **16.78%** |
| Medium | 40.11% | 13.12% |
| Hard | 30.17% | 4.59% |
| Infeasible | 8.13% | **16.67%** |
| Feasible | 91.87% | 13.34% |
| Single-App | 72.63% | **13.74%** |
| Multi-App Workflow | 27.37% | 6.57% |

**Feasibility**    We also divide tasks into groups of tasks in-
feasible (*e.g.*, deprecated features or hallucinated features)
and tasks feasible, which requires the agents to have the ability to judge based on their own knowledge and
exploration results. As shown in Tab. 15, we observe that agents currently perform slightly better in terms of
infeasibility (16.67% to 13.34%), but overall, they are at a relatively low level. It is noteworthy that we also
observe in some methods and settings (such as under the pure screenshot setting with the Gemini-Pro model),
agents tend to easily output FAIL and refuse to continue trying. This situation leads to some false positives in
infeasible tasks. The focus needs to be on improving overall performance.

**Number of apps involved**    We also examined the performance based on whether the task involved apps
software or within a single app. As shown in Tab. 15, the average performance for tasks involving a single app is
low, at 13.74%, but still more than double the 6.57% observed for subsets of tasks involving workflows across
multiple apps. Within single-app scenarios, tasks involving GUI-intensive Office apps generally performed the
worst, with subsets such as LibreOffice Calc often scoring zero (we show more detailed results in App. C.6).
These findings highlight the need for improved collaboration capabilities between software and enhanced
proficiency in specific scenarios.

## D.2    A11y tree-based Observation Length Distribution

The main experiment revealed the decisive role of the a11y tree in performance within the current technological
context. Even when we retain key attribute elements based on heuristic rules (keep nodes with tags of the
document, item, button, heading, label, *etc.*), LLMs still require a sufficiently large context to process this
information effectively. To further understand this, we sample some a11y tree observations from OSWORLD
and conducted the statistical analysis, as shown in Figure 10. The analysis indicates that a context length of
6000 is needed to accommodate about 90% of cases for a single observation.

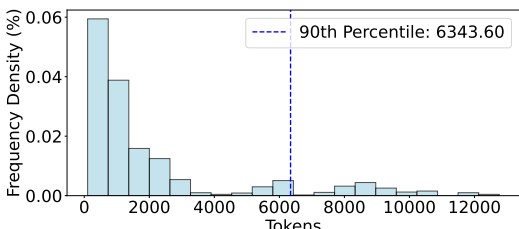

Figure 10: The length distribution of a11y tree as observation from sampled trajectories.

## D.3    Performance across Different Operating Systems

Another key challenge in building universal digital agents
is ensuring that these agents can maintain efficient and con-
sistent performance across different operating system envi-
ronments. The differences between OS and their software
ecosystems can significantly impact an agent's observation
and action spaces, leading to performance uncertainties. Here,
we explore and analyze the correlation between the success
of agents in completing tasks on Windows after migrating
from Ubuntu using examples from OSWORLD.

Table 16: Comparison of model perfor-
mance and correlation across operating sys-
tems.

| OS | SR (%) | Correlation Coefficient |
|---|---|---|
| Ubuntu | 4.88 | 0.7 |
| Windows | 2.55 | |

We enhance the functionality of the OSWORLD environment to support setting up initial experiment states, final
evaluations, and obtaining observations such as the a11y tree and screenshots in Windows OS. Additionally,
we have made example-wise fine-tuning modifications to the existing subset in OSWORLD for migration to
Windows. We conduct evaluations using the GPT-4V screenshot-only method and present the correlation of
performance across the two operating systems. As shown in Tab. 16, the model's performance on Ubuntu and
Windows is 4.88% and 2.55%, respectively, with a correlation coefficient of 0.7, despite the differences in their
observation spaces. This implies that insights and methodologies developed within the OSWORLD framework
can be effectively transferred to Windows environments with a high degree of reliability.

## D.4    Qualitative Analysis

In this section we highlight representative examples of success, failure, and surprising outcomes, alongside a
comparative study between GPT-4V and Claude-3 agents, to elucidate the unique challenges and insights our
environment introduces. See App. D.5 for more details.

**Success and failure cases**  We find agents, particularly based on GPT-4V, can successfully solve tasks that involve complex problem-solving or creative thinking, showcasing the advanced understanding and processing capabilities of the model already. One successful task is shown in the first row of Figure 11. The agent is requested to extract subtitle files from the video stream and save them locally. The agent first divides the screen into two parts, with the VLC application window on the left and the terminal window open on the right, and uses the ffmpeg command twice. The first use removes the subtitles embedded in the original video, and the second use saves the extracted subtitles locally.

Task Instruction: I downloaded an episode of Friends to practice listening, but I don't know how to remove the subtitles. Please help me remove the subtitles from the video and export it as "subtitles.srt" and store it in the same directory as the video.

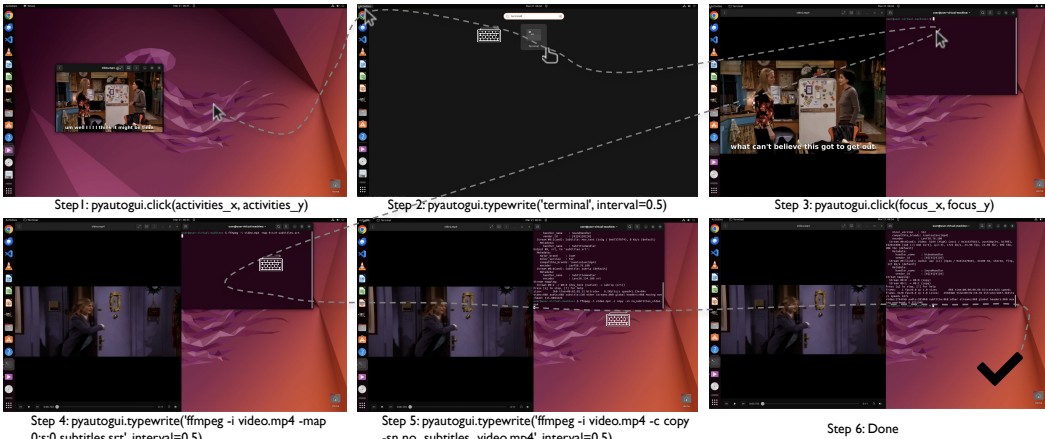

Figure 11: The agent successfully understood the complex task instructions, extracted the subtitle file from the video, and generated a pure video without embedded subtitles.

Despite the successes, there are notable failures that highlight the limitations of current models. In the task of "center-aligning the title of the document" (Fig. 12 line 1), the agent fails to ground the relatively simple requirement of "center alignment of texts", performing many useless actions such as selecting irrelevant words, opening irrelevant menus, *etc*.

Moreover, we find that the agent lacks prior knowledge in using software, performing poorly in many specialized tasks (as shown in Fig. 13, with GIMP, LibreOffice Calc, and Chrome selected). Taking GIMP as an example, for the instruction "reduce brightness" the agent does not know which menu in the toolbar is for brightness adjustment and instead randomly tries until exhausting the maximum number of steps.

**Common errors by GPT-4V agents**  Among the 550 failed examples from different settings in our sample, more than 75% exist *mouse click inaccuracies*, which is the most common error. The agent fails to click the correct coordinates despite planning detailed and accurate steps in their code comments, indicating strong planning but weak execution capabilities. Mouse click inaccuracies lead to two other frequent errors: *repetitive clicks* and *environmental noise dilemma*. Repetitive clicks occur when the agent repeatedly misclicks, adjusts, and fails, consuming too many steps. Environmental noise arises from clicking unintended objects, causing pop-ups, or opening unrelated applications. Due to a lack of prior knowledge about most professional software, it falls into a mismatch dilemma between the actions taken and the current state, and don't know how to get back to normal. Moreover, the agent lacks basic human-like cognition of web pages, such as not closing pop-ups in real-world web pages or being attracted by advertisement content, which affects its original correct judgment. Failures also arise from *misinterpretation of instructions* and *visual oversight*, highlighting the need for improvement in language and visual processing. See App. D.5.2 for the specific execution process.

**Discrepancies in task difficulty between agent and human**  We identify notable disparities in the perceived difficulty of tasks between humans and AI agents. Tasks that are intuitively simple for humans often present substantial challenges to agents, and conversely, tasks that humans find demanding can be more straightforward for agents to execute. You can find more details in Fig. 16 and App. D.5.3.

**Tasks where humans outperform agents**  These tasks mainly involve text-based and design-related work, such as "bold the font on this slide and add notes" or "erase all the highlighted marks in this document" (Fig. 12 Line 2). Since the Internet lacks such fine-grained data as the software execution process, the agent also lacks the corresponding training process, so its grounding ability is not good enough. The lack of understanding of GUI logic also causes poor performance on operations like selecting and scrolling.

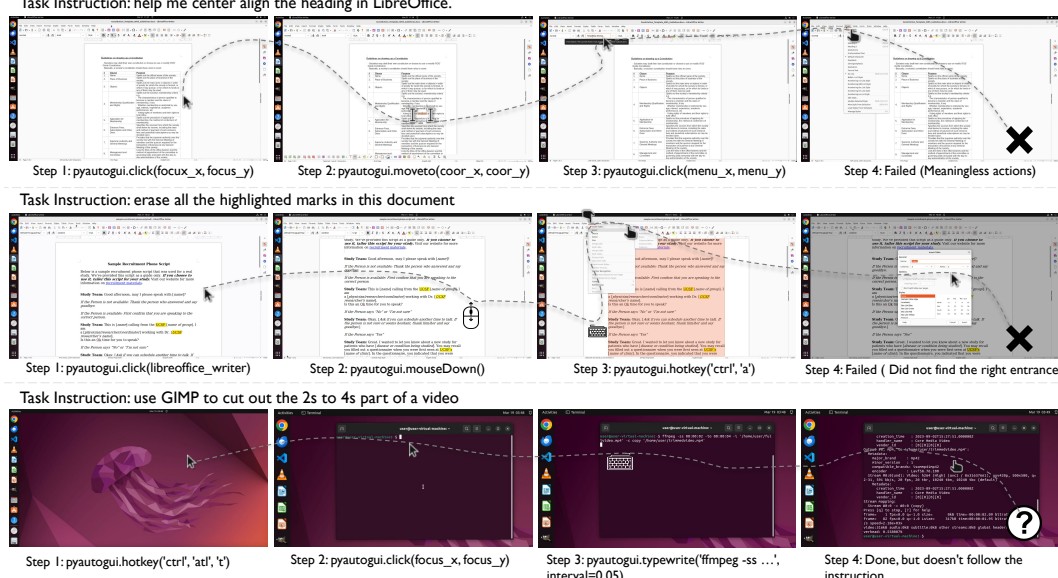

Figure 12: Screenshots of the three examples mentioned in the quality analysis. The first line is an example of GPT-4V failing at a very simple task, the second line is one example where agents face more difficulty than humans, and the third line is one example that is more difficult for humans than for agents.

**Tasks where agents outperform humans**     Tasks that the agent considers simple but humans find difficult are concentrated in "code solvability tasks", such as "monitor the system CPU for 30s and output the results" and "force close a process". These tasks require little or no GUI interaction and can be completed by executing complex codes and instructions. It's worth noting that completing through code sometimes mismatches with human instructions. In the task "use GIMP to cut out the 2s to 4s part of a video,(Fig. 12 Line 3)" the agent used "ffmpeg" command to complete the video cropping, ignoring the "use GIMP" requirement in the instructions.

Surprisingly, we discovered that agents are as prone to inefficiency in mechanically repetitive tasks, such as copying, pasting, and batch editing of Excel sheets, as humans. Humans frequently commit careless errors during execution. The shortcomings in agents stem either from the absence of an API or from insufficient training data related to the API, hindering their ability to efficiently process tasks in batches. Furthermore, sluggish response times can cause tasks to either time out or surpass the maximum allowed steps.

**Comparative analysis: Claude-3 vs. GPT-4V**     Although Claude outperforms GPT-4 in many benchmarks such as GSM8K, HumanEval, *etc.*, in our main experiment, we find that Claude has an average lower accuracy rate compared to GPT-4V by 2.84% to 7.76%. We find that Claude can provide satisfactory high-level solutions, but its grounding ability contains hallucinations in detail. For instance, Claude would interpret double-clicking a file as selecting it instead of opening it, treat column B in LibreOffice Calc software as column C, and enter text in the VS Code text replacement box without clicking on global replace. This shows that Claude can align well with human planning in problem-solving, but lacks excellent grounding ability when it comes to execution. Details can be seen in Fig. 17 and App. D.5.4.

## D.5    Qualitative Analysis Examples

Here we give the specific code execution process for the examples in the qualitative analysis in §D.4 and add more error cause analysis, trying to get insights for further improvement.

### D.5.1 Success and Failure Cases

> **Success Task:** *I downloaded an episode of Friends to practice listening, but I don't know how to remove the subtitles. Please help me remove the subtitles from the video and export it as "subtitles.srt" and store it in the same directory as the video.*

```python
# Action 1
import pyautogui
import time
# Move the mouse to the top bar of the media player window to access the menu
pyautogui.moveTo(800, 10)
# Assuming the menu is at the top of the screen
pyautogui.click()
time.sleep(0.5)

# Action 2
# Move the mouse to the Activities corner to open the GNOME Activities Overview
pyautogui.moveTo(10, 10)
# Assuming the Activities corner is at the top-left
pyautogui.click()
# Type `terminal' to search for the Terminal application
pyautogui.write(`terminal')
time.sleep(0.5)
# Press Enter to open the Terminal application
pyautogui.press(`enter')

# Action 3
# Focus the Terminal window and type the command to extract subtitles
pyautogui.click(640, 360)
# Clicking in the center of the screen to focus the Terminal
time.sleep(0.5)
pyautogui.write(`ffmpeg -i video.mp4 -map 0:s:0 subtitles.srt')
pyautogui.press(`enter')

# Action 4
# Focus the Terminal window and type the command
 to create a new video without subtitles
pyautogui.click(1100, 350)
# Clicking inside the Terminal window to focus it
pyautogui.write(`ffmpeg -i video.mp4 -c copy -sn no_subtitles_video.mp4')

Done!
```

> **Failure Task:** *Help me center align the heading in LibreOffice.*

```python
# Action 1
import pyautogui
import time
# Move the mouse to the LibreOffice Writer document icon and double-click
pyautogui.moveTo(1100, 720, duration=1)
pyautogui.doubleClick()

# Action 2
# Move the mouse to the beginning of the heading text
 and click and drag to select the heading
pyautogui.moveTo(200, 230, duration=1)
pyautogui.mouseDown()
pyautogui.moveTo(800, 230, duration=1)
pyautogui.mouseUp()
# Move the mouse to the center align button in the toolbar and click
pyautogui.moveTo(400, 150, duration=1)
```

```
pyautogui.click()

# Action 3
# Assuming the text is still selected,
move the mouse to the center align button in the toolbar and click
pyautogui.moveTo(400, 150, duration=1)
pyautogui.click()

# Action 4
# Move the mouse to the center align button in the toolbar and click
pyautogui.moveTo(340, 80, duration=1)
pyautogui.click()

Failed.
```

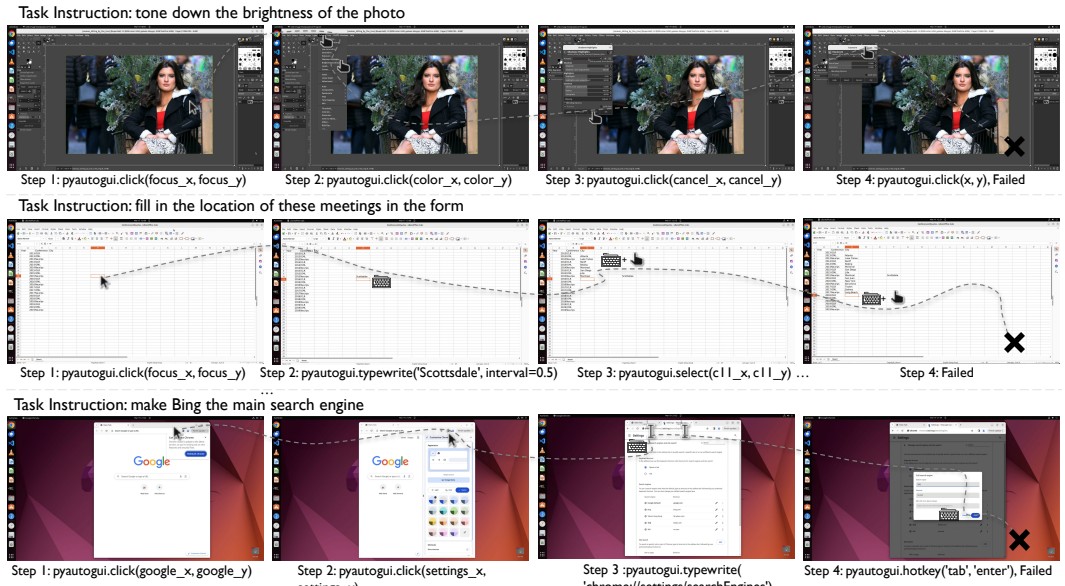

Task Instruction: tone down the brightness of the photo

Step 1: pyautogui.click(focus_x, focus_y)  Step 2: pyautogui.click(color_x, color_y)  Step 3: pyautogui.click(cancel_x, cancel_y)  Step 4: pyautogui.click(x, y), Failed

Task Instruction: fill in the location of these meetings in the form

Step 1: pyautogui.click(focus_x, focus_y)  Step 2: pyautogui.typewrite('Scottsdale', interval=0.5)  Step 3: pyautogui.select(cll_x, cll_y) …  Step 4: Failed

Task Instruction: make Bing the main search engine

Step 1: pyautogui.click(google_x, google_y)  Step 2: pyautogui.click(settings_x, settings_y)  Step 3 :pyautogui.typewrite('chrome://settings/searchEngines')  Step 4: pyautogui.hotkey('tab', 'enter'), Failed

Figure 13: The figure shows examples of usage errors in three types of software. The first row is for GIMP, the second row is for LibreOffice Calc, and the third row is for Google Chrome. Detailed code execution processes are provided below.

**GIMP Task:** *Could you tone down the brightness of my photo?*

```
# Action 1
import pyautogui
import time
# Coordinates for the image icon on the desktop
# Adding half of the width and height to get the center
image_icon_x, image_icon_y = 1793 + 60, 920 + 17
# Double-click on the image icon
pyautogui.doubleClick(image_icon_x, image_icon_y)

# Action 2
# Coordinates for the `Colors' menu
# Adding half of the width and height to get the center
colors_menu_x, colors_menu_y = 344 + 28, 64 + 12
# Click on the `Colors' menu
pyautogui.click(colors_menu_x, colors_menu_y)
```

```python
# Action 3
# Coordinates for the `Brightness-Contrast...' option in the `Colors' dropdown
# Assuming each menu item is approximately 28 pixels apart
brightness_contrast_option_x, brightness_contrast_option_y
= 344 + 28, 64 + 12 + 28 * 6
# Click on the `Brightness-Contrast...' option
pyautogui.click(brightness_contrast_option_x, brightness_contrast_option_y)

# Action 4
# Coordinates for the `Cancel' button in the `Shadows-Highlights' dialog
# Adding half the width and height to get the center
cancel_button_x, cancel_button_y = 375 + 42, 625 + 16
# Click the `Cancel' button
pyautogui.click(cancel_button_x, cancel_button_y)
# Coordinates for the `Brightness-Contrast...' option in the `Colors' dropdown
# Adjusted index for Brightness-Contrast option
brightness_contrast_option_x, brightness_contrast_option_y
= 344 + 28, 64 + 12 + 28 * 5
pyautogui.click(brightness_contrast_option_x, brightness_contrast_option_y)

# Action 5
# Coordinates for the `Cancel' button in the `Exposure' dialog
# Adding half the width and height to get the center
cancel_button_x, cancel_button_y
= 375 + 42, 391 + 16
pyautogui.click(cancel_button_x, cancel_button_y)
# Coordinates for the `Brightness-Contrast...' option in the `Colors' dropdown
# Adjusted index for Brightness-Contrast option
brightness_contrast_option_x, brightness_contrast_option_y
= 344 + 28, 64 + 12 + 28 $*$ 6
pyautogui.click(brightness_contrast_option_x, brightness_contrast_option_y)

# Action 6
# Coordinates for the `Cancel' button in the `Color Balance' dialog
# Adding half the width and height to get the center
cancel_button_x, cancel_button_y
= 375 + 42, 588 + 16
pyautogui.click(cancel_button_x, cancel_button_y)
# Since the `Brightness-Contrast...' option was not correctly opened before,
we will use an estimated position based on the dropdown menu.
# Assuming `Brightness-Contrast...' is the second option in the `Colors' menu.
brightness_contrast_option_x, brightness_contrast_option_y
= colors_menu_x, colors_menu_y + 28 * 2
pyautogui.click(brightness_contrast_option_x, brightness_contrast_option_y)

# Action 7
# Coordinates for the `Cancel' button in the `Color Temperature' dialog
# Adding half the width and height to get the center
cancel_button_x, cancel_button_y = 375 + 42, 403 + 16
pyautogui.click(cancel_button_x, cancel_button_y)
# Since the `Brightness-Contrast...' option was not correctly opened before,
we will use an estimated position based on the dropdown menu.
# Assuming `Brightness-Contrast...' is the second option in the `Colors' menu.
brightness_contrast_option_x, brightness_contrast_option_y
= colors_menu_x, colors_menu_y + 28 * 2
# Click on the `Brightness-Contrast...' option
pyautogui.click(brightness_contrast_option_x, brightness_contrast_option_y)

# ... More Wrong Duplicate Actions; Can not find the right coordination.

# Failed
```

Libreoffice calc Task: *I now want to count the meeting cities of the three machine learning conferences in the past ten years from 2013 to 2019(including 2013 and 2019). I have listed the names and years of the conferences in excel. Please fill in the vacant locations.*

```
# Action 1
# Move to cell C2 which is the first empty cell under `City' for ICLR 2013
pyautogui.click(319, 222)
# Coordinates of cell C2
time.sleep(0.5)
# Type the city name for ICLR 2013
pyautogui.write(`Scottsdale')
# Press Enter to move to the next cell
pyautogui.press('enter')

# Action 2
# Move to cell C3 which is the city cell for ICML 2013
pyautogui.click(319, 268)
# Coordinates of cell C3
# Type the city name for ICML 2013
pyautogui.write(`Atlanta')
# Press Enter to move to the next cell
pyautogui.press('enter')

# Action 3
# Move to cell C4 which is the city cell for NeurIPS 2013
pyautogui.click(319, 289)
# Coordinates of cell C4
# Type the city name for NeurIPS 2013
pyautogui.write(`Lake Tahoe')
# Press Enter to move to the next cell
pyautogui.press('enter')"

... More copy-and-paste operations

# Action 15
# Move to cell C16 which is the city cell for NeurIPS 2017
pyautogui.click(319, 541)
# Coordinates of cell C16
# Type the city name for NeurIPS 2017
pyautogui.write(`Long Beach')
# Press Enter to move to the next cell
pyautogui.press('enter')

Failed.
```

Chrome Task: *Can you make Bing the main search thingy when I look stuff up on the internet?*

```
# Action 1
pyautogui.click(x=1280, y=50)
# Approximate coordinates for the `More' button
pyautogui.move(0, 150)
# Moving the cursor down to `Settings' in the drop-down menu
pyautogui.click()
# Clicking on `Settings'

# Action 2
pyautogui.click(tag_31)
# Click on the `Finish update' button to
approximate the location of `More' button
pyautogui.move(-50, 0)
# Move left from the `Finish update' button to approximate the `More' button
```

```
pyautogui.click()
# Click on the `More' button
time.sleep(0.5)
pyautogui.move(0, 150)
# Move down to the `Settings' option in the drop-down menu
pyautogui.click()
# Click on `Settings'

# Action 3
# Click on the `Customise Chrome' button which should be the `More' button
time.sleep(1)
# Wait for the dropdown menu to appear
# The `Settings' option is usually located towards the bottom of
the dropdown menu, so I'll move the cursor down to reach it
pyautogui.move(0, 200)
# Move down to the `Settings' option in the drop-down menu
pyautogui.click()
# Click on `Settings'

# Action 4
pyautogui.click(tag_47)
# Click on the `Close' button to close the `Customise Chrome' menu
pyautogui.click(tag_25)
# Click on the `New Tab' button to open a new tab
time.sleep(1)
# Wait for the new tab to open
pyautogui.typewrite(`chrome://settings/searchEngines')
# Type the direct URL to the search engine settings page
pyautogui.press(`enter')

# Action 5
# Simulate pressing the `Tab' key multiple times to navigate through the options
for _ in range(10):
# Assuming it might take around 10 tabs to reach the search engine list
    pyautogui.press(`tab')
    time.sleep(0.5)
# Wait for the focus to move to the next element
# Simulate pressing the `Enter' key to select an option,
which would be done once we reach the Bing option
pyautogui.press('enter')

# Action 6
pyautogui.click(tag_42)
# Click on the `Cancel' button to exit the edit mode for the '360' search engine

Failed.
```

### D.5.2 Common Errors by GPT-4V Agents

In the following two figures ( Figure 15 and Figure 14), we take the web page tasks as examples to show the two most common types of errors in GPT4-V: *mouse click inaccuracies* and *inadequate handling of environmental noise*.

Task Instruction: On next Monday, look up a flight from Mumbai to Stockholm.

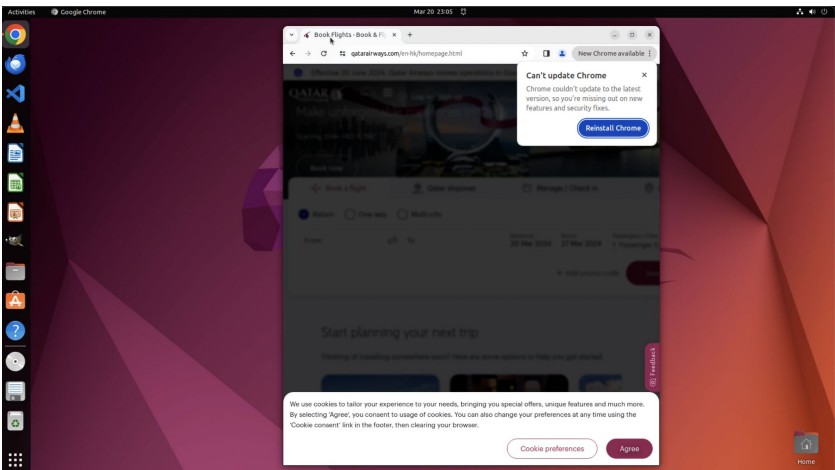

Figure 14: The error shown in the figure is due to mouse click inaccuracies. The agent was supposed to click on the product category images of the Nike official store and follow the instructions to search for women's jerseys priced over 60 dollars. However, due to a coordinate calculation error, it mistakenly clicked on the favorite button, creating a significant discrepancy between the current state and the target state. Under these circumstances, the agent is unable to backtrack to the previous state and start over.

Task Instruction: Browse the list of women's Nike jerseys over $60.

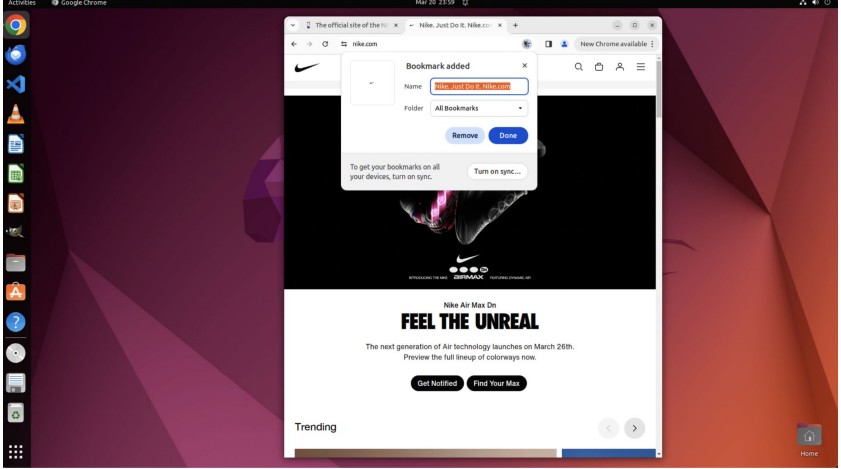

Figure 15: The error shown in the figure is due to inadequate handling of environmental noise. When the agent encounters web pages with cookies or pop-up ads, its unfamiliarity with the structure of such pages leads to attempts to interact directly with the web page without bypassing the cookies or pop-ups. Since the pop-ups or cookie requests are not closed, all actions are ineffective.

### D.5.3 Discrepancies in Task Difficulty between Agent and Human

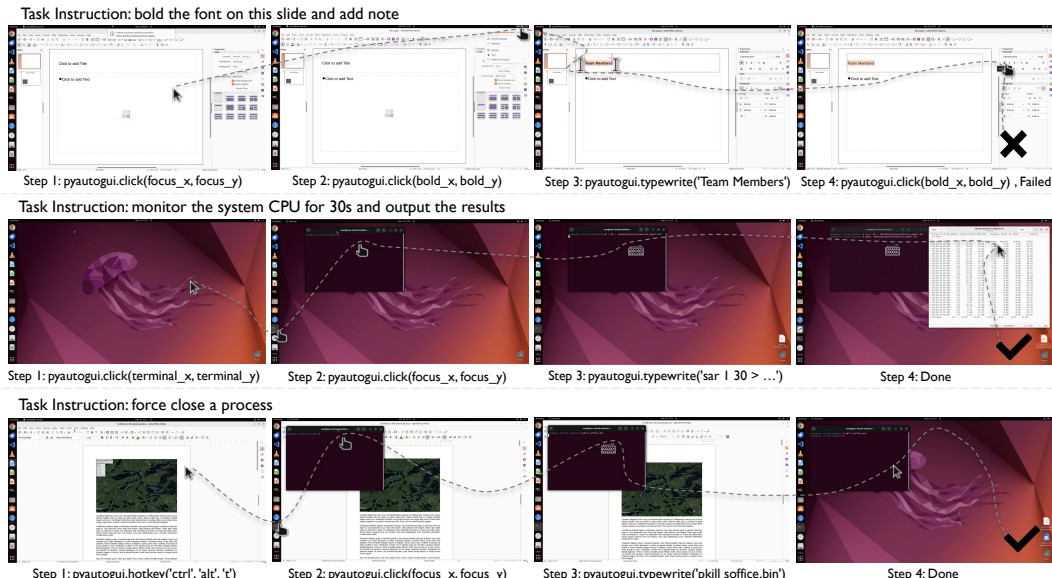

Figure 16: Supplementary examples of tasks performed by humans and agents.

---

**Human > Agent Task:** *erase all the highlighted marks in this document*

```python
# Action 1
import pyautogui
import time
# Move to the `Character Highlighting Color' button and click
pyautogui.moveTo(1020, 122)
pyautogui.click()

# Action 2
# Assuming the `No Color' option is the first option in the dropdown
# Move the mouse slightly below the `Character Highlighting Color' button
to click on `No Color'
pyautogui.moveTo(1020, 180)
time.sleep(0.5)
pyautogui.click()

# Action 3
# Move to the `Save' button and click
pyautogui.moveTo(184, 108)
time.sleep(0.5)
pyautogui.click()

# Failed.
```

---

**Human > Agent Task:** *On page 2, add a note same to the title into the slide. Make the font of title bold.*

```python
# Action 1
# Click on the vertical scrollbar to scroll down and reveal more slides
pyautogui.click(tag_43)

# Action 2
# Click on the second slide thumbnail to select it
```

```
pyautogui.click(tag_2)

# Action 3
# Click on the text box to add text
pyautogui.click(tag_42)
pyautogui.typewrite(`Team Members', interval=0.1)
# Type the title
pyautogui.hotkey(`ctrl', `a')
# Select all text
pyautogui.hotkey(`ctrl', `b')
# Make the text bold

# Action 4
# Click on the Bold button to apply bold formatting
pyautogui.click(tag_68)

Failed.
```

## Agent > Human Task: *use GIMP to cut out the 2s to 4s part of a video*

```
# Action 1
import pyautogui
import time
# Move the mouse to the assumed terminal icon position
at the bottom of the dock and click to open it
# Assuming the terminal icon is at the bottom of the dock
pyautogui.moveTo(70, 1350, duration=1)
pyautogui.click()
# Type the command to use ffmpeg to trim the video
ffmpeg_command = `ffmpeg -ss 00:00:02 -to 00:00:04 -i /home/user/fullvideo.mp4
-c copy /home/user/trimmedvideo.mp4'
pyautogui.write(ffmpeg_command, interval=0.05)
pyautogui.press('enter')

# Action 2
# Close the search dialog
pyautogui.press('esc')
# Open the terminal using a keyboard shortcut (Ctrl+Alt+T is a common shortcut
for opening a terminal in many Linux distributions)
pyautogui.hotkey('ctrl', 'alt', 't')

# Action 3
# Move the mouse to the terminal and click to focus on the terminal window
pyautogui.moveTo(260 + 722/2, 103 + 434/2, duration=1)
pyautogui.click()
# Type the command to use ffmpeg to trim the video
pyautogui.write(ffmpeg_command, interval=0.05)
pyautogui.press(`enter')

# Done, but not follow the demanding in instruction.
```

## Agent > Human Task: *I want to understand the resource usage of my Ubuntu system under normal workloads. Please use the 'sar' command in the 'sysstat' toolkit to monitor system activity, evaluate the status once every second for 30 seconds, output the results to "System Resources Report.txt" under Desktop.*

```
# Action 1
import pyautogui
import time
```

```python
# Press the keyboard shortcut to open the terminal
pyautogui.hotkey(`ctrl', `alt', `t')
time.sleep(1)
# Wait a moment for the terminal to open

# Action 2
# Click inside the terminal to focus on it
pyautogui.click(600, 300)
# Assuming the terminal is in the upper left area of the screen
n# Type the command to monitor system activity
pyautogui.typewrite(`sar 1 30 > ~/Desktop/System Resources Report.txt')

# Action 3
# Click on the `System Resources Report.txt' file to select it
pyautogui.click(1850, 940)
# Coordinates taken from the label position and size
time.sleep(0.5)
# Right-click to open the context menu
pyautogui.click(1850, 940, button=`right')
# Move to the `Open With Other Application' option in the context menu
pyautogui.move(0, 70)
# Assuming `Open With Other Application' is 70 pixels
down from the right-click position
# Click on `Open With Other Application'
pyautogui.click()
time.sleep(0.5)
# Type `gedit' to select the text editor
pyautogui.typewrite(`gedit')
# Press `Enter' to open the file with gedit
pyautogui.press(`enter')

# Action 4
# Click inside the terminal to ensure it's in focus
pyautogui.click(500, 200)
# Coordinates are estimated to be within the terminal window
# Type the command to open the file with gedit
pyautogui.typewrite(`gedit ~/Desktop/System Resources Report.txt')

Done!
```

Agent > Human Task: *Hey, my LibreOffice Writer seems to have frozen and I can't get it to close normally. Can you help me force quit the application from the command line? I'm on Ubuntu and I don't want to restart my computer or lose any other work I have open.*

```python
# Action 1
import pyautogui
import time
# Open a terminal window
pyautogui.hotkey(`ctrl', `alt', `t')
time.sleep(1)
# Type the command to kill all instances of LibreOffice Writer
pyautogui.typewrite(`killall soffice.bin')

# Action 2
# Open a terminal window
pyautogui.hotkey(`ctrl', `alt', `t')
time.sleep(1)
# Type the command to kill all instances of LibreOffice Writer
pyautogui.typewrite(`killall soffice.bin')

# Action 3
```

```
# Ensure the terminal window is focused
pyautogui.click(500, 200)
# Assuming the terminal is at this position
# Type the pkill command to forcefully kill the LibreOffice Writer application
pyautogui.typewrite(`pkill -9 soffice.bin')

Done!
```

### D.5.4 Comparative Analysis: Claude-3 vs. GPT-4V

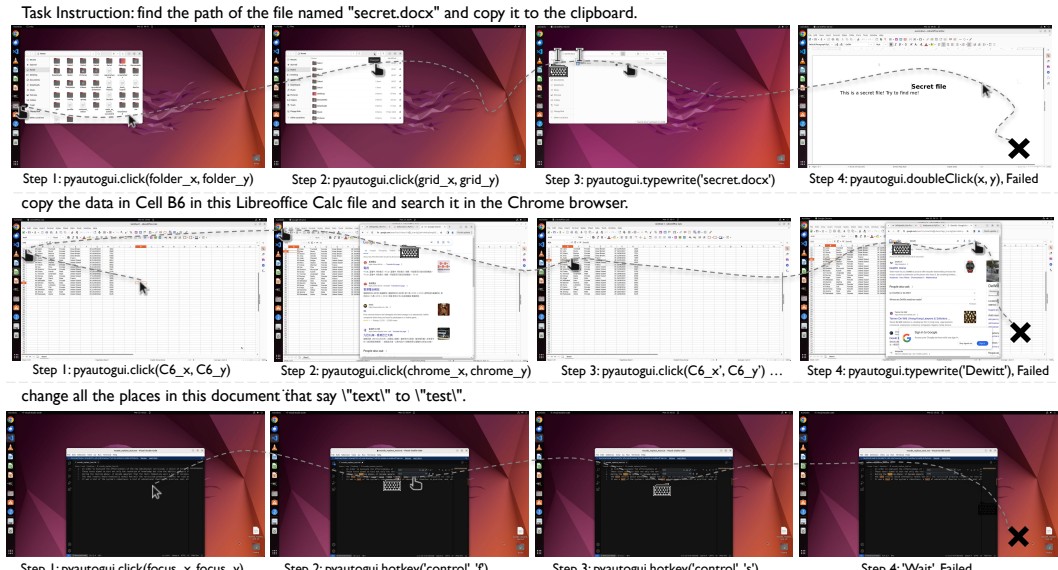

Figure 17: Screenshots of error example analysis of Claude-3 Opus.

Claude3 Failed Task1: *I remember there is a file named "secret.docx" on this computer, but I can't remember where it is. Please find the path where this file is stored and copy it to the clipboard.*

```
# Action 1
pyautogui.click(tag_10)
# click the Files icon to open the file manager

# Action 2
pyautogui.click(tag_21)
# click the search icon to open the search bar

# Action 3
pyautogui.typewrite(`secret.docx')
# type the file name to search for
pyautogui.press(`enter')
# press enter to start the search

# Action 4
pyautogui.doubleClick(tag_52)
# double click the secret.docx file to select it
pyautogui.hotkey(`ctrl', 'c')
# press Ctrl+C to copy the selected file path to clipboard

Failed.
```

**Claude3 Failed Task2:** *copy the data in Cell B6 in this Libreoffice Calc file and search it in the Chrome browser.*

```python
# Action 1
import pyautogui
import time
# Scroll down to view more search results if needed
pyautogui.press(`pagedown')
time.sleep(1)
pyautogui.press(`pagedown')
time.sleep(2)
# The answer appears to be in the snippet under the first search result
# `TV 31. u76f4u64adu4e2d.' which translates to `TV 31. Live.'
# So channel 31 in Hong Kong is RTHK TV 31
print(`Found the answer. Channel 31 in Hong Kong is RTHK TV 31.')
# The original task of copying cell B36 from the spreadsheet
and searching it in the browser is complete
print(`Task complete.')

# Action 2
# Activate the Chrome browser window
pyautogui.click(tag_3)
time.sleep(1)
# Click on the address/search bar
pyautogui.click(tag_30)
# Paste the copied value `Dewitt'
pyautogui.hotkey(`ctrl', `v')
# Press Enter to perform the search
pyautogui.press(`enter')
print(`Searched for `Dewitt' in the Chrome browser.')

# Action 3
print(`DONE')

# Action 4
print(`DONE')

Failed.
```

**Claude3 Failed Task3:** *Please help me change all the places in this document that say 'text' to 'test'.*

```python
# Action 1
pyautogui.hotkey(`ctrl', `h')
pyautogui.write(`text')
pyautogui.press(`tab')
pyautogui.write(`test')
pyautogui.press(`enter')

# Action 2
pyautogui.press(`enter')

# Action 3
pyautogui.hotkey(`ctrl', `s')

Failed.
```

## D.6 Analysis on SoM Effectiveness

**Task w/o SoM:** *turning off the code checking function of the python plugin in vscode*

```python
# Action 1
import pyautogui
import time
# Move the mouse to the gear icon (Manage) and click it
pyautogui.moveTo(472, 1034, duration=0.5)
pyautogui.click()

# Action 2
# Move the mouse to the Šettings¨ menu item and click it
# A rough central point of the `Settings' menu item
pyautogui.moveTo(497 + 385/2, 832 + 25/2, duration=0.5)
pyautogui.click()

# Action 3
# Move the mouse to the search settings bar and click it
# A rough central point of the search settings bar
pyautogui.moveTo(783 + 587/2, 390 + 19/2, duration=0.5)
pyautogui.click()
# Type the query to search for Python-related settings
pyautogui.write('python', interval=0.05)

# Action 4
# Move the mouse to the `Edit in settings.json' button and click it
# A rough central point of the button
pyautogui.moveTo(990 + 116/2, 624 + 15/2, duration=0.5)
pyautogui.click()

# Action 5
# Move the cursor to the correct position
pyautogui.moveTo(889 + 1/2, 432 + 1/2, duration=0.5)
pyautogui.click()
# Type the configuration to disable Python missing imports error reporting
pyautogui.write(`reportMissingImports: `none'', interval=0.05)
pyautogui.press(`enter')
# Save the changes
pyautogui.hotkey(`ctrl', `s')

# Done
```

**Task w/ SoM:** *turning off the code checking function of the python plugin in vscode*

```python
# Action 1
import pyautogui
import time
pyautogui.hotkey(`ctrl', `shift', `p')

# Action 2
pyautogui.write(`Preferences: Open Settings (UI)', interval=0.05)
time.sleep(0.5)
pyautogui.press(`enter')

# Action 3
# Click on the search settings input field
pyautogui.click(x=600, y=175)
# These coordinates may need to be adjusted for your screen resolution
# Type `Python' into the search field
pyautogui.write(`Python', interval=0.05)
```

```
pyautogui.press(`enter')

# Action 4
# Click on the search settings bar and type in the query again
pyautogui.click(x=800, y=150)
# Click on the search bar, the coordinates may need adjustment
pyautogui.typewrite(`Python missing imports', interval=0.25)

# ... Can not find the right click area

# Failed
```

### D.6.1 Analysis on SoM Effectiveness

The SoM, effective for web tasks focusing only on labeled bounding boxes, **did not** excel in our diverse tasks that required more expertise and flexibility. It shortens the action space of the agent and thus hinders the agent's exploration and adaptability. For example in Figure 18, in the task of "turning off the code checking function of the python plugin." in VS Code, agents without SoM succeeded by editing the settings.json, whereas SoM-equipped agents struggled with finding and deselecting the checkbox in the settings. The latter has a longer action path and fails due to incorrect mouse clicks. Furthermore, SoM's labeling of elements in professional software also has such errors, that agents without SoM can use a11y tree to calculate and adjust coordinates, while the elements' blocks under SoM can be hard to change during the task.

Task Instruction: turning off the code checking function of the python plugin in vscode

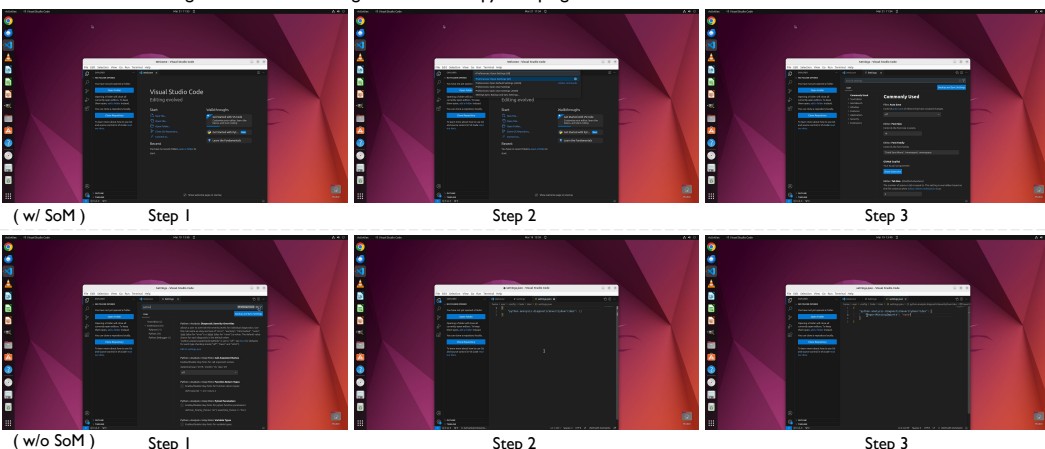

Figure 18: A task showcase where the SoM-equipped agent tried to find interactive settings, while the screen-a11ytree-equipped agents directly modified the value in the setting.json file.

## E    Limitations and Future Work

We identify several potential directions for community development and progress toward general-purpose agents for computer operation:

**Enhancing VLM capabilities for efficient and robust GUI interactions**    For foundation model development, we need to boost the efficiency of our models, enabling them to process much longer contexts and perform inference computations efficiently, akin to the robotics community [6, 7] to better handle real-world cases. Enhancements in VLMs' GUI grounding capabilities that is robust to application windows changes and are also sought, focusing on the accurate understanding and generation of precise actions aligned with given instructions. Moreover, amplifying VLMs' ability to comprehend context in the form of images is a pivotal goal, since it is crucial to enable history encoding using images so that we can build memory and reflection upon that. These improvements may require more efforts in the upstream pre-training stage, downstream fine-tuning stage, and even in the model structure itself, as pointed out in previous work [12, 21, 37].

**Advancing agent methodologies for exploration, memory, and reflection**    The next-level approach encompasses designing more effective agent architectures that augment the agents' abilities to explore autonomously and synthesize their findings. The agents face challenges in leveraging lengthy raw observation and

action records. It's fascinating to explore novel methods for encoding this history, incorporating efficient memory and reflection solutions to condense contextual information and aid the agent in extracting key information. Additionally, integrating knowledge grounding into (V)LLM agents through memory mechanisms is a promising avenue as well. Moreover, practice GUI assistants also require features of personalization and customization. These features rely on techniques such as user profiling and retaining memories from long-term user-assistant interactions. Additionally, crafting protocols specifically for digital agents operating within GUI and CLI interfaces aims at facilitating efficient actions is also an essential thing for the feasibility of general-purpose digital agents in the mid-short term.

**Addressing the safety challenges of agents in realistic environments**   The safety of agents is a critical issue if applying a built agent in fully realistic environments, the developed universal digital agent could potentially be used to bypass CAPTCHA systems in the future, as noted in [46]. However, due to the currently limited capabilities of agents, we have not observed any harmful and damaging behaviors during our experiments, an automatic agent has the opportunity to damage patent rights, abuse accounts, attempt to exploit software vulnerabilities to create viruses, or engage in attacks. Currently, we adopt virtual machines to make it difficult for developing digital agents to cause irreversible damage to our host machines. However, there still lacks a reliable metric to assess the safety of an agent developed in an isolated environment. The current evaluation functions mainly focus on the results closely regarding the task instructions, assess only the correctness of task completion, and pay little attention to potential unnecessary damaging actions of agents. Owing to the complexity of a complete computer environment, we didn't work out an efficient way to detect the latent side effects of the agent. Consequently, how to assess and control potential behaviors in open and real environments through environmental constraints and agent training is an important further direction of research.

**Expanding and refining data and environments for agent development**   In terms of datasets and environments, we can broaden the scope to cover more specialized domains, including real-sector needs in healthcare, education, industry, transportation, and personalized requirements. Efforts can be made to ensure our environment's seamless deployment across various hardware and software settings. The variance of a11y tree quality across different applications is also noticed. Although the problem is not remarkable in the applications currently included, there is no guarantee of that the application developers obey the a11y convention and offer clear and meaningful descriptions for GUI elements. More intelligent approaches to filter redundant a11y tree elements and to handle latently missing elements deserve careful investigation as well. We also highlight the necessity of a painless data collection method, allowing for the effortless acquisition of computer operation data and its transformation into agent capabilities.

# F   Distribution & License

The proposed OSWORLD platform and task set are open-sourced under the Apache-2.0 license and are available at `https://os-world.github.io/` and `https://github.com/xlang-ai/OSWorld`. The authors claim that OSWORLD will be permanently maintained for future environments.

A non-exhaustive list of artifacts used in the development of OSWORLD environment includes: pyautogui [20], flask [21], python-pptx [22], python-docx [23], OpenCV [24], EasyOCR [25], *etc*. They are released under licenses BSD-3-Clause, MIT, and Apache-2.0. A non-exhaustive list of artifacts used in the experiments includes: openai-python [26], google-generativeai [27], groq-python [28], *etc*. They are released under licenses Apache-2.0. The authors claim that the usage completely obeys the licenses.

# G   Datasheet

## G.1   Motivation

- **For what purpose was the dataset created?** Was there a specific task in mind? Was there a specific gap that needed to be filled? Please provide a description.

---

[20]`https://github.com/asweigart/pyautogui`
[21]`https://github.com/pallets/flask`
[22]`https://github.com/scanny/python-pptx`
[23]`https://github.com/python-openxml/python-docx`
[24]`https://github.com/opencv/opencv-python`
[25]`https://github.com/JaidedAI/EasyOCR`
[26]`https://github.com/openai/openai-python`
[27]`https://github.com/GoogleCloudPlatform/generative-ai`
[28]`https://github.com/groq/groq-python`

The environment and dataset for OSWORLD are created to address a critical need for a comprehensive benchmark capable of evaluating the performance and capabilities of universal digital agents across real-world, operating system-level tasks. The motivation stemmed from the recognized gaps in existing benchmarks focusing predominantly on isolated applications or single-domain tasks. No existing dataset adequately represents the range of actions, coupled with real-world complexities, faced by operational software agents in a truly interactive operating system environment. This dataset enables the holistic assessment of agents in multi-application contexts and multi-modal interactions, with tasks requiring both comprehension and navigation of complex user interfaces across various commonly used software and operating systems.

- **Who created the dataset (*e.g.*, which team, research group) and on behalf of which entity (*e.g.*, company, institution, organization)?**
  Tianbao Xie, Danyang Zhang, Jixuan Chen, Xiaochuan Li, Siheng Zhao, Ruisheng Cao, Toh Jing Hua, Zhoujun Cheng, Dongchan Shin, Fangyu Lei, Yitao Liu, Yiheng Xu, Shuyan Zhou, Silvio Savarese, Caiming Xiong, Victor Zhong, and Tao Yu from the XLANG Lab [29] from University of Hong Kong, Carnegie Mellon University, Salesforce Research and University of Waterloo create the environment and the task set.

## G.2   Composition

- **What do the instances that comprise the dataset represent (*e.g.*, documents, photos, people, countries)?**  Are there multiple types of instances (*e.g.*, movies, users, and ratings; people and interactions between them; nodes and edges)? Please provide a description.

  The instances within the OSWORLD dataset comprise a diverse collection of tasks simulated within real operating system environments, specifically focusing on Ubuntu and Windows systems. These tasks represent a variety of real-world computer usage scenarios, encompassing multiple applications and interfaces. Each instance includes detailed natural language instructions, setup configurations with corresponding files, and setup actions for environment initialization. The instances further encompass a variety of operation types such as document editing, file operations, email activities, and multimedia management. This design ensures a comprehensive evaluation of digital agents' ability to perform highly integrated, multi-modal interactions across different software applications and user interfaces. The dataset, therefore, offers a rich milieu of complex tasks requiring both cognitive understanding and interaction capabilities from the executing agents.

- **How many instances are there in total (of each type, if appropriate)?**
  There are a total of 369 tasks for the Ubuntu operating system and 43 tasks designed for Windows within the OSWORLD benchmark. These tasks encompass a variety of task types including multi-app workflow tasks, single-application tasks, and integrated tasks from related datasets. Specifically, the dataset includes 268 single-app tasks (72.6%), 101 multi-app workflow tasks (27.4%), and 30 infeasible tasks (8.1%). Additionally, 84 tasks (22.8%) are integrated from other datasets.

- **Does the dataset contain all possible instances or is it a sample (not necessarily random) of instances from a larger set?** If the dataset is a sample, then what is the larger set? Is the sample representative of the larger set (*e.g.*, geographic coverage)? If so, please describe how this representativeness was validated/verified. If it is not representative of the larger set, please describe why not (*e.g.*, to cover a more diverse range of instances, because instances were withheld or unavailable).

  The dataset in OSWORLD is a curated sample from a larger set of potential real-world tasks. These samples were selected based on their popularity, helpfulness, and diversity, as judged by view counts and user votes on various platforms including official guidelines, tutorials, forums, and educational courses. This selection process intends to cover a broad spectrum of typical and critical tasks that users perform using computer systems across different operating systems and software environments. While this dataset aims to cover various practical and high-impact scenarios, it does not claim to represent all possible tasks due to the vastness and evolving nature of computer operations. It emphasizes tasks with significant educational and practical utility to ensure that the benchmark provides both comprehensive and challenging objectives for developing advanced digital agents. The representativeness has not been systematically verified statistically due to the qualitative selection criteria based on task relevance and utility rather than mere statistical representation.

- **What data does each instance consist of?** "Raw" data (*e.g.*, unprocessed text or images) or features? In either case, please provide a description.

  Each instance within the OSWORLD dataset consists of both "raw" data and structured annotations. The main components of each task instance are:
  - **Natural language instruction:** Each task is accompanied by a natural language description that guides the agent on how to execute the task. These instructions are crafted from real-world guidelines and scenarios.

---
[29]https://www.xlang.ai/

- **Setup configuration:** This includes necessary files and setup actions required for initializing the task environment as the cases of the real world.
- **Evaluation scripts:** Scripts designed to programmatically assess whether the task has been successfully completed by the agent. These scripts function as objective measures of task performance.
- **Accessibility Trees and Screenshots:** For GUI-based tasks, environments from instances include accessibility trees that provide structured data regarding the GUI elements, and screenshots that offer visual context. These components are crucial for agents that rely on visual and structural interpretations to navigate and interact within the tasks.

These data elements combine raw and processed formats to deliver comprehensive task-specific information that supports both the execution and the evaluation of automated agents across varied operating systems and interfaces.

- **Is there a label or target associated with each instance?** If so, please provide a description.

Yes, each task in the OSWORLD benchmark is associated with a specific label describing the objective of the task. The label is a natural language instruction that indicates what the computer agent needs to achieve. Additionally, each task is accompanied by a setup configuration and an evaluation script, which are used to verify if the task has been accomplished successfully according to the predefined criteria. This structured approach ensures that agents are assessed accurately on their ability to perform a variety of real-world computing tasks.

- **Is any information missing from individual instances?** If so, please provide a description, explaining why this information is missing (*e.g.*, because it was unavailable). This does not include intentionally removed information, but might include, *e.g.*, redacted text.

Yes, some instances may lack certain details that were not available from the source materials. These sources include forums, tutorials, how-to websites, and video content from platforms like YouTube and TikTok. This missing information could be due to incomplete guidelines or advice given in the source materials, which were not exhaustively detailed. Additionally, although extensive efforts were made to create comprehensive annotations, some specifics might be inherently unobtainable due to the nature of the source content.

- **Are there any errors, sources of noise, or redundancies in the dataset?** If so, please provide a description.

Yes, our dataset may include various sources of noise or redundancies which typically arise from the variety of different real-world sources used for task examples, such as forums, tutorials, and guidelines. Additionally, the dataset includes deliberately collected infeasible tasks due to feature deprecation or hallucinated features, as highlighted in the design process. These instances are used to test the robustness and error-handling capabilities of digital agents but may be perceived as noise if not correctly accounted for in training. Redundancies may occur as multiple tasks may involve similar applications or operations, albeit with different specific instructions or end goals.

- **Is the dataset self-contained, or does it link to or otherwise rely on external resources (*e.g.*, websites, tweets, other datasets)?** If it links to or relies on external resources, a) are there guarantees that they will exist, and remain constant, over time; b) are there official archival versions of the complete dataset (*i.e.*, including the external resources as they existed at the time the dataset was created); c) are there any restrictions (*e.g.*, licenses, fees) associated with any of the external resources that might apply to a dataset consumer? Please provide descriptions of all external resources and any restrictions associated with them, as well as links or other access points, as appropriate.

The OSWORLD dataset is not entirely self-contained as it integrates tasks and guidelines derived from various external resources such as forums, tutorials, how-to websites, and public guidelines. These include websites like WikiHow, Reddit, Quora, StackOverflow, and video tutorials from platforms like YouTube and TikTok, among others.

a) There are no guarantees that the external resources will exist indefinitely or remain constant over time, which could impact the dataset's reliability and the reproducibility of the tasks.

b) There are no official archival versions of the complete dataset that include external resources as they existed at the time the dataset was created. Consumers of the dataset will need to rely on the current versions of these resources, which may be subject to changes.

c) Some of the external resources may have their own copyright or licensing restrictions which might apply to a dataset consumer. Each resource's terms of service and copyright laws should be reviewed prior to use to ensure compliance.

External resources vary in nature and can often have different terms of use; it is advised for users of the OSWORLD dataset to verify any potential restrictions or licensing requirements individually.

- **Does the dataset contain data that might be considered confidential (*e.g.*, data that is protected by legal privilege or by doctor–patient confidentiality, data that includes the content of individuals' non-public communications)?** If so, please provide a description.

No, the dataset does not contain any data that could be considered confidential. The tasks and examples within the OSWORLD dataset are created from widely accessible sources such as public forums, official tutorials, and openly available guidelines. All data in use, including task instructions and configurations, are derived from non-confidential, non-privileged, and public communications aimed at demonstrating real-world computer usage scenarios.

- **Does the dataset contain data that, if viewed directly, might be offensive, insulting, threatening, or might otherwise cause anxiety?** If so, please describe why.

No, the dataset primarily consists of tasks defined and executed on operating systems using data from tutorials, guides, and how-to information that is publicly available and widely considered to be educational and informational. Therefore, it is unlikely to contain content that could be viewed as offensive, insulting, or threatening.

- **Does the dataset identify any subpopulations (*e.g.*, by age, gender)?** If so, please describe how these subpopulations are identified and provide a description of their respective distributions within the dataset.

No, the dataset does not identify or differentiate any subpopulations based on age, gender, or other demographic factors. The focus is on the diversity of tasks related to computer usage across different operating systems and software environments.

- **Is it possible to identify individuals (*i.e.*, one or more natural persons), either directly or indirectly (*i.e.*, in combination with other data) from the dataset?** If so, please describe how.

No, it is not possible to identify individuals directly or indirectly from the dataset. The tasks and examples in OSWORLD are derived from general public sources such as forums, tutorials, and guidelines that are widely accessible and do not contain personal information. Furthermore, any task setup or descriptive data is specifically designed or curated to avoid inclusion of any personal identifiers or sensitive information that could lead to the re-identification of individuals.

- **Does the dataset contain data that might be considered sensitive in any way (*e.g.*, data that reveals race or ethnic origins, sexual orientations, religious beliefs, political opinions or union memberships, or locations; financial or health data; biometric or genetic data; forms of government identification, such as social security numbers; criminal history)?** If so, please provide a description.

The OSWORLD dataset does not contain data that might be considered sensitive. The dataset is constructed from publicly available sources such as forums, tutorials, and guidelines and solely focuses on operational tasks within computer environments. It does not include any personal data categories or information related to individuals. The tasks are generated and anonymized without incorporating any sensitive or personal data elements.

### G.3 Collection Process

- **How was the data associated with each instance acquired?** Was the data directly observable (*e.g.*, raw text, movie ratings), reported by subjects (*e.g.*, survey responses), or indirectly inferred/derived from other data (*e.g.*, part-of-speech tags, model-based guesses for age or language)? If the data was reported by subjects or indirectly inferred/derived from other data, was the data validated/verified? If so, please describe how.

The data for each task in OSWORLD was meticulously collected from a variety of credible sources including forums, official tutorials, instructional video content from platforms like YouTube and TikTok, and Q&A websites such as Reddit and Quora. The tasks were inspired by real-world computer usage scenarios which are often discussed in these sources. Each example was carefully annotated with natural language instructions, and detailed setup configurations, and included manually crafted evaluation scripts to ensure the task accurately represents real-world functionalities. To validate the realism and feasibility of these tasks, every task was cross-verified by at least two other authors. In particular, these reviewers checked the instructions for clarity, the setup for correctness, and the evaluation scripts for comprehensiveness. Additionally, some tasks inherently deemed infeasible due to outdated features or user misconceptions were also included to represent the full spectrum of real-world challenges.

- **What mechanisms or procedures were used to collect the data (*e.g.*, hardware apparatuses or sensors, manual human curation, software programs, software APIs)?** How were these mechanisms or procedures validated?

Data collection for the OSWORLD benchmark involved a combination of manual human curation and usage of software programs. Specifically, task instructions and scenarios were gathered from diverse sources including official guidelines, online tutorials, how-to websites, public forums, and personal blogs. These were then annotated manually by the authors and supplemented with setup actions and evaluation scripts using software tools designed for task setup and verification.

The validation of these mechanisms and procedures was conducted through a rigorous peer-review process within the research team. Each task and its associated data were cross-checked by multiple authors to ensure feasibility, clarity, and alignment with the baseline sources. Additionally, the examples gathered were tested by independent evaluators not involved in the initial annotation, acting as agents to execute the tasks, and providing feedback which was used to refine the tasks and annotations further.

- **If the dataset is a sample from a larger set, what was the sampling strategy (*e.g.*, deterministic, probabilistic with specific sampling probabilities)?**

The tasks in the OSWORLD represent a carefully curated subset of potential real-world computing tasks. The selection was made based on a combination strategy that included both deterministic and probabilistic elements. We chose examples based on their popularity, helpfulness, and diversity, with popularity assessed by view counts and votes on various platforms such as forums, tutorials, and online courses. This involved deterministic selection of application types to ensure coverage across a broad spectrum of daily, professional, and workflow scenarios. Additionally, we incorporated tasks that were inherently challenging or infeasible due to software limitations or feature deprecations, enhancing the dataset's practical relevance and complexity.

- **Who was involved in the data collection process (*e.g.*, students, crowdworkers, contractors) and how were they compensated (*e.g.*, how much were crowdworkers paid)?**

All the development of platform, app data collection, and task definition creation are completed by the authors.

- **Over what timeframe was the data collected?** Does this timeframe match the creation timeframe of the data associated with the instances (*e.g.*, recent crawl of old news articles)? If not, please describe the timeframe in which the data associated with the instances was created.

- **Did you collect the data from the individuals in question directly, or obtain it via third parties or other sources (e.g., websites)?**

The data for OSWORLD were primarily collected via third parties and other sources. This included a diverse array of platforms such as official guidelines and tutorials, video tutorials on TikTok and YouTube, how-to websites like WikiHow, Q&A forums such as Reddit, Quora, Superuser, and StackOverflow, formal online courses from Coursera and Udemy, and publicly available personal blogs and guidelines.

- **Were the individuals in question notified about the data collection?** If so, please describe (or show with screenshots or other information) how notice was provided, and provide a link or other access point to, or otherwise reproduce, the exact language of the notification itself.

No, the nature of the data collected for the OSWORLD benchmark did not involve any individual-based data that would require the notification of individuals. The tasks and environment setups used in OSWORLD were sourced from publicly available materials such as forums, tutorials, and guidelines, which do not contain personally identifiable information. Thus, notification to individuals was not applicable in this context.

- **Did the individuals in question consent to the collection and use of their data?** If so, please describe (or show with screenshots or other information) how consent was requested and provided, and provide a link or other access point to, or otherwise reproduce, the exact language to which the individuals consented.

The data collected for the OSWORLD includes tasks sourced from publicly accessible forums, tutorials, and community-driven platforms, which inherently involve contributions from public individuals. However, the paper does not specify that these individuals provided direct consent for the use of such data in the context of creating the benchmark. Typically, these public platforms have their own terms of service that users agree to, which may cover data usage for research, but exact consent for this specific study was not detailed. Hence, more explicit consent mechanisms would potentially enhance ethical compliance.

- **If consent was obtained, were the consenting individuals provided with a mechanism to revoke their consent in the future or for certain uses?** If so, please provide a description, as well as a link or other access point to the mechanism (if appropriate).

As the data collection involved publicly available sources and no specific documentation of direct consent is provided, there is also no mention of a mechanism for the individuals to revoke their consent in the future or for certain uses of the data. Implementing a consent revocation mechanism would be an important step towards enhancing the ethical handling of the data used in OSWORLD.

## G.4 Uses

- **Has the dataset been used for any tasks already?** If so, please provide a description.

Yes, the OSWORLD dataset has been utilized to support the development and evaluation of automated computer agents across various operating systems. It includes a set of 369 tasks on Ubuntu and 43 tasks on Windows, which are designed to assess various capabilities such as commonsense reasoning, software navigation, and precise input control. These tasks have been employed in comprehensive testing and performance evaluation scenarios, which have involved comparing different LLM and VLM agents' abilities to handle multi-app workflows and other operation types within real computing environments. Consequently, the dataset has facilitated research into creating more effective and versatile digital agents. Detailed results and analyses stemming from these uses are documented in the paper.

- **Is there a repository that links to any or all papers or systems that use the dataset?** If so, please provide a link or other access point.

  Yes, the OSWORLD dataset is accompanied by a repository that includes links to all papers and systems utilizing the dataset for research and development purposes. It serves as a comprehensive resource for tracing the impact and applications of the dataset across various studies and implementations. The repository can be accessed via `https://github.com/xlang-ai/OSWorld`.

- **Is there anything about the composition of the dataset or the way it was collected and prepro-cessed/cleaned/labeled that might impact future uses?** For example, is there anything that a dataset consumer might need to know to avoid uses that could result in unfair treatment of individuals or groups (*e.g.*, stereotyping, quality of service issues) or other risks or harms (*e.g.*, legal risks, financial harms)? If so, please provide a description. Is there anything a dataset consumer could do to mitigate these risks or harms?

  No.

## G.5  Distribution

- **Will the dataset be distributed to third parties outside of the entity (*e.g.*, company, institution, organization) on behalf of which the dataset was created?** If so, please provide a description.

  Yes, the dataset created as part of the OSWORLD project is made available publicly in `https://github.com/xlang-ai/OSWorld`. It is open-sourced under the Apache-2.0 license, and it can be accessed via the official project URLs provided on GitHub and the OSWORLD website. This facilitates wide accessibility and utilization across different sectors and by various entities beyond the originating organization, promoting broader research and development in digital agent technology.

- **How will the dataset will be distributed (*e.g.*, tarball on website, API, GitHub)?** Does the dataset have a digital object identifier (DOI)?

  The platform and dataset are open-sourced at GitHub. We do not apply for a DOI.

- **When will the dataset be distributed?**

  Both the environment and the task set have already been made public.

- **Will the dataset be distributed under a copyright or other intellectual property (IP) license, and/or under applicable terms of use (ToU)?** If so, please describe this license and/or ToU, and provide a link or other access point to, or otherwise reproduce, any relevant licensing terms or ToU, as well as any fees associated with these restrictions.

  Both the environment and dataset are open-sourced under Apache-2.0 license.

- **Have any third parties imposed IP-based or other restrictions on the data associated with the instances?** If so, please describe these restrictions, and provide a link or other access point to, or otherwise reproduce, any relevant licensing terms, as well as any fees associated with these restrictions.

  No third-party IP-based or other restrictions are imposed on the primary data utilized by the OS-WORLD benchmark. The components comprising the tasks are derived from widely accessible and openly available sources, such as forums, tutorials, and free-to-use websites. All efforts have been made to ensure the tasks do not infringe upon proprietary data. However, the platform for virtual machines and tasks adapted for Windows require activation due to copyright considerations post-implementation, which can involve associated fees, although optional. For more detailed licensing terms and specific conditions, refer to the original sources and legal advice where relevant.

- **Do any export controls or other regulatory restrictions apply to the dataset or to individual instances?** If so, please describe these restrictions, and provide a link or other access point to, or otherwise reproduce, any supporting documentation.

  No.

## G.6  Maintenance

- **Who will be supporting/hosting/maintaining the dataset?**

  The authors will support, host, and maintain permanently.

- **How can the owner/curator/manager of the dataset be contacted (*e.g.*, email address)?**
  Issues and discussions on GitHub and Hugging Face are welcome. One can also seek help from Tianbao Xie (`tbxie@cs.hku.hk`), Danyang Zhang (`zhang-dy20@sjtu.edu.cn`), and Tao Yu (`taoyds@hku.hk`).

- **Is there an erratum?** If so, please provide a link or other access point.
  Currently, no. Errata will be announced if there is any in the future.

- **Will the dataset be updated (*e.g.*, to correct labeling errors, add new instances, delete instances)?**
  If so, please describe how often, by whom, and how updates will be communicated to dataset consumers (*e.g.*, mailing list, GitHub)?
  Yes, the OSWORLD dataset will be periodically updated to correct any labeling errors, add new instances, and delete outdated or incorrect instances as necessary. The updates will be conducted annually by the core team of maintainers comprising computer science researchers and student contributors. Notice of updates, including details of changes made, will be communicated to dataset consumers through a dedicated mailing list and updates will be posted on the project's GitHub repository at `https://github.com/xlang-ai/OSWorld`.

- **If the dataset relates to people, are there applicable limits on the retention of the data associated with the instances (*e.g.*, were the individuals in question told that their data would be retained for a fixed period of time and then deleted)?** If so, please describe these limits and explain how they will be enforced.
  No.

- **Will older versions of the dataset continue to be supported/hosted/maintained?** If so, please describe how. If not, please describe how its obsolescence will be communicated to dataset consumers.
  Older versions of the OSWORLD dataset will be available for use, but they may not receive updates or support once a new version is released. We aim to keep archival versions available for research reproducibility and comparison purposes. However, these archival versions will be hosted without active maintenance, meaning that issues or bugs identified in these versions will only be addressed in new releases. Obsolescence of any version will be communicated through our official website and version-release notes associated with the OSWORLD dataset.

- **If others want to extend/augment/build on/contribute to the dataset, is there a mechanism for them to do so?** If so, please provide a description. Will these contributions be validated/verified? If so, please describe how. If not, why not? Is there a process for communicating/distributing these contributions to dataset consumers? If so, please provide a description.
  Yes, the OSWORLD framework encourages the academic and developer community to extend, augment, and contribute to the dataset. Contributions can be made via GitHub, where the project and its datasets are hosted. Contributors can submit pull requests with their enhancements or new task datasets. All contributions undergo a rigorous review process by the existing project maintainers to ensure that they meet the quality standards of the dataset and are consistent with the goals of OSWORLD. This review process includes verifying the accuracy of the information, the relevance, and the usability of the tasks. Once approved and merged, contributions are incorporated into the main branch of the dataset repository and are made available for immediate access and use. Updates to the dataset are regularly communicated through the project's website and GitHub repository, as well as through mailing lists and social media to ensure that dataset consumers and contributors stay informed. Additionally, periodic releases of the dataset include detailed changelogs that summarize new additions, changes, and improvements, thus ensuring transparency and ease of access for all users interested in building upon or utilizing the OSWORLD dataset.

