# OpenReview forum: "OSWorld: Benchmarking Multimodal Agents for Open-Ended Tasks in Real Computer Environments"
_NeurIPS.cc/2024/Datasets_and_Benchmarks_Track — NeurIPS 2024 Track Datasets and Benchmarks Poster_

### Official Review · Reviewer_oXY7 · 2024-07-10
**Review for OSWorld**

**Rating:** 7
**Confidence:** 3

**Review:**

This paper introduces a framework and a benchmark to evaluate multimodal models in an agentic setting to see if they can perform daily tasks using various software, ranging from terminal-based to GUI-based applications like browsers and spreadsheets. It includes a virtual machine-based system capable of defining various tasks and evaluating them by executing the model in a safe environment. Additionally, it comes with snapshotting features, which can be important for reproduction purposes and setting the initial condition of the application’s state. Overall, this framework makes it a suitable environment to test agents.

Given the current trends and research direction in multimodal models, especially incorporating them as agents, this paper could be an important contribution to the community. The framework, rather than the benchmark itself, offers substantial benefits. It can be incorporated into various scenarios for different applications and measure the performance of different agent-based systems to determine if they can perform certain tasks in any arbitrary application running in an operating system.

The benchmark also includes over 369 diverse tasks spanning various categories, from simple OS-level commands to more complex workflows. Assembling this dataset required significant human labor, with the authors themselves responsible for collecting the data.

I think the benchmark conducted here is mixing too two things at the same time, which might lead to misleading results ("Knowledge" vs "Execution"). For example, consider a task like clearing cookies in a browser. Most language models can already provide step-by-step instructions for this task, indicating they possess knowledge about different browsers, the location of cookies in the browser, and the relevant menu options and steps to remove them. However, the benchmark fails to distinguish between the model’s existing knowledge of a task and its ability to execute the procedure. I think this benchmark lacks another baseline; in addition to the human baseline already provided, we need a baseline where humans act as executors. These models can generate many step-by-step plans, but they often fail to execute actions correctly on the screen due to limitations in visual grounding, even when we use set-of-marks.


When we run a benchmark and one of the conclusions is:

> LLMs and VLMs are still far from being digital agents on real computers.

It raises many questions about the model’s capabilities, which might not be accurate. The experiment conducted is not thorough enough to draw such conclusions. While it is true in the case of this benchmark setup (forcing the model to use `pyautogui`), it might not be true in general due to various reasons. The Set-of-Mark prompting and the A11y tree method used in this paper might not be suitable because they are too noisy and distracting. We might achieve much better results by simplifying those methods (removing distractions). Since we don’t have a human (as an executor) baseline, it is hard to conclude that these models are not capable of performing such tasks.

**Strengths:**

- In my opinion, the provided framework and the data is valuable and even outweighs the benchmark and results.
- Having the multi-app workflow category is interesting since it requires expertise and control over more than one application.

**Additional Feedback:**

This study is a good step in the right direction toward employing vision-based LLMs for general computer tasks. However, I have some concerns about the claims made in the paper regarding the model’s capabilities. I believe the benchmark does not measure what actually matters for the problem.

**Clarity:**

The presentation of the paper and writing can be improved, for example generally it is not a good idea to write "see appendix for more details" without a hyperlink (Line 277).

Also please see my comments in  _Opportunities For Improvement_ section

**Correctness:**

Yes. Please also see my comments in _Review_ section regarding "Knowledge" vs "Execution"

**Documentation:**

Yes, The paper comes with a website, dataset explorer, and detailed documentation.

**Ethics:**

No. The paper is good.

**Limitations:**

See my comments above.

**Opportunities For Improvement:**

Some comments about task formulation and clarify to the paper:

- This paper uses a POMDP formulation for task definitions, which consists of \(S\), \(O\), \(A\), \(T\), \(R\) (State-Observation-Action-Task-Reward). I'm not currently sure why the instruction is part of the observation. Does it mean that the instruction is not stationary and can change based on the observation?
- Also, what is the role of the reward? In traditional reinforcement learning, the reward signal is a way to guide the agent to know when it has completed the task, even if it is very sparse. It serves as a learning signal, and we do credit assignment on different steps in a rollout, updating model parameters through backpropagation. I’m not sure about the reward in this case. Do we use the reward for in-context learning? I am pretty sure you did not fine-tune GPT-4 or use the reward to fine-tune any of these models. Also, do we have one rollout per experiment?

**Relation To Prior Work:**

I think it is okay.

**Summary And Contributions:**

This paper presents a framework and benchmark for evaluating multimodal models in an agentic setting, assessing their ability to perform daily tasks across various software. It features a virtual machine-based system for task definition and evaluation. This framework provides an ideal environment for testing agents.

The benchmark introduced in the paper consists of 369+43 diverse tasks spanning five different categories: OS level, daily, office, professional, and workflow. Each task is carefully sourced by humans, accompanied by detailed instructions and additional annotations.

---

> ### Author Rebuttal · Authors · 2024-08-17
>
> Thank you for your kind feedback and constructive comments. We appreciate your recognition of our framework's value and the potential importance of our contribution to the community, especially in the context of current trends in multimodal models and agent-based systems.
>
> We have revised our manuscript and are addressing the specific questions in the following reply, including:
>
> - Clarification on the distinction between "Knowledge" and "Execution" in our benchmark
> - Results of additional experiments comparing "Human plan, model act" and "Model plan, human act"
> - Explanation of the POMDP formulation and the role of instructions and rewards
> - Improvements in paper presentation, including proper hyperlinking
>
> We have also added these new results and clarifications to the paper to provide a more comprehensive analysis. Please let us know if you have any further questions and we can provide any additional clarifications to help finalize your assessment and rating of our paper.
>
> ---
>
> ### **W1: I think the benchmark conducted here is mixing too two things at the same time, which might lead to misleading results ("Knowledge" vs "Execution")**
> **A:** Thanks for pointing this out. A recent work, SeeAct[1], conducted an experiment similar to what you described, where the model gives natural language instructions for each step, and humans complete the specific execution actions (examining Knowledge). This showed that human execution can help the model. Conversely, in a recent follow-up work to OSWorld, Spider2-V[2], the authors had humans provide natural language descriptions of specific actions to be taken in advance, like clear documentation. The model then executed actions according to this doc (examining the model's Execution ability, some researchers also name it “Grounding” ability). Results show that the help is little due to the weak ability in execution.
> Our OSWorld actually aims to examine both of these abilities simultaneously. While there has been considerable work on examining model knowledge or specific instruction execution[3, 4], constructing entire environments is more challenging and thus less explored. However, we believe that environment-based approaches can generate more fine-grained benchmarks to test different types of abilities in the future.
>
> To make our paper more comprehensive and provide more insights, we conducted tests on the OSWorld subset with similar experimental settings. The results of these tests are presented in the following table, which has been incorporated into the analysis section of our paper:
>
> | Condition | Success Rate |
> |-----------|--------------|
> | Model Only | 17.95% |
> | Human Knowledge | 20.51% |
> | Human Execution | 35.90% |
>
> The results indicate that, in essence, the model performs poorly in both knowledge and execution abilities when it comes to using software in real-world scenarios, with execution ability being particularly weak. On the execution side, we found that even with very detailed step-by-step language descriptions, the model still fails to correctly execute the actions corresponding to human descriptions. This suggests that grounded models need significant improvement. On the knowledge side, we discovered that the model can understand and describe quite a number of tasks well, responding according to specific situations. However, it still performs poorly in error correction and some finer-grained software knowledge, resulting in a success rate that remains below 50%.
>
> [1]. Zheng, B., Gou, B., Kil, J., Sun, H., & Su, Y. (2024). Gpt-4v (ision) is a generalist web agent, if grounded. arXiv preprint arXiv:2401.01614.
>
> [2]. Cao, R., Lei, F., Wu, H., Chen, J., Fu, Y., Gao, H., ... & Yu, T. (2024). Spider2-V: How Far Are Multimodal Agents From Automating Data Science and Engineering Workflows?. arXiv preprint arXiv:2407.10956.
>
> [3]. Hendrycks, D., Burns, C., Basart, S., Zou, A., Mazeika, M., Song, D., & Steinhardt, J. (2020). Measuring massive multitask language understanding. arXiv preprint arXiv:2009.03300.
>
> [4]. Cheng, K., Sun, Q., Chu, Y., Xu, F., Li, Y., Zhang, J., & Wu, Z. (2024). Seeclick: Harnessing gui grounding for advanced visual gui agents. arXiv preprint arXiv:2401.10935.
>
> ---
>
> ### **W2: Proper hyperlinking in paper**
> **A:** Thanks for pointing that out! We have made changes to address this in our latest manuscript.
>
> ---
>
> ### **Q1: Instruction in POMDP task formulation**
> **A:** Thank you for the question! In our work, we define instructions as part of the state. This approach considers that in real-world scenarios, users might modify, reduce, or add to the instructions while the agent is running in every iteration of the POMDP, even though our benchmark data items don't examine these dynamic situations. We have defined this aspect and left it for future work to explore.
>
> ---
>
> ### **Q2: Role of reward and rollout per task**
> **A:** Thank you for your question. In our work, we did not use rewards as signals to improve model training. Instead, we only used LLMs/VLMs as agents under a prompt-based method. Additionally, in our annotations, we did not shape dense rewards, so the rewards are relatively sparse. However, our primary focus is on evaluating the capabilities of existing large language models and vision-language models in performing real-world computer tasks. For future improvements in this area, writing more dense rewards, exploring the effectiveness of rewards in desktop environments, and corresponding algorithms could potentially be used to train RL policies. We have left this for future work.

---

> > ### Comment · Reviewer_oXY7 · 2024-08-23
> > **POMDP task formulation**
> >
> > > In our work, we define instructions as part of the state
> >
> > Why? Your task definition is not sound and hurts your work more than it helps. I encourage you to look at some papers related to multi-task or goal-conditioned reinforcement learning. They have concepts like Goal-Augmented MDP that include additional tuples to specify the goal.

---

> > > ### Author Response · Authors · 2024-08-25
> > >
> > > Dear Reviewer,
> > >
> > > Thanks for the advice!
> > > We understand the method of representing objectives using specialized tuples, but this introduces complexity in representation. Therefore, in the paper, we considered it as part of the state. We did not realize what specific damage or cost this modeling approach might entail. It would be helpful to have more specific references that address this issue, so that we can address your concern correspondingly.
> > >
> > > Best,
> > > Authors

---

> > > > ### Comment · Reviewer_oXY7 · 2024-08-26
> > > >
> > > > I believe I have already expressed that I like this work and consider the contribution valuable. However, my concerns are about the presentation and how concepts are formulated in this paper. If this paper gets accepted (which it likely will), it might set a bad example for other researchers to use loosely defined tasks or, to some extent, engage in notation abuse.
> > > >
> > > > The question is, why use a POMDP formulation for task definition? What are the authors trying to solve here that requires a POMDP formulation? I’m not even sure why we need to define this type of agent in a partially observable setting rather than a fully observable one. Shouldn’t this type of agent have full access/visibility to the entire OS so they can assist us better? An instruction clearly does not belong to the "state", but you can define it as a function over "state", as the "state" is information that an agent has about the environment at a given time.  So the question is, what was the goal behind this task formulation, and how does it impact the paper? Do we even need it in the first place?
> > > >
> > > > If you look at `Contrastive Learning as Goal-Conditioned Reinforcement Learning` [1] or any other relevant paper, you’ll see that goals are defined as a distribution over states. You could probably do something similar, but of course, the task definition and the actual benchmark seem likely to be two separate things in this paper.
> > > > You can also take a look at this paper: `Goal-Conditioned Reinforcement Learning: Problems and Solutions` [2]
> > > >
> > > > [1] Eysenbach, Benjamin, et al. "Contrastive learning as goal-conditioned reinforcement learning." Advances in Neural Information Processing Systems 35 (2022): 35603-35620.
> > > > [2] Liu, Minghuan, Menghui Zhu, and Weinan Zhang. "Goal-conditioned reinforcement learning: Problems and solutions." arXiv preprint arXiv:2201.08299 (2022).

---

> > > > > ### Author Response · Authors · 2024-08-26
> > > > >
> > > > > Dear Reviewer,
> > > > >
> > > > > Thank you for your continued engagement and valuable references. We appreciate your concerns about our task formulation and presentation.
> > > > >
> > > > > We chose a POMDP formulation to capture the complexity of real-world computer interactions because many system states are not directly observable, even through scripts or APIs. For instance:
> > > > >
> > > > > 1. The internal state of applications (e.g., unsaved changes, undo history).
> > > > > 2. The state of external services that the local system depends on.
> > > > > 3. Temporary network disruptions or latency spikes.
> > > > > 4. Hardware-level states that affect performance but are not exposed to the OS.
> > > > >
> > > > > Additionally, the agent may encounter unexpected elements such as pop-up notifications or advertisements, further complicating the observability.
> > > > >
> > > > > However, we acknowledge that our formulation could be improved. After reviewing the suggested papers, we agree that a goal-conditioned approach might be more appropriate. We propose reformulating our problem as a Goal-Augmented POMDP with the tuple <S, O, A, T, Ω, r, γ, ρ0, G, pg, φ>, where:
> > > > >
> > > > > - S is the full state space (including hidden system states)
> > > > > - O is the observation space (what's visible or accessible to the agent)
> > > > > - G is the space of goals (instructions in our case)
> > > > > - pg is the distribution of desired goals (instructions)
> > > > > - φ : O → G is a mapping function from observations to goals
> > > > >
> > > > > This reformulation maintains the partial observability aspect while aligning better with goal-conditioned reinforcement learning literature.
> > > > >
> > > > > We commit to updating our task formulation in the camera-ready version to reflect this Goal-Augmented POMDP approach as well as the citations for related-work. This change will enhance the clarity and theoretical soundness of our work without altering our main findings.
> > > > >
> > > > > Thank you again for your valuable input in helping us improve our paper.
> > > > >
> > > > > Best regards,
> > > > > The Authors

---

> > > > > > ### Comment · Reviewer_oXY7 · 2024-08-27
> > > > > >
> > > > > > I thank the authors for their understanding and responsibility. I have revised my original rating.

---

### Official Review · Reviewer_bfsS · 2024-07-14
**A great contribution to open-ended multimodal agent benchmarks**

**Rating:** 9
**Confidence:** 4
**Clarity:** The paper is mostly well written.

**Review:**

This paper is a highly engaging read, with much of its valuable content extending beyond the main text into an extensive 34-page appendix, which provides a wealth of crucial details. These comprehensive explanations not only enhance the transparency and reproducibility of the current research but also serve as an excellent resource for researchers looking to conduct similar studies or build upon this work. The authors' commitment to providing such thorough documentation demonstrates the depth of their work and significantly contributes to the paper's overall value and impact in the field of multimodal AI agents for computer interaction.

See strengths and weaknesses in later comments.

**Strengths:**

1. Novel contribution: OSWORLD is presented as the first scalable real computer environment for evaluating multimodal agents across multiple OS platforms.
2. Comprehensive benchmark: The authors create a diverse benchmark of 369 desktop computer tasks covering multiple domains and applications.
3. Realistic scenarios: The tasks are based on real-world use cases and include detailed setup and execution-based evaluation for reproducibility.
4. Cross-platform support: The environment supports task setup, interactive learning, and execution-based evaluation across Ubuntu, Windows, and macOS.
5. Detailed description of environment and benchmark: The paper provides comprehensive description of the environment and the benchmark in the appendix, which is very useful for understanding and using the dataset.
6. Open-source: The implementation and experiments are made available for further research and development.
7. Complement input formats: The benchmark considers different input formats like screenshot and accessibility tree, which is a complement to each other.

**Additional Feedback:**

1. LLM and VLM agent baselines: In the baseline, the authors only provide the 3 most recent observation-action pairs to the models. This seems to limit the context of the models. Is there any reason of setting this limitation? Could you improve over this?

**Correctness:**

The authors provide detailed information about benchmark construction and evaluation, which seems correct. However, since the authors refrain from developing the benchmark on macOS to avoid copyright issues, I am not sure to which extent the claim of supporting macOS still holds.

**Documentation:**

Yes.

**Ethics:**

No ethical concerns identified.

**Limitations:**

1. Limited statistical robustness: Due to budget constraints with advanced models like GPT-4V, the experiments lack error bars. This omission potentially impacts the fairness and reliability of comparisons between models with close performance, as the statistical significance of differences cannot be established.
2. Benchmark scalability challenges: While the benchmark covers a wide range of tasks, real-life workloads evolve over time and vary significantly across different domains. Although the authors provide a standard environment interface and specifications, extending the benchmark to include new types of tasks or applications appears non-trivial. This limitation may affect the long-term relevance and comprehensiveness of the benchmark as computer use patterns change.
3. Limited scope: The authors refrain from developing the benchmark on macOS to avoid copyright issues.

**Opportunities For Improvement:**

1. Multi-agent scenarios: Currently there is a trend of solving a task with a team of collaborative agents, some of them could be specialized with different tasks. It would be interesting to see whether multi-agent systems outperform single-agent systems in the paper.
2. Details about human evaluation: Maybe I miss something, in the paper it only mentions that annotators being computer science major college students, without further details like how many they are, how many human annotators per task etc.
3. Accessibility improvements: Given the noted variance in accessibility tree quality across applications, the paper could explore more methods to standardize or improve the quality of accessibility information provided to agents.
4. Long-term interaction studies: It seems like most tasks in the system can be done in relatively short period of time. Incorporating tasks that require long-term memory or learning over multiple sessions could provide insights into how agents perform in more realistic, ongoing use scenarios.
5. User study expansion: Conducting a larger-scale user study with a more diverse group of participants (instead of students only) could provide more robust human performance baselines and insights into task difficulty across different user groups.
6. Framework for contribution: It would be useful to write some general guidance for external contributors to contribute new tasks easily.

**Relation To Prior Work:**

Yes.

**Summary And Contributions:**

This work introduces OSWorld, a scalable realistic computer environment designed to evaluate multimodal agents. With OSWorld, the authors craft a comprehensive benchmark consisting of 369 desktop computer tasks spanning various domains. The authors further evaluate AI agents with the benchmark and reveals a substantial capability gap between human users and current state-of-the-art AI systems in completing complex computer tasks.

---

> ### Author Rebuttal · Authors · 2024-08-17
>
> Thank you for your appreciation and detailed evaluation of our work. We are glad you think our paper is a highly engaging read with valuable content extending into the extensive appendix, providing crucial details that enhance transparency and reproducibility. We appreciate your recognition of OSWorld as a novel contribution, offering a comprehensive and realistic benchmark for evaluating multimodal agents across multiple OS platforms. We are pleased that you noted the strengths of our work, including the diverse task set, cross-platform support, and open-source implementation.
> In our response, we have addressed your questions and suggestions regarding our work. Please let us know if you have any further questions and we can provide any additional clarifications to help finalize your assessment and rating of our paper.
>
> ---
>
> ### **W1: Multi-agent scenarios**
> **A:** Thanks for your suggestion! We believe that multi-agent systems could be very helpful in building better performing systems and creating more realistic tasks. However, this might be beyond the scope of our current paper. We'd like to leave this for future work!
>
> ---
>
> ### **W2: Details about human evaluation**
> **A:** Thanks for the question! Actually, we have detailed these annotations in section 3.2 of our paper. To summarize, the annotation process involved 9 computer science students working for about 1800 man-hours over 3 months for the initial task creation and double-checking. An additional 400 man-hours were spent on collecting examples and designing initial states and evaluations. Furthermore, we dedicated over 400 man-hours to four rounds of checks and fixes during experiments for human performance and baselines. In total, the entire process consumed more than 2600 man-hours of work. Each example underwent at least four rounds of checks to ensure quality and accuracy.
>
> ---
>
> ### **W3: Accessibility improvements**
> **A:** Thank you for pointing that out. In the latest release of OSWorld, we have published all the [code](https://github.com/xlang-ai/OSWorld/commit/a961d2276de8996586988871da07e7de0c4d3d9a) necessary for acquiring accessibility (a11y) trees across Ubuntu, Windows, and macOS. We've also unified the agent's ability to operate based on these observations. We hope this contribution will help advance research in this direction.
>
> ---
>
> ### **W4: Long-term interaction studies**
> **A:** Thanks for the question. In our statistics, tasks require an average of about 15 steps to complete, which exceeds the 3.6 steps in MiniWoB++ and 7.3 steps in Mind2Web. Some tasks even require up to 200 steps to complete. In terms of time, OSWorld has a median completion time of 111.94 seconds compared to 35.38 seconds in WebArena benchmark, with some tasks taking over 1200 seconds to complete. OSWorld also has a lower accuracy rate of 72.36% versus 88% for pure web tasks in WebArena. We believe that OSWorld is already sufficiently challenging, even when considering long-distance interactions.
>
> ---
>
> ### **W5: User study expansion and framework for contribution**
> **A:** Thank you for pointing that out! We have prepared detailed [documentation](https://timothyxxx.github.io/OSWorld/) and a [Discord channel](https://discord.gg/4Gnw7eTEZR) for the community, providing guidance on adding new tasks for those who wish to use the OSWorld environment. We're leaving the collection of more human metrics and additional tasks for future work!
>
> ---
>
> ### **W7: Limited statistical robustness**
> **A:** Like other challenging benchmarks such as WebArena and SWE-bench, running an agent based on GPT-4 or Claude-3 is prohibitively expensive, often costing thousands of dollars. In our preliminary experiments, we didn't observe significant fluctuations in the results, so we didn't run this test in the full experiment. We hope to promote progress in open-source and cost-effective high-performance models in the future. At that point, we can conduct more comprehensive research, including statistical robustness studies from multiple perspectives.
>
> ---
>
> ### **W8: Benchmark scalability challenges**
> **A:** Thank you for pointing that out! We indeed want to emphasize our advantages in extending to new tasks. To facilitate this, we have prepared detailed documentation and established a Discord channel for the community. These resources provide comprehensive guidance on adding new tasks for those who wish to use the OSWorld environment. We'd also like to highlight that there are no time limitations restricting the types of tasks that can be implemented, which significantly broadens the scope of potential applications.
>
> ---
>
> ### **W9: Expanding OS Coverage**
> **A:** Thank you for your insightful observation. Our initial release focused on Ubuntu and Windows to ensure robust functionality and comprehensive task coverage across widely used operating systems. We recognize the importance of macOS in the computing landscape and are actively working on extending OSWorld's capabilities to include macOS support. Our team is exploring ways to incorporate macOS tasks while respecting intellectual property considerations. We welcome collaboration from the research community to help us achieve this goal and create an even more inclusive and versatile benchmark for evaluating multimodal agents.

---

> > ### Comment · Reviewer_bfsS · 2024-08-25
> >
> > Thanks!
> >
> > You mentioned some tasks even require up to 200 steps to complete. But in the LLM and VLM agent baselines, it seems you only provide the 3 most recent observation-action pairs to the models. This seems to limit the context of the models. Is there any reason of setting this limitation? Could you improve over this?

---

> > > ### Author Response · Authors · 2024-08-26
> > >
> > > Thank you for your question.
> > >
> > > The choice of providing the 3 most recent observation-action pairs to the models was primarily a balance between the maximum context length supported by the models and cost considerations. The input for each step of the agent is quite lengthy - a 1080p screenshot averages around 1000 tokens, while an accessibility (a11y) tree can reach up to 2000 tokens on average.
> > >
> > > Considering the current maximum context length commonly supported by models and our budget constraints, inputting the past 3 rounds was found to be a balanced setting in our experiments.
> > >
> > > We analyze the potential improvements that could be brought by increasing the observation-action history in the second paragraph of Section 5. As models' context input support length and comprehension capabilities improve, performance is likely to further increase.

---

### Official Review · Reviewer_AeF1 · 2024-07-25
**A frontier Benchmark for Human-Computer Interaction**

**Rating:** 6
**Confidence:** 4
**Correctness:** Mostly correct. I point out some blur…
**Clarity:** Yes

**Review:**

I think the paper benchmarks a very challenging and also serious question in the field of Human-Computer Interaction, that whether AI could replace/help human to conduct various tasks in PC. The execution-based evaluation also provides more trustworthy results than simple QA based evaluation. For pros and cons please refer to the Strengths and Opportunities section.

**Strengths:**

1. The integrity of the benchmark, OSWorld is the first benchmark which provides real environment and execution-based evaluation is OS level tasks.
2. The paper gives detailed experiment and analysis on current sota open/close LLMs and VLMs. The results indicate that OSWorld is a challanging task.

**Additional Feedback:**

NA

**Documentation:**

Yes

**Limitations:**

yes

**Opportunities For Improvement:**

1. It's hard for me to understand how some executions are regarded as success, such as “Could you make the background
of this image transparent for me?”， I think the author should explain more about it in the main content (section 2.1) since it is the main metric of the proposed benchmark. How do you make sure that the assesment is acurate to avoid bias judement in the reward as many targets in the appendix seem blurred to me. For example, how to define "background" in this case.
2. It's not as expected that LLM without vision ability actually gets the highest score in the multimodal benchmark (Table4), can you explain the results?
3. I think the analysis of different tasks is distracted, given the low overall scores for all models. It's recommended to give some detailed case analysis for each task.

**Relation To Prior Work:**

I think many embodied benchmarks for multimodal agents are missing from the related work part: such as Minedojo, Alfred, PCA-Bench, EgoThink. OSWorld is set in an embodied environments, which should be reflected and compared in the related work part.

**Summary And Contributions:**

The paper presents OS-World, the first real computer environment designed for developing multimodal agents. It supports task setup, interactive learning, and execution-based evaluation across various mainstream operating systems like Ubuntu, Windows, and macOS. This environment allows for the development of agents that can perform a wide range of real computer tasks beyond isolated interfaces and applications. The paper also evaluates different multimodal LLMs and find that there is still a large performance gap on these models compared to human.

---

> ### Author Rebuttal · Authors · 2024-08-17
>
> Thank you for your thoughtful review and appreciation of our work! We're pleased that you recognize OSWorld as the first benchmark providing a real environment and execution-based evaluation for OS-level tasks. We appreciate your acknowledgment of the integrity of our benchmark and the detailed experiments and analysis on current state-of-the-art open and closed LLMs and VLMs.
>
> We have revised our manuscript and are addressing the specific questions in the following reply, including:
>
> - Clarifications on execution-based evaluation details
> - Discussion on the unexpected performance of LLMs without vision ability
> - Inclusion of additional embodied benchmarks in the related work section
>
> Please let us know if you have any further questions and we can provide any additional clarifications to help finalize your assessment and rating of our paper.
>
> ---
>
> ### **W1: Unclear success criteria for certain tasks, e.g., "Make the background transparent". More explanation needed in section 2.1 on how success is determined and "background" is defined. How is assessment accuracy ensured to avoid bias, given that many targets in the appendix appear blurred?**
> **A:** Thank you for your suggestions. We have added more descriptions in section 2.1 to provide clearer details on how we conduct our evaluations. Regarding the "Make the background transparent" example, we actually compare the image obtained after the agent completes the action with the correct image (manually created by removing the background) by calculating the structural similarity (based on the `structural_similarity` function in `skimage.metrics` in this case) and determining if it reaches a threshold. We have carefully designed and implemented different parameters for various benchmark samples, prepared for multiple possibilities, and completed red team-blue team testing to ensure that each sample has been attempted and verified by four annotators. This process aims to eliminate bias as much as possible.
>
> ---
> ### **W2: It's not as expected that LLM without vision ability actually gets the highest score in the multimodal benchmark (Table 4), can you explain the results?**
> **A:** Thank you for your question. In the limit, we do expect visual models to perform better than text models. However, we must acknowledge that current visual models have certain limitations especially when acting act computer agents.
>
> 1. **Visual information can be distracting.**
> We have tried several strategies to check the robustness of visual information, such as changing the window's position, changing the window's size to the minimum, opening some irrelevant software, and maximizing them to clutter the screen. The results reported in Sec. 5 reveal that the agent performances with visual information are sensitive to these disturbances and the SRs vary fiercely. As for text information, these methods will not affect the overall structure of the a11y tree except for changing some element properties on it.
> 2. **After multimodal learning, GPT-4 may have lost some of its text reasoning capabilities, which is essential for agent tasks**
> In pure text a11y tree conditions, GPT-4 achieved state-of-the-art performance. For our settings with both image and text inputs (screenshot, screenshot + a11y tree, som), we used the GPT-4V model. Previous work[1] has also pointed out that multimodal post-training can lead to some loss in pure text processing abilities. This could potentially explain the performance differences we observed.
>
>     [1]. Lin, J., Yin, H., Ping, W., Molchanov, P., Shoeybi, M., & Han, S. (2024). Vila: On pre-training for visual language models. In Proceedings of the IEEE/CVF Conference on Computer Vision and Pattern Recognition (pp. 26689-26699).
>
> ---
>
> ### **W3: I think the analysis of different tasks is distracting, given the low overall scores for all models. It's recommended to give some detailed case analysis for each task.**
> **A:** Thank you for your valuable suggestions. Indeed, our analysis extends far beyond mere scores. In Appendix D, particularly in section D.4 Qualitative Analysis, we have conducted a comprehensive and in-depth examination of the results. Furthermore, in section D.5 Qualitative Analysis Examples, we have presented a selection of representative cases showcasing both successes and failures of large language model-based agents, along with notable error patterns we observed during our study. These detailed analyses offer substantial insights that we believe will be instrumental in guiding future research efforts in this field. Due to page length constraints, we allocated more space to the environment and benchmark construction, placing the analysis in the appendix. We plan to refine and expand this analysis further in future work.
>
> ---
>
> ### **W4: Missed citation.**
> **A:** Thank you for bringing this to our attention. Initially, we didn't include these works as we considered OSWorld to be uniquely focused on digital operating system interactions, which differs significantly from the physical embodied environments of Minedojo, Alfred, PCA-Bench, and EgoThink. However, we appreciate your suggestion and have now incorporated these relevant papers into our related work section, specifically under the discussion of similar embodied environments. This addition helps to better position OSWorld within the broader landscape of multimodal agent research and highlights its unique contributions to the field.

---

> ### Author Response · Authors · 2024-08-26
>
> Dear reviewer AeF1,
>
> As the open discussion period draws to a close in a few days, we wanted to check back to see whether you have any remaining concerns. We appreciate your recognition of OSWorld as the first benchmark providing a real environment and execution-based evaluation for OS-level tasks. We believe that we have sufficiently responded to your earlier queries, and we provide a short summary here for your convenience:
>
> 1. We have addressed your concern about unclear success criteria for certain tasks by adding more descriptions in section 2.1. We've explained our evaluation process, including the use of structural similarity metrics and the rigorous verification process involving multiple annotators to ensure accuracy and minimize bias.
>
> 2. We have provided an explanation for the unexpected performance of LLMs without vision ability in the multimodal benchmark. We discussed the potential distractions of visual information and the possible trade-offs in text reasoning capabilities after multimodal learning.
>
> 3. We have highlighted our detailed case analysis for different tasks, which is presented in Appendix D, particularly in sections D.4 Qualitative Analysis and D.5 Qualitative Analysis Examples. These sections offer in-depth examinations of both successful and failed cases, along with notable error patterns.
>
> 4. We have incorporated the suggested embodied benchmarks (Minedojo, Alfred, PCA-Bench, and EgoThink) into our related work section, better positioning OSWorld within the broader landscape of multimodal agent research.
>
> We have updated our manuscript to reflect these changes and additions. Please let us know if/how we can address any remaining concerns, and we are grateful for any additional feedback and suggestions!
>
> Best regards,
> Authors

---

### Official Review · Reviewer_bitD · 2024-08-01
**Scalable real computer environment benchmarks**

**Rating:** 5
**Confidence:** 3
**Correctness:** Yes
**Clarity:** Yes

**Review:**

The paper introduces OSWORLD, a comprehensive benchmark for evaluating computer control agents across multiple operating systems with a wide range of tasks. It excels in task coverage and detailed annotations, providing valuable insights into current models' limitations. However, it lacks comparisons with similar works like ScreenAgent, does not explore additional open source models, and provides insufficient analysis of experimental results. Additionally, scalability and adaptability issues are not fully addressed. Overall, the paper makes a significant contribution but would benefit from addressing these gaps.

There is so little exploration of open source models, why not try MiniCPM which are nice multimodal open source models. Also Cogagent is too weak to be meaningful for analysis, please try more similar open source models.

The analysis of the experimental results is not thorough enough. For example, in Table 5, we observe that in the related experiments with GPT-4V, the second method (using only screenshots) performs worse than the first method (using only the A11y tree) in several tasks. Additionally, the third method (combining both A11y tree and screenshots) does not outperform the single strategy, as seen in Daily tasks and Office tasks. Why is this the case? Moreover, the trend in the experimental results of GPT-4 is slightly different from Gemini. Gemini shows significant performance improvement when using either visual information alone or in combination with the A11y tree. Why is there such a difference? The authors need to provide a detailed analysis and explanation

**Strengths:**

Multi-OS Support: Covers Ubuntu, Windows, and macOS operating systems.

Broad Task Coverage: Includes 369 tasks involving web and desktop applications, covering file I/O and multi-application workflows.

Detailed Task Setup and Evaluation: Each task includes detailed setup and execution-based evaluation scripts to ensure reproducibility.

**Additional Feedback:**

N/A

**Documentation:**

Yes

**Limitations:**

The scalability of the OSWORLD environment across different hardware configurations and its adaptability to future advancements in operating systems and applications remain areas for further exploration. Additionally, the evaluation metrics and success criteria used may not encompass all aspects of human-computer interaction, potentially overlooking subtleties in task execution and user experience.

**Opportunities For Improvement:**

There is a similar work named ScreenAgent[1]. Why is there no comparison and analysis with this work?

There is so little exploration of open source models.

The analysis of the experimental results is not thorough enough.

[1] ScreenAgent : A Vision Language Model-driven Computer Control Agent

**Relation To Prior Work:**

OSWORLD significantly advances over previous benchmarks by addressing their limitations in scope, evaluation methods, and task diversity. Prior benchmarks often simplified tasks and restricted them to specific domains, assuming a single correct solution and missing opportunities for interactive learning. In contrast, OSWORLD supports a wide range of real-world tasks across various applications and interfaces, using execution-based evaluations that allow for multiple correct solutions, providing a more comprehensive assessment of agent capabilities.

Figure 4 illustrates the broader range of applications and more intricate interactions supported by OSWORLD compared to previous benchmarks. Human evaluations show that tasks from OSWORLD are more complex and time-consuming, with a median completion time of 111.94 seconds compared to 35.38 seconds in WebArena, and a lower accuracy rate of 72.36% versus 88% for pure web tasks. This highlights the higher level of understanding and proficiency required, setting a new standard in multimodal agent evaluation.

**Summary And Contributions:**

This paper introduces OSWORLD, a scalable real computer environment designed to address the limitations of existing benchmarks in interactivity and task diversity. OSWORLD provides free-form keyboard and mouse control across major operating systems (e.g., Ubuntu, Windows, macOS), supporting initial task state configuration and execution-based evaluation. The main contributions of this paper include: creating a benchmark with 369 tasks based on real user scenarios, covering web and desktop applications, file I/O, and multi-application workflows; meticulously annotating each task to ensure reliable and reproducible evaluation; and conducting extensive evaluations of various advanced large language models (LLMs) and vision-language models (VLMs), revealing the current agents' shortcomings in handling complex and dynamic computer tasks. Additionally, the paper highlights future improvement directions, such as enhancing agents' visual perception and contextual understanding abilities to better adapt to dynamic and non-standard interface layouts.

---

> ### Author Rebuttal · Authors · 2024-08-17
>
> Dear Reviewer bitD,
>
> Thank you for your appreciation and detailed evaluation of our work. We are glad you think our paper introduces a comprehensive benchmark for evaluating computer control agents across multiple operating systems with a wide range of tasks, excelling in task coverage and detailed annotations, and providing valuable insights into current models' limitations.
>
> We have revised our manuscript and are addressing the specific questions in the following reply, including:
> - Expanded our experiments to include additional open-source multimodal models
> - Provided a more thorough analysis of our experimental results, including detailed explanations for observed performance differences
> - Enhanced our discussion of related work, including a comparison with ScreenAgent
>
> Please let us know if you have any further questions and we can provide any additional clarifications to help finalize your assessment and rating of our paper.
>
> ---
>
> ### **W1:  Limited exploration of open-source models**
> **A:** Thank you for your advice! For open-source models, we considered representative models such as Mistral 7*8B, Llama3-70B, and CogAgent. We chose CogAgent as a representative among multimodal open-source models since it is optimized for web and mobile scenarios, and has been adopted by both the VisualWebArena [1] paper and the OpenInterpreter [2] project from the open-source community. To enhance coverage and gain additional insights, we have added experiments with the latest batch of open-source multimodal models. Specifically, we tested InternVL2, MiniCPM-V-2.6, and Llava-OneVision models, and will include the full results in the Appendix.
>
> | Model | OS | Office | Daily | Professional | Workflow | Overall |
> |-------|----|----|----|----|----|----|
> | CogAgent | 4.17% | 0.85% | 2.71% | 0.00% | 0.00% | 1.11% |
> | InternVL2 | 12.50% | 1.87% | 2.71% | 8.16% | 0.99% | 3.33% |
> | MiniCPM-V-2.6 | 8.33% | 2.72% | 1.42% | 0.00% | 0.63% | 1.88% |
> | Llava-OneVision | 8.33% | 2.72% | 2.71% | 0.00% | 1.62% | 2.42% |
> | GPT-4o-mini | 12.50% | 3.58% | 3.99% | 4.08% | 1.62% | 3.77% |
>
> Also note that with the open-source models getting better soon later this year, we will keep updating the results of these more powerful open-source models.
>
> [1]. Koh, J. Y., Lo, R., Jang, L., Duvvur, V., Lim, M. C., Huang, P. Y., ... & Fried, D. (2024). Visualwebarena: Evaluating multimodal agents on realistic visual web tasks. arXiv preprint arXiv:2401.13649.
>
> [2]. https://github.com/OpenInterpreter/open-interpreter
>
> ---
>
> ### **W2: Insufficient analysis of the experimental results**
>
> #### **W2.1: Why does adding visual information not perform better than text information only?**
> **A:**
> 1. **Visual information can be distracting.**
> We have tried several strategies to check the robustness of visual information, such as changing the window's position, changing the window's size to the minimum, opening some irrelevant software, and maximizing them to clutter the screen. The results reported in Sec. 5 reveal that the agent performances with visual information are sensitive to these disturbances and the SRs vary fiercely. As for text information, these methods will not affect the overall structure of the a11y tree except for changing some element properties on it.
> 2. **After multimodal learning, GPT-4 may have lost some of its text reasoning capabilities, which is essential for agent tasks**
> In pure text a11y tree conditions, GPT-4 achieved state-of-the-art performance. For our settings with both image and text inputs (screenshot, screenshot + a11y tree, som), we used the GPT-4V model. Previous work[1] has also pointed out that multimodal post-training can lead to some loss in pure text processing abilities. This could potentially explain the performance differences we observed.
>
>     [1]. Lin, J., Yin, H., Ping, W., Molchanov, P., Shoeybi, M., & Han, S. (2024). Vila: On pre-training for visual language models. In Proceedings of the IEEE/CVF Conference on Computer Vision and Pattern Recognition (pp. 26689-26699).
>
> #### **W1.2: Why the third method (combining both the A11y tree and screenshots) does not outperform the single strategy, as seen in Daily tasks and Office tasks?**
> **A:** In our response to W1.1, we explained the considerations of task formulation and model capability. For instance, while the model successfully extracts accurate coordinates from accessibility (a11y) input, the presence of additional visual input may lead it to estimate coordinates from the screenshot rather than the a11y tree. This shift in approach can result in the model failing to accurately locate the correct element. We will include new case analyses in the appendix to further illustrate this phenomenon.
>
>
> #### **W1.3: Why does the trend of Gemini differ from the trend of GPT4-V?**
> **A:** While we didn't analyze every single model individually, we carefully examined cases in our paper to understand why Claude and GPT-4V exhibit different performance trends. Our findings reveal that Claude excels at providing satisfactory high-level solutions, but its grounding ability sometimes contains hallucinations in the details.
>
> ---
>
> ### **W3: Missed citation.**
> **A:** Thank you for pointing that out! In the new version of the manuscript, we have added ScreenAgent to the comparison table for analysis and comparison as shown below.
>
> | | # Instances (# Templates) | Control. Exec. Env.? | Environment Scalability? | Multimodal Support? | Cross-App? | Intermediate Init. State? | # Exec.-based Eval. Func. |
> |-|:-:|:-:|:-:|:-:|:-:|:-:|:-:|
> …
> | SCREENAGENT  | 70                        | ✗                   | -                       | ✓                  |  ✗    | ✓     | 0 |
> …
> | OSWORLD           | 369                       | Computer            | ✓                       | ✓                   | ✓         | ✓                        | 134                       |

---

> ### Author Response · Authors · 2024-08-26
>
> Dear reviewer bitD,
>
> As the open discussion period draws to a close in a few days, we wanted to check back to see whether you have any remaining concerns. We believe that we have sufficiently responded to your earlier queries on various aspects of this work, and we provide a short summary here for your convenience:
>
> 1. We have addressed your concern about the limited exploration of open-source models by including additional experiments with InternVL2, MiniCPM-V-2.6, and Llava-OneVision models. We have provided the results and will include the full details in the Appendix of our revised paper.
>
> 2. We have provided a more thorough analysis of our experimental results, including:
>    a) Explaining why adding visual information sometimes does not perform better than text information only.
>    b) Clarifying why combining both the A11y tree and screenshots does not always outperform single strategies in certain tasks.
>    c) Discussing the different performance trends observed between GPT-4V and Gemini.
>
> 3. We have addressed the missed citation by adding ScreenAgent to our comparison table and including an analysis of it in relation to our work.
>
> We have updated our manuscript to reflect these changes and additions. Please let us know if/how we can address any remaining concerns, and we are grateful for any additional feedback and suggestions!
>
> Best regards,
> Authors

---

### Author Rebuttal · Authors · 2024-08-17

We sincerely thank all reviewers for their thorough and constructive feedback. We are delighted that our work was recognized as "a highly engaging read" with "valuable content" (R#3), "the first benchmark providing real environment and execution-based evaluation for OS level tasks" (R#2), addressing "a very challenging and serious question in Human-Computer Interaction" (R#1), and potentially "an important contribution to the community" with a framework and data that is "valuable and even outweighs the benchmark and results" (R#4).

We have addressed the specific queries and points raised by each reviewer in our individual responses. We have also conducted additional experiments and revised our manuscript to address the reviewers' comments (all revisions will be highlighted in the camera-ready version). The key updates are summarized as follows:

- **Expanded experiments:** We've included additional open-source multimodal models (InternVL2, MiniCPM-V-2.6, and Llava-OneVision) in our evaluation (R#1).
- **Detailed analysis:** We've provided a more thorough analysis of our experimental results, including explanations for observed performance differences between models with and without vision capabilities (R#1, R#2).
- **Related work:** We've expanded our discussion of related work to include additional embodied benchmarks such as ScreenAgent (R#1), Minedojo, Alfred, PCA-Bench, and EgoThink (R#2).
- **Evaluation clarification:** We've added more details on our execution-based evaluation process, particularly for tasks like "Make the background transparent" (R#2).
- **Human evaluation details:** We've included more information about our human evaluation process, including the number of annotators and man-hours spent (R#3).
- **Accessibility improvements:** We've highlighted our efforts to improve and standardize accessibility tree information across different operating systems (R#3).
- **Additional baselines:** We've conducted and included results for "Human plan, model act" and "Model plan, human act" experiments to provide a more comprehensive analysis of model capabilities (R#4).
- **POMDP formulation clarification:** We've provided more explanation on our task formulation, including the role of instructions and rewards (R#4).

Once again, we sincerely thank all reviewers for their valuable contributions towards enhancing our manuscript. We believe these revisions and additional experiments have significantly strengthened our paper. If there's any need for further clarification to help finalize the assessment of our work, please don't hesitate to let us know.

Thank you for your review!

---

### Comment · Area_Chair_AJ6J · 2024-08-25

Dear Reviewers,

Thank you for taking the time to review this submission. :)

This is a gentle reminder regarding the reviewer-author discussion.

Please respond to the author's rebuttal at your earliest convenience, especially if you have any points of disagreement.

The deadline is August 31 at 11:59 PM AoE!

Early discussion is always appreciated.

Best,

AC

---

### Comment · Area_Chair_AJ6J · 2024-08-29

Dear Reviewers,

We are just three days away from the end of the discussion period.

Please take a moment to review the author's responses and share any additional feedback you may have.

Best regards,

Your AC

---

### Decision · Program_Chairs · 2024-09-26

**Decision:**

Accept (Poster)

**Comment:**

This work introduces OSWorld, a benchmark designed to evaluate Multimodal agents in a real computer environment. I recommend the acceptance of this submission for the following reasons:

* Realistic computer environments, offering both action and observation spaces tailored to Multimodal agents.
* Comprehensive set of 369 tasks based on real user scenarios, covering web and desktop applications, file I/O, and multi-application workflows.
* Extensive analysis of various multi-modal agents, highlighting a significant room for improvement.

Overall, all reviewers agreed that this is a very solid submission and the authors also handled concerns from reviewers during the discussion period. I recommend acceptance.